# A predictive index for health status using species-level gut microbiome profiling

Vinod K. Gupta[1,2], Minsuk Kim [1,2], Utpal Bakshi [1,2], Kevin Y. Cunningham[3,4], John M. Davis III[5], Konstantinos N. Lazaridis [6], Heidi Nelson[7], Nicholas Chia [1,2] & Jaeyun Sung [1,2,5 ✉]

Providing insight into one's health status from a gut microbiome sample is an important clinical goal in current human microbiome research. Herein, we introduce the Gut Microbiome Health Index (GMHI), a biologically-interpretable mathematical formula for predicting the likelihood of disease independent of the clinical diagnosis. GMHI is formulated upon 50 microbial species associated with healthy gut ecosystems. These species are identified through a multi-study, integrative analysis on 4347 human stool metagenomes from 34 published studies across healthy and 12 different nonhealthy conditions, i.e., disease or abnormal bodyweight. When demonstrated on our population-scale meta-dataset, GMHI is the most robust and consistent predictor of disease presence (or absence) compared to α-diversity indices. Validation on 679 samples from 9 additional studies results in a balanced accuracy of 73.7% in distinguishing healthy from non-healthy groups. Our findings suggest that gut taxonomic signatures can predict health status, and highlight how data sharing efforts can provide broadly applicable discoveries.

[1] Microbiome Program, Center for Individualized Medicine, Mayo Clinic, Rochester, MN 55905, USA. [2] Division of Surgery Research, Department of Surgery, Mayo Clinic, Rochester, MN 55905, USA. [3] Graduate Research Education Program (GREP), Mayo Clinic, Rochester, MN 55905, USA. [4] Department of Computer Science and Engineering, University of Minnesota Twin-Cities, Minneapolis, MN 55455, USA. [5] Division of Rheumatology, Department of Medicine, Mayo Clinic, Rochester, MN 55905, USA. [6] Division of Gastroenterology and Hepatology, Mayo Clinic College of Medicine and Science, Rochester, MN 55905, USA. [7] Emeritus Chair, Department of Surgery, Mayo Clinic, Rochester, MN 55905, USA. ✉email: Sung.Jaeyun@mayo.edu

Recent advances in the field of human gut microbiome research have revealed significant associations and potential mechanistic insights regarding a vast array of complex, chronic diseases, including cancer[1,2], auto-immune disease[3–5], and metabolic syndrome[6–8]. Undoubtedly, the many microbiome studies that focused on disease contexts have been essential for elucidating underlying pathophysiological mechanisms, and for developing potential intervention strategies. As researchers uncover more details regarding which gut commensals may play a significant role in host health and disease, a promising translational application of this knowledge would be towards developing analytical tests or quantitative methods that provide indication of one's health based upon a gut microbiome snapshot[9–12].

The creation of algorithm-driven markers for detecting early signs of disease prior to the occurrence of specific, diagnosable symptoms, especially from biospecimens that can be collected regularly and noninvasively, is an exciting avenue forward for personalized medicine[13–16]. To this end, current gut microbiome science could play a significant role in the development of stool-based tests for dynamically monitoring and predicting wellness. We can even imagine a hypothetical scenario wherein one can, through continuous monitoring, be able to detect significant changes or abnormalities in comparison to her/his normal baseline measurements; in turn, this will be the cue for additional tests or lifestyle interventions. In this sense, a proof of principle of collecting diverse longitudinal biomolecular data from human subjects, and translating the corresponding complex datasets into actionable possibilities, has been previously demonstrated by Price et al.[17]

Since new insights regarding the gut microbiome and human health need to be reliable and robust across a wide range of human subjects and conditions, population-level analyses can serve as an important platform for the discovery of broadly applicable principles and methodologies[18–22]. Traditionally, the collection of a substantial amount of microbiome samples (on the order of hundreds to thousands) for large cohort investigations has been undertaken at well-funded, major research centers and consortiums, mainly due to the prohibitive costs and/or lack of infrastructure for lone laboratories. However, with the recent progress in the call for broad data sharing policies and practices[23], conducting analyses by crowdsourcing or repurposing data from existing published studies have already begun to play important roles in either hypothesis generation or validation in microbiome research[1,24–27]. As much 16s rRNA gene amplicon and shotgun metagenomic sequencing data are now readily available from multiple, independent studies conducted around the world, integration by pooling these datasets would provide a promising strategy to study health- and disease-associated signatures at large scale, as well as to gain new, holistic insights not offered by smaller, individual studies.

In this study, we introduce the Gut Microbiome Health Index (GMHI), a robust index for evaluating health status (i.e., degree of presence/absence of diagnosed disease) based on the species-level taxonomic profile of a stool shotgun metagenome (gut microbiome) sample. GMHI determines the likelihood of having a disease, independent of the clinical diagnosis; this is done so by comparing the relative abundances of two sets of microbial species associated with good and adverse health conditions, which are identified from an integrated dataset of 4347 publicly available, human stool metagenomes pooled across multiple studies encompassing various disease states. By applying GMHI to each sample in our population-scale meta-dataset, we found that GMHI distinguishes healthy from nonhealthy groups far better than ecological indices (e.g., Shannon diversity and richness) generally considered as markers for gut health and dysbiosis.

Intra-study comparisons of stool metagenomes between healthy and nonhealthy phenotypes demonstrate that GMHI is the most robust and consistent predictor of health. Finally, to confirm that GMHI classification accuracy was not a result of over-fitting on the discovery cohort, we test our approach on a validation set of 679 samples from eight additional published studies and one new cohort (this study). We find that GMHI not only demonstrates strong reproducibility in stratifying healthy and nonhealthy groups, but also outperforms α-diversity indices.

## Results

**A meta-dataset of integrated human stool metagenomes.** An overview of our multi-study integration approach, wherein we acquired 4347 raw shotgun stool metagenomes (2636 and 1711 metagenomes from healthy and nonhealthy individuals, respectively) from 34 independently published studies, is depicted in Fig. 1a. In this study, "healthy" subjects were defined as those who were reported as not having any overt disease nor adverse symptoms at the time of the original study; alternatively, "nonhealthy" subjects were defined as those who were clinically diagnosed with a specific disease, or determined to have abnormal bodyweight based on body mass index (BMI). Accordingly, 1711 stool metagenomes from patients across 12 different disease or abnormal bodyweight conditions were pooled together into a single aggregate nonhealthy group. (Our sample selection criteria are described in "Methods." Importantly, all metagenomes were reprocessed uniformly, thereby removing a major nonbiological source of variance among different studies, as previously demonstrated[28].) A description of the studies whose human stool metagenomes were collected and processed through our computational pipeline is provided in Table 1. We note that, in order to eventually identify features of the gut microbiome associated exclusively with health, it is important to be disease agnostic by considering a broad range of nonhealthy phenotypes. We provide all subjects' phenotype, age, sex, BMI, and other questionnaire measures (as provided in their respective original study) in Supplementary Data 1. Along with the additional 679 stool metagenome samples used for validation purposes (discussed below), this study provides the largest metagenomic (pooled) analysis of the human gut microbiome to date, in regards to the number of samples, phenotypes, and studies.

Importantly, we chose to integrate datasets from independent studies for two notable advantages: (i) the expansion of sample number could help to amplify the primary biological signal of interest and improve statistical power[29,30]; and (ii) the identified health/disease-associated signatures could encompass a wide range of heterogeneity across different sources and conditions (e.g., host genetics, geography, dietary and lifestyle patterns, age, sex, birth mode, early life exposures, medication history), thereby helping to identify robust findings despite systematic biases from batch effects or other confounding factors[28,31].

After downloading, reprocessing, and performing quality filtration on all raw metagenomes, species-level taxonomic profiling was carried out using the MetaPhlAn2 pipeline[32] ("Methods"). Of note, our study was mainly conducted upon species-level taxonomy information to obtain as much precise and comprehensive information about the gut microbiome as possible. A total of 1201 species were detected in at least one metagenome sample; after removing viruses, and species that were rarely observed or of unknown/unclassified identity ("Methods"), 313 species remained for further analysis (Fig. 1b and Supplementary Data 2; a phylogenetic tree showing the evolutionary relationships among these species is shown in Supplementary Fig. 1). Interestingly, six species (*Bacteroides ovatus*, *Bacteroides uniformis*, *Bacteroides vulgatus*, *Faecalibacterium*

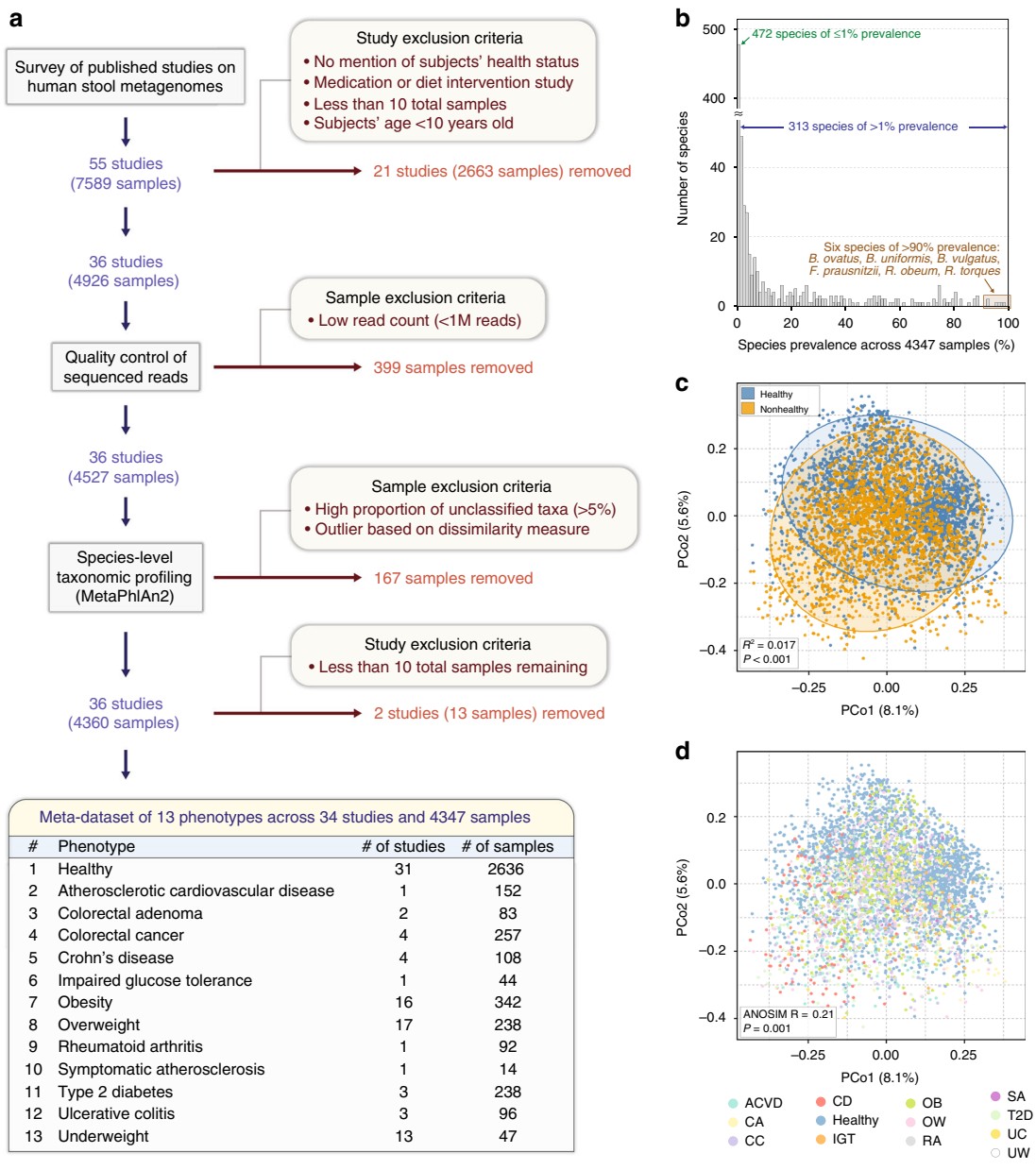

**Fig. 1 Multi-study integration of human stool metagenomes leads to a meta-dataset of healthy and nonhealthy gut microbiomes. a** Schematic overview. A survey was conducted in PubMed and Google Scholar to search for published studies with publicly available human stool metagenome (gut microbiome) samples from healthy and nonhealthy individuals. The initial collection of stool metagenomes consisted of 7589 samples from 55 independent studies. All samples (.fastq files) were downloaded and reprocessed uniformly using identical bioinformatics methods. After quality control of sequenced reads, species-level taxonomic profiling was then performed. Studies and metagenome samples were removed based on several exclusion criterias. Finally, a total of 4347 samples (2636 and 1711 metagenomes from healthy and nonhealthy individuals, respectively) from 34 studies ranging across healthy and 12 nonhealthy phenotypes were assembled into a meta-dataset for downstream analyses. **b** Distribution of microbial species' prevalence across the 4347 stool metagenome samples in the meta-dataset. After removing viruses, unknown/unclassified species-level entities, and rarely observed species (i.e., detected <1% of all samples), 313 species remained for further analyses. **c** Principal coordinates analysis (PCoA) ordination plot based on Bray–Curtis distances shows that healthy (blue; $n = 2636$) and nonhealthy (orange; $n = 1711$) groups have significantly different distributions of gut microbiome profiles according to PERMANOVA ($R^2 = 0.017$, $P < 0.001$) after adjusting for each sample's study origin. Each point corresponds to a sample. Ellipses correspond to 95% confidence regions. **d** In an identical PCoA plot, each color represents one of the 13 different phenotypes of health or disease. Among- and within-group dissimilarities differ only weakly (ANOSIM $R = 0.21$, $P = 0.001$).

*prausnitzii*, *Ruminococcus obeum*, and *Ruminococcus torques*) were of high prevalence (i.e., detected in >90% of all 4347 samples).

**Healthy and nonhealthy guts show species-level differences.** The overall ecology of the gut microbiome has often been associated with host health[8,33–35]. Using species-level relative abundance (i.e., proportion) profiles, we examined the differences in

gut microbial diversity between the healthy and nonhealthy groups. First, when using principal coordinate analysis (PCoA) ordination, we identified a significant difference between the distributions of these two groups (permutational multivariate analysis of variance (PERMANOVA), $R^2 = 0.02$, $P < 0.001$; Fig. 1c). In the same PCoA plot in which the healthy and 12 nonhealthy phenotypes were presented simultaneously (Fig. 1d),

**Table 1 Human stool metagenome datasets analyzed in this study.**

| Author (year) | Healthy (n) | Disease (n) | Obese[a] (n) | Overweight[a] (n) | Underweight[a] (n) | Total nonhealthy (n) | Total from Study (n) | Sequencing platform | Geography (ethnicity/race[b]) |
|---|---|---|---|---|---|---|---|---|---|
| Backhed (2015) | 89 | – | 0 | 0 | 0 | 0 | 89 | Illumina HiSeq 2000 | Denmark |
| Feng (2015) | 21 | CA (42), CC (43) | 20 | 20 | 0 | 125 | 146 | Illumina HiSeq 2000 | Austria (Caucasian) |
| Guthrie (2017) | 16 | – | 0 | 3 | 1 | 4 | 20 | Illumina Hiseq 2500 | USA |
| Hall (2017) | 3 | CD (6), UC (5) | 0 | 0 | 0 | 11 | 14 | Illumina HiSeq 2000 | USA |
| He (2017) | 39 | CD (47) | 1 | 7 | 5 | 60 | 99 | Illumina Hiseq 2000 | China |
| Huttenhower (2012) and Lloyd-Price (2017)[c] | 222 | – | 0 | 0 | 0 | 0 | 222 | Illumina Genome Analyzer II | USA |
| Jie (2017) | 75 | ACVD (152) | 8 | 40 | 7 | 207 | 282 | Illumina HiSeq 2000 | China |
| Karlsson (2012) | 8 | SA (14) | 0 | 3 | 0 | 17 | 25 | Illumina Hiseq 2000 | Sweden |
| Karlsson (2013) | 18 | IGT (44), T2D (48) | 6 | 17 | 1 | 116 | 134 | Illumina HiSeq 2000 | Sweden |
| Le Chatelier (2013) | 39 | OB (69) | 0 | 4 | 1 | 74 | 113 | Illumina Genome Analyzer II | Denmark (Northern European) |
| Li (2017) | 35 | | 0 | 0 | 0 | 0 | 35 | Illumina HiSeq 2500 | China |
| Lim (2014) | 16 | | 0 | 0 | 0 | 0 | 16 | Illumina HiSeq 2000 | South Korea |
| Liu (2016) | 107 | | 0 | 0 | 0 | 0 | 107 | Illumina Hiseq 4000 | Mongolia, China (Mangolian) |
| Liu (2017) | 101 | OB (104) | 0 | 0 | 0 | 104 | 205 | Illumina HiSeq 2500 | China |
| Louis (2016) | 0 | OB (14) | 0 | 0 | 0 | 14 | 14 | Illumina HiSeq 2500 | Germany |
| Nielsen (2014) | 58 | CD (12), UC (66) | 71 | 19 | 1 | 169 | 227 | Illumina Genome Analyzer IIx | Denmark (Danish and Spanish) |
| Nishijima (2016) | 26 | – | 1 | 2 | 2 | 5 | 31 | Illumina MiSeq | Japan |
| Obregon-Tito (2015) | 20 | OB (3) | 0 | 9 | 3 | 15 | 35 | Illumina HiSeq 2500 | Peru (Matses and Tunapuco), USA |
| Palleja (2016) | 0 | OB (13) | 0 | 0 | 0 | 13 | 13 | Illumina HiSeq 2000 | Denmark |
| Petersen (2017) | 33 | – | 0 | 0 | 0 | 0 | 33 | Illumina NextSeq 500 | USA |
| Qin (2012) | 61 | T2D (171) | 2 | 48 | 16 | 237 | 298 | Illumina Genome Analyzer II | China |
| Qin (2014) | 92 | – | 0 | 7 | 2 | 9 | 101 | Illumina HiSeq 2000 | China |
| Rampelli (2015) | 30 | – | 0 | 0 | 0 | 0 | 30 | Illumina Genome Analyzer IIx | Tanzania (Hadza) |
| Raymond (2015) | 19 | – | 0 | 4 | 0 | 4 | 23 | Illumina HiSeq 1000 | Canada |
| Sankaranarayanan (2015) | 1 | T2D (19) | 12 | 4 | 0 | 35 | 36 | Illumina HiSeq 2000 | USA (Arapaho and Cheyenne) |
| Schirmer (2016) | 467 | – | 0 | 0 | 0 | 0 | 467 | Illumina HiSeq 2000 | Netherlands (Western European) |
| Schirmer (2018) | 17 | CD (43), UC (25) | 0 | 0 | 0 | 68 | 85 | Illumina HiSeq 2000 | USA |
| Tanca (2017) | 13 | – | 0 | 0 | 0 | 0 | 13 | Illimina HScanSQ | Italy (Sardinian) |
| Vogtmann (2016) | 30 | CC (49) | 8 | 12 | 1 | 70 | 100 | Illumina HiSeq 2000 | USA |

**Table 1 (continued)**

| Author (year) | Healthy (n) | Disease (n) | Obese[a] (n) | Overweight[a] (n) | Underweight[a] (n) | Total nonhealthy (n) | Total from Study (n) | Sequencing platform | Geography (ethnicity/race[b]) |
|---|---|---|---|---|---|---|---|---|---|
| | | | | | | | | | USA (non-hispanic white, non-hispanic black, and others) |
| Yu (2015) | 0 | CC (70) | 0 | 0 | 0 | 70 | 70 | Illumina HiSeq 2000 | China |
| Zeevi (2015) | 883 | - | 0 | 0 | 0 | 0 | 883 | Illumina NextSeq 500 Illumina HiSeq 2500 | Israel |
| Zeller (2014) | 42 | CA (41), CC (95) | 6 | 12 | 1 | 155 | 197 | Illumina MiSeq | France and Germany |
| Zhang (2015) | 55 | RA (92) | 4 | 27 | 6 | 129 | 184 | Illumina HiSeq 2000 | China |
| Total | 2636 | 1287 | 139 | 238 | 47 | 1711 | 4347 | - | - |

*ACVD* atherosclerotic cardiovascular disease, *CA* colorectal adenoma, *CC* colorectal cancer, *CD* Crohn's disease, *IGT* impaired glucose tolerance, *OB* obesity, *RA* rheumatoid arthritis, *SA* symptomatic atherosclerosis, *T2D* type 2 diabetes, *UC* ulcerative colitis.
[a]Reclassification of previously reported healthy samples to abnormal bodyweight condition according to reported BMI (when provided in the original study).
[b]As provided in the original study.
[c]Samples combined from both phases of the Human Microbiome Project (HMP1 and HMP1-II).

we found only a weak difference among groups (analysis of similarities (ANOSIM) $R = 0.21$, $P = 0.001$).

**Design rationale for a GMHI.** We envision that the most intuitive way to determine how closely one's microbiome resembles that of a healthy (or nonhealthy) population is to quantify the balance between health-associated microbes relative to disease-associated microbes. Therefore, we propose an index in the form of a rational equation (and thereby yielding a dimensionless quantity) between two sets of microbial species: those that are more frequently observed in healthy compared to nonhealthy groups vs. those that are less frequently observed in healthy compared to nonhealthy groups. Next, we use our compendium of publicly available datasets, which were derived from healthy and nonhealthy human subjects, to identify these two sets of species. Finally, with these species, we tune the parameters of a predefined formula, as well as evaluate its classification accuracy. The logical rationale of each major step during the development, demonstration, and validation of our index for predicting general health status (i.e., presence/absence of diagnosed disease) from a gut microbiome sample is detailed below. In addition, a step-by-step protocol is provided in "Methods."

**A prevalence-based strategy to identify health-associated microbes.** We set out to identify distinct microbial species associated with healthy ($H$) and nonhealthy ($N$) groups. Here, we use a prevalence-based strategy to deal with the sparse nature of microbiome datasets. For this, we first determine $p_{H,m}$ and $p_{N,m}$, or the prevalence of microbial species $m$ in $H$ and $N$, respectively (prevalence is defined as the proportion of samples in a given group wherein $m$ is considered "present," i.e., relative abundance $\geq 1.0 \times 10^{-5}$.) Next, for comparing the two prevalences in $H$ and $N$, we apply the following two criteria: prevalence fold change $f_m^{H,N}$ and prevalence difference $d_m^{H,N}$, defined as $\frac{p_{H,m}}{p_{N,m}}$ and $p_{H,m} - p_{N,m}$, respectively. A significant effect size between the two prevalences is considered to exist if both criteria satisfy (predetermined) minimum thresholds for prevalence fold change $\theta_f$ and prevalence difference $\theta_d$ (how we determine the best pair of thresholds is described below). For all detectable microbial species that simultaneously satisfy $f_m^{H,N} \geq \theta_f$ and $d_m^{H,N} \geq \theta_d$, we term these species observed more frequently in $H$ (than in $N$) as "health-prevalent" species $M_H$. Analogously, we identify "health-scarce" species $M_N$, or the species observed less frequently in $H$ (than in $N$), as those that satisfy $f_m^{N,H} \geq \theta_f$ and $d_m^{N,H} \geq \theta_d$, where $f_m^{N,H}$ and $d_m^{N,H}$ is defined as $\frac{p_{N,m}}{p_{H,m}}$ and $p_{N,m} - p_{H,m}$, respectively. In this regard, the species that are eventually chosen to compose $M_H$ and $M_N$ are both dependent on $\theta_f$ and $\theta_d$. An important strength of our prevalence-based strategy for identifying microbial associations is that it does not calculate or compare averages of measurements taken from various sources, which is challenging to justify when biological and technical heterogeneity could vary greatly across independent studies. Rather, our approach compares frequencies of a signal—on a sample-by-sample basis—between two groups, and represents a strategy more applicable to the context of integrating high-throughput data from different studies. Importantly, we chose to simultaneously test two thresholds, rather than one, in order to increase our confidence in the robustness of $M_H$ and $M_N$, as well as to overcome biases that can occur from using only one type of threshold.

**Collective abundances of two sets of microbial taxonomies.** Having a strategy to identify microbial species associated with healthy (i.e., health-prevalent species $M_H$) and nonhealthy (i.e., health-scarce species $M_N$), we next couple these two species sets

with a computational procedure that quantifies the presence/absence of diagnosed disease for any gut microbiome sample. To this end, we developed the following mathematical formula: for species of $M_H$ in sample $i$, their "collective abundance" $\psi_{M_H,i}$ is defined as

$$\psi_{M_H,i} = \frac{R_{M_H,i}}{|M_H|} \sum_{j \in I_{M_H}} \left| n_{j,i} \ln\left(n_{j,i}\right) \right|, \qquad (1)$$

where $R_{M_H,i}$ is the richness of $M_H$ species in sample $i$, $|M_H|$ is the set size of $M_H$, $I_{M_H}$ is the index set of $M_H$, and $n_{j,i}$ is the relative abundance of species $j$ in sample $i$. In brief, $\psi_{M_H,i}$ is the product of the (i) richness, that is, the numeric count of "present" taxonomies, of $M_H$ species; and (ii) the geometric mean of their relative abundances. Full details on the physical meaning and derivation of $\psi_{M_H,i}$ are described in "Methods." For the species of $M_N$ in the same sample $i$, their "collective abundance" $\psi_{M_N,i}$ can be defined analogously. Next, the collective abundances of species in sets $M_H$ and $M_N$ in sample $i$ are compared using the ratio of $\psi_{M_H,i}$ to $\psi_{M_N,i}$ as

$$h_{i,M_H,M_N} = \log_{10}\left(\frac{\psi_{M_H,i}}{\psi_{M_N,i}}\right), \qquad (2)$$

where $h_{i,M_H,M_N}$ denotes the degree to which sample $i$ portrays the collective abundance of $M_H$ to that of $M_N$. More specifically, a positive or negative $h_{i,M_H,M_N}$ suggests that sample $i$ is characterized more by the microbes of $M_H$ or $M_N$, respectively; an $h_{i,M_H,M_N}$ of 0 indicates that there is an equal balance of both species sets. Full details on the derivation of $h_{i,M_H,M_N}$ are provided in "Methods."

**Determining health-prevalent and health-scarce species.** The minimum thresholds $\theta_f$ and $\theta_d$ for prevalence fold change and prevalence difference, respectively, are used to control for the number of health-prevalent species $M_H$ and health-scarce species $M_N$; species that simultaneously satisfy the two types of thresholds are selected to be included in one of either group. Afterwards, $M_H$ and $M_N$ is provided as input features for $\psi_{M_H,i}$ and $\psi_{M_N,i}$, respectively, and for the calculation of $h_{i,M_H,M_N}$, which in turn can classify stool metagenome sample $i$ as healthy (i.e., $h_{i,M_H,M_N} > 0$), nonhealthy (i.e., $h_{i,M_H,M_N} < 0$), or neither (i.e., $h_{i,M_H,M_N} = 0$). Lastly, $h_{i,M_H,M_N}$ is tested on all 4347 stool metagenomes in our meta-dataset to find the balanced accuracy $\chi_{M_H,M_N}$, that is, an average of the proportions of healthy and nonhealthy samples that were correctly classified, or

$$\chi_{M_H,M_N} = \frac{P\left(h_{i,M_H,M_N} > 0 | i \in H\right) + P\left(h_{i,M_H,M_N} < 0 | i \in N\right)}{2}, \qquad (3)$$

where $P(h_{i,M_H,M_N} > 0 | i \in H)$ is the proportion of samples in the healthy group $(H)$ whose $h_{i,M_H,M_N}$s are positive, and $P(h_{i,M_H,M_N} < 0 | i \in N)$ is the proportion of samples in the nonhealthy group $(N)$ whose $h_{i,M_H,M_N}$s are negative. We determine the final, optimal sets of health-prevalent and health-scarce species (and their corresponding prevalence thresholds) as those that result in the highest balanced accuracy $\chi_{M_H,M_N}^{max}$. This was done accordingly: after systematically testing across a range of two different thresholds (every pair of $\theta_f$ and $\theta_d$ gives different sets of $M_H$ and $M_N$, and in turn, a different $\chi_{M_H,M_N}$), we found $\chi_{M_H,M_N}^{max}$ to be 69.7% when $\theta_f$ and $\theta_d$ were set to 1.4 and 10%, respectively (Supplementary Table 1). When applying the same approach for abundance profiles of all other taxonomic ranks, as

well as of MetaCyc pathways, the highest accuracies found in these were as follows: Phylum, 42.1%; Class, 60.1%; Order, 62.4%; Family, 67.2%; Genus, 68.2%; and MetaCyc pathway, 59.4% (Supplementary Table 2). As evidenced by these results, taxonomic species shows the best classification accuracy. In addition, performing our method in 10-fold cross-validation using species-level abundances resulted in an accuracy of 69.6% (Supplementary Table 3), which is nearly identical to the balanced accuracy of 69.7% achieved by testing on the set of samples from which the classifier was derived. Lastly, in Supplementary Fig. 2, we show a sensitivity analysis of how the balanced accuracy $\chi$ changes with respect to the species' prevalence thresholds $\theta_f$ and $\theta_d$.

We identified 50 microbial species that satisfy both of the aforementioned thresholds for the highest balanced accuracy; among these 50 species, 7 and 43 comprise the health-prevalent and health-scarce groups, respectively (Table 2). Interestingly, we found higher relative abundance levels of health-prevalent and health-scarce species in the healthy and nonhealthy group, respectively (Supplementary Fig. 3). Furthermore, in Supplementary Fig. 4, we show the prevalence of these species in case (i.e., nonhealthy) and/or control (i.e., healthy) for the 34 published studies upon which our stool metagenome meta-dataset was derived. Despite the heterogeneity and unevenness in prevalences across all studies, we found that, by and large, health-prevalent and health-scarce species were observed more frequently in the control and case samples, respectively. We report known associations between health-prevalent/-scarce species and health/disease in Supplementary Data 3.

Henceforth, we term the ratio $h_{i,M_H,M_N}$ between these two groups of 7 Health-prevalent and 43 Health-scarce species as the GMHI. GMHI is a dimensionless metric designed to simplify the accumulation of Health-prevalent and Health-scarce species observed to be present in a microbiome sample. In practice, GMHI indicates the degree to which a subject's stool metagenome sample portrays microbial taxonomies associated with either healthy or nonhealthy. An additional summary of the design process of GMHI, enumerated in three key steps, is provided in Supplementary Note 1.

Analogous to the example mentioned above, a positive or negative GMHI allows the sample to be classified as healthy or nonhealthy, respectively; a GMHI of 0 indicates an equal balance of Health-prevalent and Health-scarce species, and thereby classified as neither. Therefore, GMHI is especially favorable in terms of the simplicity of the decision rule and the biological interpretation regarding the two sets of microbes involved in classification. Importantly, our metric can be measured on a per-sample basis, requires very little parameter tuning, and foregoes the use of qualitative assessments, for example, "low" or "high" α-diversity. Furthermore, we found no significant association between library size and GMHI (mixed-effects linear regression, $P = 0.45$; Supplementary Fig. 5), and that, by and large, the distributions of the index for healthy individuals do not vary much between studies (Supplementary Fig. 6).

**GMHI is associated with high-density lipoprotein cholesterol.** To see whether GMHI can encompass certain physiological features of health, we looked for statistical associations between GMHI and well-recognized components of physiological wellness from clinical lab tests. More specifically, we searched for correlations with GMHI and the following, as reported in their original studies: circulating blood concentrations of fasting blood glucose (from 785 subjects), triglycerides (from 915 subjects), total cholesterol (from 521 subjects), low-density lipoprotein cholesterol (LDLC; from 848 subjects), and high-density lipoprotein

**Table 2 Microbial species of the Health-prevalent and Health-scarce groups.**

| Group[a] | Species name[b] | Prevalence in healthy samples, $P_H$ (%) | Prevalence in nonhealthy samples, $P_N$ (%) | Difference, $P_H - P_N$ (%) | Fold change[c], $P_H/P_N$ or $P_N/P_H$ |
|---|---|---|---|---|---|
| H+ | Alistipes senegalensis | 58.5 | 39.9 | 18.5 | 1.5 |
| H+ | Bacteroidales bacterium ph8 | 73.1 | 51.4 | 21.8 | 1.4 |
| H+ | Bifidobacterium adolescentis | 68.2 | 46.4 | 21.9 | 1.5 |
| H+ | Bifidobacterium angulatum | 11.9 | 1.5 | 10.4 | 7.8 |
| H+ | Bifidobacterium catenulatum | 30.8 | 13.5 | 17.3 | 2.3 |
| H+ | Lachnospiraceae bacterium 8_1_57FAA | 44.8 | 26.9 | 17.9 | 1.7 |
| H+ | Sutterella wadsworthensis | 48.1 | 26.2 | 21.8 | 1.8 |
| H− | Anaerotruncus colihominis | 23.1 | 37.4 | −14.3 | 1.6 |
| H− | Atopobium parvulum | 2.3 | 12.7 | −10.4 | 5.6 |
| H− | Bifidobacterium dentium | 6.6 | 16.7 | −10.1 | 2.5 |
| H− | Blautia producta | 5.0 | 15.6 | −10.6 | 3.1 |
| H− | Candidatus Saccharibacteria TM7c | 1.9 | 13.2 | −11.3 | 6.8 |
| H− | Clostridiales bacterium 1_7_47FAA | 16.2 | 40.0 | −23.8 | 2.5 |
| H− | Clostridium asparagiforme | 20.8 | 44.1 | −23.3 | 2.1 |
| H− | Clostridium bolteae | 34.5 | 69.7 | −35.3 | 2.0 |
| H− | Clostridium citroniae | 23.9 | 50.4 | −26.5 | 2.1 |
| H− | Clostridium clostridioforme | 9.8 | 26.1 | −16.4 | 2.7 |
| H− | Clostridium hathewayi | 26.9 | 56.1 | −29.2 | 2.1 |
| H− | Clostridium nexile | 14.3 | 31.7 | −17.4 | 2.2 |
| H− | Clostridium ramosum | 10.8 | 31.9 | −21.1 | 3.0 |
| H− | Clostridium symbiosum | 20.9 | 47.5 | −26.6 | 2.3 |
| H− | Eggerthella lenta | 18.9 | 37.4 | −18.5 | 2.0 |
| H− | Erysipelotrichaceae bacterium 2_2_44A | 18.5 | 33.4 | −14.9 | 1.8 |
| H− | Flavonifractor plautii | 34.9 | 56.5 | −21.6 | 1.6 |
| H− | Fusobacterium nucleatum | 2.2 | 12.3 | −10.1 | 5.7 |
| H− | Gemella morbillorum | 1.4 | 11.7 | −10.3 | 8.1 |
| H− | Gemella sanguinis | 5.9 | 21.0 | −15.2 | 3.6 |
| H− | Granulicatella adiacens | 3.4 | 21.4 | −18.0 | 6.3 |
| H− | Holdemania filiformis | 36.9 | 57.3 | −20.4 | 1.6 |
| H− | Klebsiella pneumoniae | 17.4 | 34.3 | −16.9 | 2.0 |
| H− | Lachnospiraceae bacterium 1_4_56FAA | 18.1 | 38.0 | −19.9 | 2.1 |
| H− | Lachnospiraceae bacterium 2_1_58FAA | 33.6 | 47.1 | −13.5 | 1.4 |
| H− | Lachnospiraceae bacterium 3_1_57FAA_CT1 | 18.3 | 35.9 | −17.7 | 2.0 |
| H− | Lachnospiraceae bacterium 5_1_57FAA | 6.9 | 18.9 | −12.0 | 2.7 |
| H− | Lachnospiraceae bacterium 9_1_43BFAA | 5.5 | 16.2 | −10.7 | 3.0 |
| H− | Lactobacillus salivarius | 3.8 | 14.3 | −10.5 | 3.7 |
| H− | Peptostreptococcus stomatis | 1.9 | 13.7 | −11.8 | 7.4 |
| H− | Ruminococcaceae bacterium D16 | 16.4 | 31.3 | −15.0 | 1.9 |
| H− | Ruminococcus gnavus | 41.8 | 68.0 | −26.2 | 1.6 |
| H− | Solobacterium moorei | 7.4 | 32.2 | −24.8 | 4.4 |
| H− | Streptococcus anginosus | 10.8 | 30.2 | −19.4 | 2.8 |
| H− | Streptococcus australis | 26.9 | 42.0 | −15.0 | 1.6 |
| H− | Streptococcus gordonii | 6.7 | 22.6 | −15.9 | 3.4 |
| H− | Streptococcus infantis | 12.5 | 28.2 | −15.8 | 2.3 |
| H− | Streptococcus mitis/oralis/ pneumoniae | 12.3 | 32.9 | −20.6 | 2.7 |
| H− | Streptococcus sanguinis | 14.4 | 31.9 | −17.5 | 2.2 |
| H− | Streptococcus vestibularis | 16.2 | 30.7 | −14.5 | 1.9 |
| H− | Subdoligranulum sp. 4_3_54A2FAA | 5.9 | 16.7 | −10.8 | 2.8 |
| H− | Subdoligranulum variabile | 6.9 | 17.3 | −10.4 | 2.5 |
| H− | Veillonella atypica | 22.5 | 34.5 | −12.0 | 1.5 |

[a]H+ Health-prevalent species, H− Health-scarce species.
[b]According to the species-level taxonomies designated by MetaPhlAn2.
[c]Ratio of larger value to smaller value.

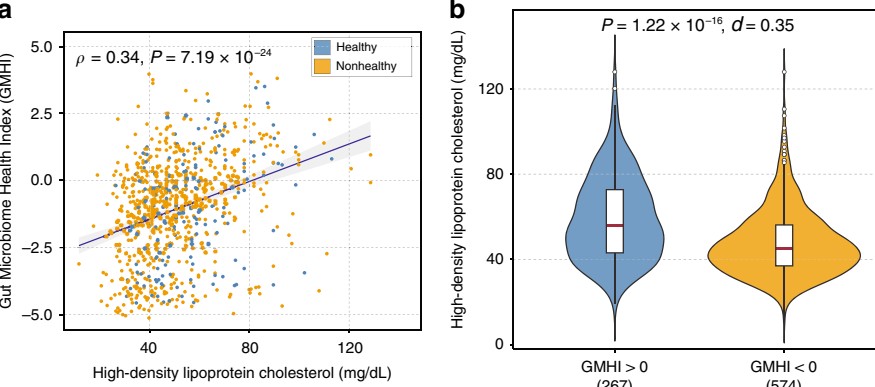

**Fig. 2 GMHI is associated with high-density lipoprotein cholesterol (HDLC). a** GMHI shows a moderately positive correlation with HDLC (Spearman's $\rho = 0.34$, 95% CI: [0.28, 0.40], $P = 7.19 \times 10^{-24}$), which is a key parameter of cardiovascular health, in 841 subjects. **b** Significantly higher abundances of HDLC were observed in subjects with positive GMHI compared to those with negative GMHI (two-sided Mann–Whitney $U$ test, $P = 1.22 \times 10^{-16}$). **d** Cliff's Delta. The sample size of each group, whose subjects' HDLC records were available in the original studies, is shown within parentheses. Standard box-and-whisker plots (e.g., center line, median; box limits, upper and lower quartiles; whiskers, 1.5× interquartile range; circles, outliers) are used to depict groups of numerical data.

cholesterol (HDLC; from 841 subjects). Of note, self-reported well-being, treatment regimens, and other questionnaire data were either not provided at all or too sparsely collected to have any practical or statistical significance. When selecting for moderate correlations or better, that is, |Spearman's $\rho| \geq 0.3$ ($P < 0.001$), we identified HDLC as the only feature that was significantly associated with GMHI ($\rho = 0.34$, 95% confidence interval (CI): [0.28, 0.40], $P = 7.19 \times 10^{-24}$; Fig. 2a). In addition, we identified significantly higher abundances of HDLC in subjects with positive GMHI compared to those with negative GMHI (Mann–Whitney $U$ test, $P = 1.22 \times 10^{-16}$; Fig. 2b). This moderately positive correlation is encouraging for linking GMHI to actual health, as HDLC in the bloodstream is commonly considered as "good" cholesterol, and could be protective against heart attack and stroke, according to the American Heart Association. In relevance to this point, a recent study by Kenny et al.[36] showed that cholesterol metabolism by gut microbes can influence serum cholesterol concentrations, and may thereby impact cardiovascular health. Overall, our finding demonstrates the importance of integrating clinical data with gut microbiome, and also hints at the possibility of GMHI serving as an effective and reliable predictor of cardiovascular health. In contrast, fasting blood glucose ($\rho = -0.06$, 95% CI: [−0.12, 0.01]), triglycerides ($\rho = -0.13$, 95% CI: [−0.19, −0.06]), total cholesterol ($\rho = 0.15$, 95% CI: [0.06, 0.23]), LDLC ($\rho = 0.09$, 95% CI: [0.03, 0.16]), and even age ($\rho = 0.04$, 95% CI: [−0.01, 0.08]) were noted to have only weak or no meaningful correlations with GMHI.

**Species-level GMHI stratifies healthy and nonhealthy groups.** We calculated GMHI for each stool metagenome in our meta-dataset of 4347 samples to investigate whether the distributions of GMHI differ between healthy and nonhealthy groups. We found that the gut microbiomes in healthy have significantly higher GMHIs in comparison to gut microbiomes in nonhealthy (Mann–Whitney $U$ test, $P = 5.06 \times 10^{-212}$; Cliff's Delta effect size = 0.56; Fig. 3a). (Of note, Cliff's Delta ($d$) is a non-parametric effect-size measure that quantifies how often one value in one distribution is higher than the values in the second distribution; it is a difference between probabilities, and thus ranges from −1 to +1.) By definition of GMHI, this result reflects the dominant influence of Health-prevalent species over Health-scarce species in the healthy group, and vice versa in the nonhealthy group.

Next, to further identify differences between healthy and nonhealthy groups, we examined multiple measures of ecological characteristics that can be defined on a per-sample basis. For α-diversity based on the Shannon index, we found significantly higher values in healthy than in nonhealthy (Mann–Whitney $U$ test, $P = 8.50 \times 10^{-9}$; Cliff's Delta = 0.10; Fig. 3b). In agreement with our results, previous investigations have also reported higher diversity in the gut microbiomes of healthy controls than in those of disease patients[37–39]. In addition, we found that the minimum number of species required to comprise at least 80% of the sample's relative abundance (henceforth called "80% abundance coverage") was significantly higher in healthy compared to nonhealthy (Mann–Whitney $U$ test, $P = 2.30 \times 10^{-12}$; Cliff's Delta = 0.13; Fig. 3c). This concept, as demonstrated similarly by Kraal et al.[40], has been adopted in previous studies to estimate the membership of core microbiomes[41,42]. Finally, species richness, which is defined as the observed number of different species, was found to be significantly lower in healthy compared to nonhealthy (Mann–Whitney $U$ test, $P = 2.30 \times 10^{-46}$; Cliff's Delta = −0.26; Fig. 3d).

Finally, we investigated for differences in GMHI and in these ecological characteristics between healthy and each of the 12 phenotypes of the nonhealthy group. At the individual phenotype level, the healthy group showed significantly higher GMHI levels in all but 1 (symptomatic atherosclerosis) of the 12 different disease or abnormal bodyweight conditions (Mann–Whitney $U$ test, $P < 0.001$; Fig. 3e). For Shannon diversity and 80% abundance coverage, we found that only 3 (CD, obesity, and type 2 diabetes) of the 12 nonhealthy phenotypes showed statistically significant differences (Fig. 3f, g); both properties were higher in healthy for all three comparisons. For richness, we found that 8 of the 12 nonhealthy phenotypes were significantly different compared to healthy (Fig. 3h): seven of these eight were of higher richness, whereas one (CD) was of lower richness. Taken together, our results suggest that: (i) healthy and nonhealthy gut microbiomes show distinct ecological characteristics; (ii) GMHI embodies a gut microbiome signature of wellness that is generalizable against various nonhealthy phenotypes; and (iii) GMHI can distinguish healthy from nonhealthy individuals more reliably than Shannon diversity, 80% abundance coverage, and richness.

**Group proportions and Shannon diversity with respect to GMHI.** For increasingly higher (more positive) and lower (more

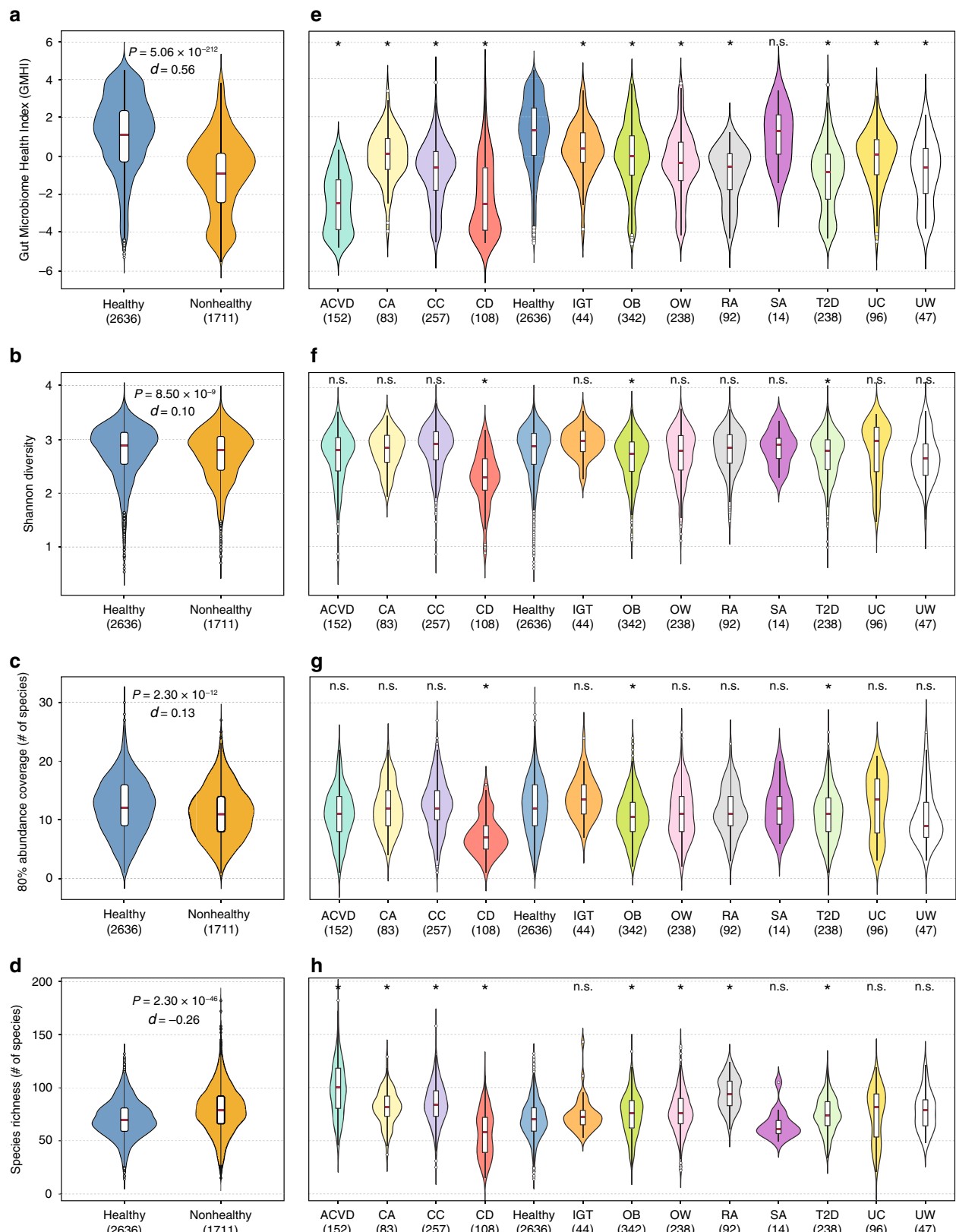

negative) values of GMHI, we observed an increasing proportion of samples from healthy and nonhealthy groups, respectively (Fig. 4a). For example, 98.2% (165 of 168) of metagenome samples with GMHIs >4.0 were from the healthy group; and 81.2% (164 of 202) of metagenome samples with GMHIs <−4.0 were of nonhealthy origin. In addition, the top 10 to 100 healthy

and nonhealthy stool metagenome groups (selected based on their GMHIs) clearly clustered apart from each other in PCoA ordination (Supplementary Fig. 7), in stark contrast to the case when all samples were projected simultaneously (Fig. 1c). These observations confirm that very high (or low) collective abundance of Health-prevalent species relative to that of

**Fig. 3 Comparisons among GMHI and other ecological metrics in stratifying healthy from nonhealthy phenotypes. a–d** Significantly higher distributions of GMHI ($P = 5.06 \times 10^{-212}$), Shannon diversity ($P = 8.50 \times 10^{-9}$), and 80% abundance coverage ($P = 2.30 \times 10^{-12}$) were observed in gut microbiomes of healthy than in those of nonhealthy individuals, whereas higher species richness ($P = 2.30 \times 10^{-46}$) was observed in nonhealthy gut microbiomes. The strongest effect size (Cliff's Delta, $d$) was seen with GMHI. **e–h** The healthy group was found to have a significantly higher distribution of GMHIs than all but one (SA) of the 12 nonhealthy phenotypes. For Shannon diversity and 80% abundance coverage, only three nonhealthy phenotypes (CD, OB, and T2D) were found to have significantly different distributions compared to healthy; both properties were higher in healthy than in CD, OB, and T2D. For species richness, 7 (ACVD, CA, CC, OB, OW, RA, and T2D) of the 12 nonhealthy phenotypes were observed to have significantly higher richness than healthy; in contrast, only CD showed significantly lower richness compared to healthy. All $P$ values shown above the violin plots were found using the two-sided Mann–Whitney $U$ test. *$P < 0.001$ in two-sided Mann–Whitney $U$ test; n.s., not significant. The sample size of each group is shown within parentheses. ACVD atherosclerotic cardiovascular disease, CA colorectal adenoma, CC colorectal cancer, CD Crohn's disease, IGT impaired glucose tolerance, OB obesity, OW overweight, RA rheumatoid arthritis, SA symptomatic atherosclerosis, T2D type 2 diabetes, UC ulcerative colitis, UW underweight. Standard box-and-whisker plots (e.g., center line, median; box limits, upper and lower quartiles; whiskers, 1.5× interquartile range; circles, outliers) are used to depict groups of numerical data.

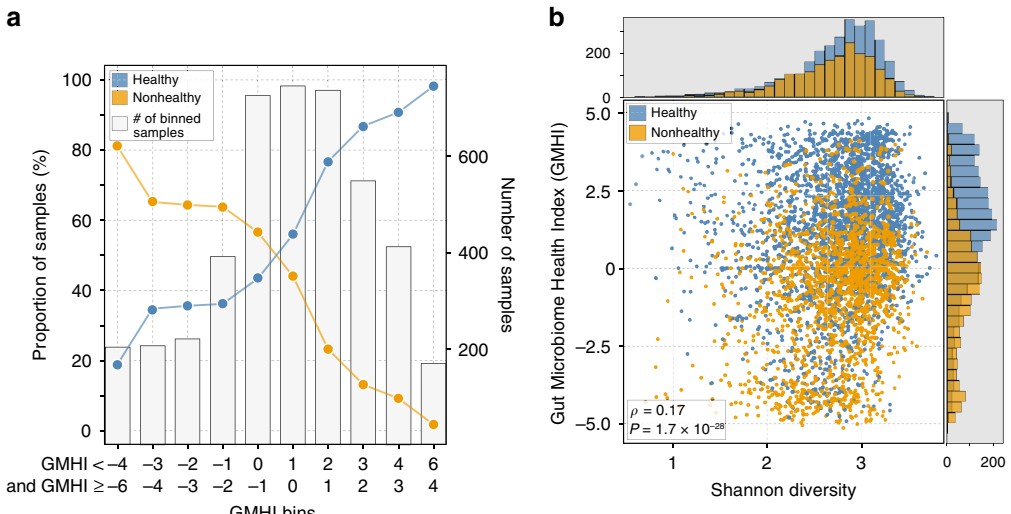

**Fig. 4 Changes in group proportions and in Shannon diversity with respect to GMHI. a** All 4347 metagenomes were binned according to their GMHI values ($x$-axis). Each gray bar indicates the total number of samples in each bin ($y$-axis, right). Points indicate proportions (i.e., percentages) of samples in each bin corresponding to either healthy or nonhealthy individuals ($y$-axis, left). In bins with a positive range of GMHIs, the majority of samples classified as healthy; in contrast, samples in bins with a negative range of GMHIs mostly classified as nonhealthy. This trend was more pronounced towards bins on the far right and left. **b** GMHI stratifies healthy ($n = 2636$) and nonhealthy ($n = 1711$) groups more strongly compared to Shannon diversity. Each point in the scatter-plot corresponds to a metagenome sample (4347 in total). Histograms show the distribution of healthy (blue) and nonhealthy (orange) samples based on the parameter of each axis. In general, GMHI and Shannon diversity demonstrate a weak correlation (Spearman's $\rho = 0.17$, 95% CI: [0.14, 0.19], $P = 1.7 \times 10^{-28}$). The $P$ value ($H_0: \rho = 0$) was determined by using a $t$-distribution with $n - 2$ degrees of freedom, where $n$ is the total number of observations.

Health-scarce species is strongly connected to being healthy (or nonhealthy).

GMHI and Shannon diversity were compared for each sample to examine their overall concordance. As shown in Fig. 4b, GMHI clearly performed much better in stratifying the healthy and nonhealthy groups compared to Shannon diversity. A small yet significant relationship was found between our metric and this conventional measure of gut health (Spearman's $\rho = 0.17$, 95% CI: [0.14, 0.19], $P = 1.66 \times 10^{-28}$). In addition, similar results were seen when GMHI was compared with 80% abundance coverage (Spearman's $\rho = 0.22$, 95% CI: [0.19, 0.25], $P = 8.48 \times 10^{-48}$) and with richness (Spearman's $\rho = -0.27$, 95% CI: [−0.30, −0.24], $P = 4.27 \times 10^{-74}$) (Supplementary Fig. 8).

**Intra-study analyses favor GMHI over other ecology metrics.** We next examined how well GMHI and other features of microbial ecology (i.e., Shannon diversity, 80% abundance coverage, and species richness) could distinguish healthy and nonhealthy phenotypes within individual studies. Specifically, in each of the 12 studies (out of 34 total) wherein at least 10 stool

metagenome samples from both case (i.e., disease or abnormal bodyweight conditions) and control (i.e., healthy) subjects were available, we compared GMHI, Shannon diversity, 80% abundance coverage, and species richness between healthy and nonhealthy phenotype(s). By focusing on datasets from individual studies one by one, this approach not only removes a major source of batch effects, but also provides a good means to investigate the robustness of our previously observed trends (when healthy and nonhealthy samples were compared against each other in aggregate groups) across multiple, smaller studies.

We found that GMHI in healthy was significantly higher than that in any nonhealthy phenotype for 11 out of 28 case–control comparisons (Fig. 5a). For Shannon diversity and 80% abundance coverage, we found significantly higher values in healthy than in nonhealthy phenotypes for two and four case–control comparisons, respectively (Fig. 5b, c). Last, we found species richness in healthy to be significantly lower than that in nonhealthy phenotypes for three case–control comparisons (Fig. 5d). Clearly, the performance of GMHI was not perfect (and likewise for other ecological characteristics), as the expected trend from the prior pooled analysis was not replicable for all case–control

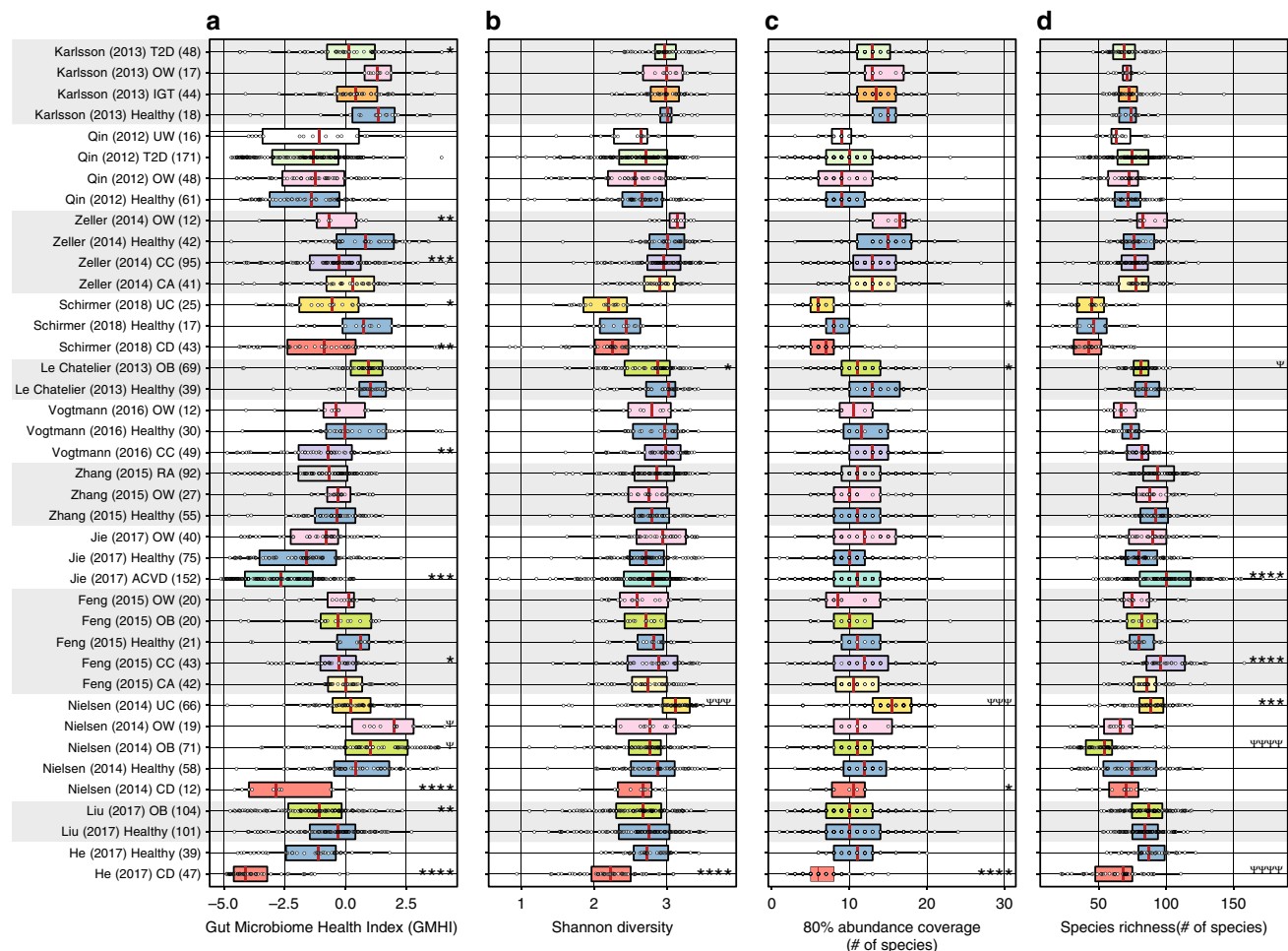

**Fig. 5 GMHI generally outperforms other microbiome ecological characteristics in distinguishing case and control across multiple study-specific comparisons.** In each of the 12 studies wherein at least 10 case (i.e., disease or abnormal bodyweight conditions) and at least 10 control (i.e., healthy) subjects were available, stool metagenomes were analyzed to compare **a** GMHI, **b** Shannon diversity, **c** 80% abundance coverage, and **d** species richness between healthy and nonhealthy phenotype(s). GMHI was found to have a significantly higher distribution in healthy for 11 (out of 28) case–control comparisons across nine different studies; Shannon diversity and 80% abundance coverage were found to have significantly higher distributions in healthy for two and four case–control comparisons (across two and four studies), respectively; and species richness was found to have a significantly lower distributions in healthy for three case–control comparisons across three different studies. Each study's phenotype sample size is shown within parentheses to the right of the phenotype abbreviation. Standard box-and-whisker plots (e.g., center line, median; box limits, upper and lower quartiles; whiskers, 1.5× interquartile range; points, samples) are used to depict groups of numerical data. The same colors in boxplots were used for the same phenotypes. P values (two-sided Mann–Whitney U test) for each study-specific comparison between healthy and nonhealthy phenotypes are shown adjacent to the boxplots accordingly: * and Ψ indicates significantly different distributions consistent with, and opposite to, respectively, the previously observed results when healthy and nonhealthy groups were compared in aggregate. * or $^{\Psi}$0.01 ≤ P value < 0.05; ** or $^{\Psi\Psi}$0.001 ≤ P value < 0.01; *** or $^{\Psi\Psi\Psi}$0.0001 ≤ P value < 0.001; **** or $^{\Psi\Psi\Psi\Psi}$P value < 0.0001. ACVD atherosclerotic cardiovascular disease, CA colorectal adenoma, CC colorectal cancer, CD Crohn's disease, IGT impaired glucose tolerance, OB obesity, OW overweight, RA rheumatoid arthritis, SA symptomatic atherosclerosis, T2D type 2 diabetes, UC ulcerative colitis, UW underweight.

comparisons within every study; overall though, GMHI strongly outperformed other microbiome ecological characteristics in distinguishing case and control.

Analogous to the analysis above (wherein healthy was compared to each separate nonhealthy phenotype within individual studies), we compared healthy against a general nonhealthy phenotype, in which all disease samples were lumped together, when applicable. Importantly, comparisons were still made within individual studies. We found that there were statistically significant differences in GMHI between cases and controls (Mann–Whitney U test, P < 0.05; Supplementary Table 4) in 6 of the 12 studies. In contrast, we found statistically significant differences in Shannon diversity, 80% abundance coverage, and richness between cases and controls in two, three, and three (of 12) studies, respectively.

**Validation of GMHI reproducibility on independent cohorts.** Evaluation of any biomarker or molecular signature on independent patient samples is the gold standard for assessing its robustness[15]. To confirm the reproducibility of our prediction results in stratifying healthy and nonhealthy phenotypes (Fig. 3), we leveraged GMHI to predict the health status of 679 individuals whose stool metagenome samples were not part of the original formulation of GMHI. For this, we used gut microbiome data from an additional eight published studies (Supplementary Table 5), which include stool metagenomes from healthy subjects and patients with ankylosing spondylitis (AS), colorectal adenoma, colorectal cancer, Crohn's disease (CD), liver cirrhosis (LC), and nonalcoholic fatty liver disease (NAFLD). In addition, we utilized our extensive biobank of stool collections to gather

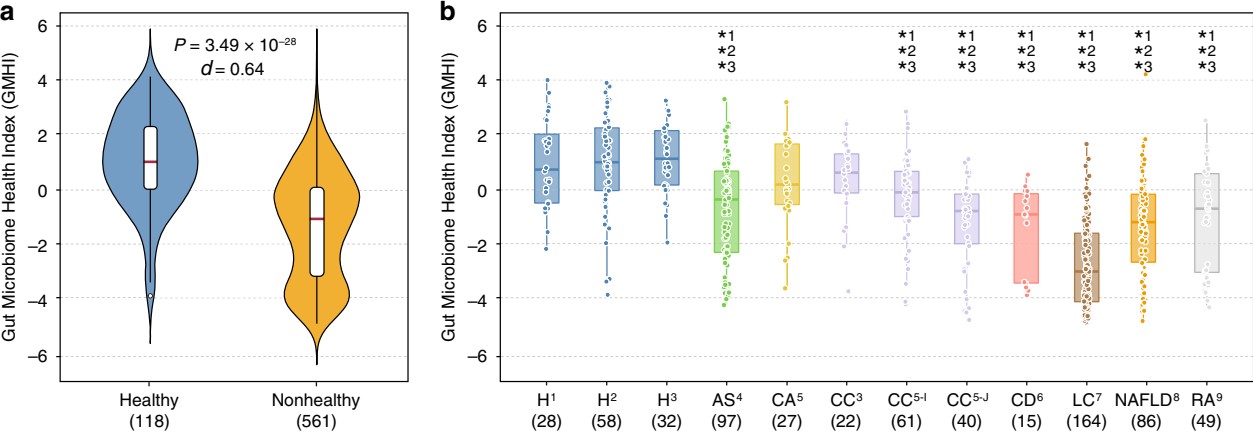

**Fig. 6 GMHI demonstrates strong reproducibility on an independent validation cohort.** The validation cohort (679 stool metagenome samples) consisted of 12 total sub-cohorts ranging across eight healthy and nonhealthy phenotypes from nine different studies. **a** GMHIs from stool metagenomes of the healthy group were significantly higher than those of the nonhealthy group (two-sided Mann–Whitney $U$ test, $P = 3.49 \times 10^{-28}$). $d$ Cliff's Delta. **b** All three healthy sub-cohorts (H[1], H[2], and H[3]) were found to have significantly higher distributions of GMHI than seven (of nine) nonhealthy sub-cohorts (AS[4], CC[5-I], CC[5-J], CD[6], LC[7], NAFLD[8], and RA[9]). No significant differences were found among H[1], H[2], and H[3]. The number in superscript adjacent to phenotype abbreviations corresponds to a particular study used in validation (see Supplementary Table 5 for study information). Standard box-and-whisker plots (e.g., center line, median; box limits, upper and lower quartiles; whiskers, 1.5× interquartile range; points, samples) are used to depict groups of numerical data. *Significantly higher distribution in healthy sub-cohort (two-sided Mann–Whitney $U$ test, $P < 0.01$). The number adjacent to * indicates the healthy sub-cohort (H[1], H[2], or H[3]) to which the respective sub-cohort was compared. The sample size of each group or cohort is shown within parentheses. AS ankylosing spondylitis, CA colorectal adenoma, CC colorectal cancer, CD Crohn's disease, H healthy, LC liver cirrhosis, NAFLD, nonalcoholic fatty liver disease, RA rheumatoid arthritis.

our own set of samples from patients with rheumatoid arthritis (RA) ("Methods"; see Supplementary Data 4 for subject meta-data relating to both clinical and nonclinical factors). All metagenome samples in this validation dataset were pooled into one of two groups (i.e., healthy or nonhealthy), as demonstrated above.

In agreement with our results on the discovery cohort (training data), GMHIs from stool metagenomes of the healthy validation group ($n = 118$) were significantly higher than those of the nonhealthy validation group ($n = 561$) (Mann–Whitney $U$ test, $P = 3.49 \times 10^{-28}$; Cliff's Delta $= 0.64$; Fig. 6a). In addition, the balanced accuracy resulted in 73.7%, as the classification accuracy for the healthy and nonhealthy validation group was 77.1% (91 of 118) and 70.2% (394 of 561), respectively. Notably, these results were better than the performances on the discovery cohort, wherein balanced accuracy was 69.7%, and accuracy on the healthy and nonhealthy group was 75.6% (1993 of 2636) and 63.8% (1092 of 1711), respectively.

Of note, we also compared the classification accuracy of GMHI to those of classifiers based upon the Health-prevalent species and Shannon diversity (see Supplementary Methods), and to that of a more intricate classification algorithm (Random Forests). In regards to balanced accuracies on the training data, the classifiers based upon Health-prevalent species ($\chi = 66.3\%$) and Shannon diversity ($\chi = 53.6\%$) performed comparable to, or much worse than, GMHI ($\chi = 69.7\%$); furthermore, balanced accuracy on the independent validation dataset for Health-prevalent species and Shannon diversity resulted in 59.3 and 47.0%, respectively (Supplementary Tables 6 and 7, respectively). On the other hand, the Random Forests classifier ("Methods") achieved a remarkable accuracy on the training data ($\chi = 98.5\%$). However, building complex decision rules entails the risk of over-fitting. Surely enough, this nearly perfect accuracy was mostly in part a result of outstanding over-fitting, evidenced by the poor accuracy of 52.3% (balanced accuracy) on the 679 samples of the validation cohort. In Supplementary Table 8, we provide a summary of all accuracies for classifying healthy vs. nonhealthy by the various classifiers reported in this study.

To investigate GMHI performances on the validation cohort more closely, we examined the 12 total sub-cohorts (defined per unique phenotype per individual study) ranging across eight healthy and nonhealthy phenotypes from eight additional published studies and one newly sequenced batch. As shown in Fig. 6b, all three healthy sub-cohorts were found to have significantly higher distributions of GMHI than seven (of nine) nonhealthy phenotype sub-cohorts (Mann–Whitney $U$ test, $P < 0.01$; see Supplementary Table 9 for Cliff's Deltas). The classification accuracies for these three healthy sub-cohorts were 87.5% (28 of 32), 74.1% (43 of 58), and 71.4% (20 of 28); alternatively, the classification accuracies for the nonhealthy phenotype sub-cohorts were the following: 94.5% (155 of 164) for LC; 75.6% (65 of 86) for NAFLD; 73.3% (11 of 15) for CD; 67.3% (33 of 49) for RA; 55.7% (54 of 97) for AS; 37.0% (10 of 27) for CA; and 77.5% (31 of 40), 47.5% (29 of 61), and 27.3% (6 of 22) for three different cohorts of CC. Strikingly, GMHI performed well (>75.0%) in predicting adverse health for LC and NAFLD, although stool metagenomes from patients with liver disease were not part of the original discovery cohort. This finding suggests that GMHI could be applied beyond the original 12 phenotypes (of the nonhealthy group) used during the index training process. Overall, the strong reproducibility of GMHI implies that the highly diverse and complex features of gut microbiome dysbiosis implicated in pathogenesis were reasonably well captured during the dataset integration and original formulation of GMHI. Thereby, our results support previous findings by Duvallet et al.[26] in regards to the presence of a generalized disease-associated gut microbial signature, which was observed to be shared across multiple studies and pathologies. Finally, from similar analyses for Shannon diversity, 80% abundance coverage, and species richness on the validation cohort, we were able to conclude that GMHI is the most accurate, robust, and clinically meaningful classifier compared to these other ecological characteristics (Supplementary Note 2 and Supplementary Fig. 9).

## Discussion

In this study, we present the GMHI, a simple and biologically interpretable metric to quantify the likelihood of disease presence from a gut microbiome sample. At first, we envisioned that the most intuitive way to determine how closely one's microbiome resembles that of a healthy (or nonhealthy) population is to compare the collective abundances of Health-prevalent and of Health-scarce species. By pooling massive amounts of publicly available data (4347 publicly available, shotgun metagenomic data of gut microbiomes from 34 published studies), we identified a small consortium of 50 microbial species associated with human health to serve as features for our classification model: 7 and 43 species were prevalent and scarce, respectively, in the healthy group compared to the nonhealthy group. In regards to classification accuracy, GMHI distinguished healthy from nonhealthy (as well as from individual diseases) far better than methods adopted from ecological principles (e.g., α-diversity indices), thereby paving a path forward to evaluate human (gut microbiome) health through stool metagenomic profiling. Notably, this framework can be applied to other body niches, for example, quantifying health in skin or oral microbiomes. When demonstrating the potential of GMHI on independent validation datasets, we obtained strong prediction results for healthy individuals, and for cohorts with autoimmunity and liver disease. The strong reproducibility on validation datasets suggests that sufficient dataset integration across a large population could lead to robust predictors of health. This may be due, in part, to the signature encompassing more of the heterogeneity across various sources and conditions, while amplifying signal (against noise) from the repeated phenotype characteristics.

Despite the strong reproducibility in classification accuracy demonstrated on the validation datasets, more is left to be desired in regards to achieving higher accuracies. We conjecture that misclassifications were partly because of the complex[43–45], stochastic[46,47], and highly personalized[18,48] nature of gut microbiome ecologies; all of which complicate the identification of reliable signatures of health. In addition, sample collection and processing procedures, laboratory personnel, study run-dates, measurement instruments, and so forth are tremendously challenging to control in population-scale investigations. In the long term, in order to find even more accurate gut microbiome-based markers, we envision integrating larger data repositories to take into consideration a higher number of samples and sources of heterogeneity.

Several limitations of our study should be noted when interpreting our results. First, as the stool metagenomes were collected from over 40 published studies, we cannot entirely exclude experimental and technical inter-study batch effects. Our efforts to curtail batch effects include: (i) consensus preprocessing, that is, downloading all raw shotgun metagenomes and reprocessing each sample uniformly using identical bioinformatics methods; (ii) using frequencies of a signal (i.e., prevalence of "present" microbes) to identify significant associations, rather than comparing or averaging effect sizes between populations, or performing data transformations that may lead to spurious conclusions[49]; and (iii) validating the reproducibility of GMHI on independent datasets. Second, given our selection criteria ("Methods"), our study does not include all publicly available gut microbiome studies and samples. Certainly, more studies and samples can be taken into consideration under more relaxed criteria. Third, in an effort to be as precise as possible in describing taxonomic features of the human gut, our metagenomic analyses were performed using species-level abundances; however, microbial *strains* are clearly the most clinically informative and actionable unit[50,51]. Moreover, different strains within the same species can have significantly different associations with

disease[52–55], which could not be considered in our study. Nevertheless, our shotgun metagenomic approach is a significant advancement over 16s rRNA gene amplicon sequencing, which are known to be mostly limited to genus-level investigations[56,57]. Fourth, in the nonhealthy group, we pooled samples from only 12 phenotypes. Certainly, many more pathological states have been linked to the gut microbiome, including neurodegenerative and psychiatric disorders[58–60]. Thus, future studies will need to continuously update and expand our findings by encompassing a much broader range of conditions as new data become available. Fifth, we did not consider metagenomic functional profiles to define gut ecosystem health as demonstrated extensively by others[27,61–63], as this too was outside the scope of our study. For microbiomes of any phenotype of interest, we posit that analyzing both taxonomic composition and functional potential are both important and complementary directions. Last, while we definitely tried to be as inclusive as possible of various geographies, ethnicities/races, and cultures, we do acknowledge that complete elimination of biases is practically impossible. Certainly, for future works, we plan to iteratively expand our application to encompass broader ranges of subjects, including those from underdeveloped countries and minority ethnicities/races, to better understand microbiome diversity and foster inclusion in microbiome research[64].

## Methods

**Multi-study integration of human stool metagenomes**. We performed exhaustive keyword searches (e.g., "gut microbiome," "metagenome," "whole-genome shotgun (WGS)") in PubMed and Google Scholar for published studies with publicly available WGS metagenome data of human stool (gut microbiome) and corresponding subject meta-data (as of March 2018). In studies wherein multiple samples were taken per individual across different time-points, we included only the first or baseline sample in the original study. We excluded studies pertaining to diet or medication interventions, or those with fewer than 10 samples. Samples from subjects who were <10 years of age were also excluded from our analysis. Last, samples that were collected from disease controls, but were not reported as healthy nor had any mentioning of diagnosed disease in the original study, were excluded from our analysis. Raw sequence files (.fastq) were downloaded from the NCBI Sequence Read Archive and European Nucleotide Archive databases (Supplementary Data 1) for the study analysis.

**Reclassification of healthy samples based on reported BMI**. Healthy individuals, regardless of whether they had been determined as healthy in the original studies, were considered to be part of the nonhealthy group if their reported BMI fell within the range of underweight (BMI < 18.5), overweight (BMI ≥ 25 and <30), or obese (BMI ≥ 30). Stool metagenome samples from such individuals were reclassified as underweight, overweight, or obese in our analysis.

**Quality control of sequenced reads**. Sequence reads were processed with the KneadData v0.5.1 quality-control pipeline (http://huttenhower.sph.harvard.edu/kneaddata), which uses Trimmomatic v0.36 and Bowtie2 v0.1 for removal of low-quality read bases and human reads, respectively. Trimmomatic v0.36 was run with parameters SLIDINGWINDOW:4:30, and Phred quality scores were thresholded at "<30." Illumina adapter sequences were removed, and trimmed nonhuman reads shorter than 60 bp in nucleotide length were discarded. Potential human contamination was filtered by removing reads that aligned to the human genome (reference genome hg19). Furthermore, stool metagenome samples of low read count after quality filtration (<1M reads) were excluded from our analysis.

**Species-level taxonomic profiling**. Taxonomic profiling was done using the MetaPhlAn2 v2.7.0 phylogenetic clade identification pipeline[32] using default parameters. Briefly, MetaPhlAn2 classifies metagenomic reads to taxonomies based on a database (mpa_v20_m200) of clade-specific marker genes derived from ~17,000 microbial genomes (corresponding to ~13,500 bacterial and archaeal, ~3500 viral, and ~110 eukaryotic species).

**Sample filtering based on taxonomic profiles**. After taxonomic profiling, the following stool metagenome samples were discarded from our analysis: (i) samples composed of >5% unclassified taxonomies (100 samples); and (ii) phenotypic outliers according to a dissimilarity measure. More specifically, Bray–Curtis distances were calculated between each sample of a particular phenotype and a hypothetical sample in which the species' abundances were taken from the medians across those samples. A sample was considered as an outlier, and thereby removed from further analysis,

when its dissimilarity exceeded the upper and inner fence (i.e., >1.5 times outside the interquartile range above the upper quartile and below the lower quartile) among all dissimilarities. This process removed 67 metagenome samples.

**Species removal based on taxonomic profiles**. As taxonomic assignment based on clade-specific marker genes may be problematic for viruses[65,66], we excluded the 298 of viral origin from our analysis. Species that were labeled as either unclassified or unknown (118 species), or those of low prevalence (i.e., observed in <1% of the samples included in our meta-dataset; 472 species), were also excluded. Eventually, 313 microbial species across 4347 stool metagenome samples remained in our study for further analysis (Supplementary Data 2).

**PCoA based on taxonomic profiles**. The R packages "ade4" v1.7-15 and "vegan" v2.5.6 were used to perform PCoA ordination with Bray–Curtis dissimilarity as the distance measure on the stool metagenome samples, which were comprised of arcsine square root-transformed relative abundances of the aforementioned 313 microbial species identified by MetaPhlAn2. 999 permutations ("adonis2" function in the R "vegan" package v2.5.6) were performed, while random permutations were constrained within studies by using the "strata" option.

**Calculation of microbiome ecological characteristics**. The R package "vegan" v2.5.6 was used to calculate Shannon diversity (Shannon index) and species richness based on the species abundance profiles for each sample of our meta-dataset. To identify the 80% abundance coverage for a stool metagenome sample, the smallest number of microbial species that comprise at least 80% of the total relative abundance was identified.

*Identifying microbial species more frequently observed in healthy than in nonhealthy (and vice versa):*

(a) Let $p_{H,m}$ and $p_{N,m}$ be the prevalence of microbial species $m$, that is, proportion of samples in a given group where $m$ is "present" (or relative abundance $\geq 1.0 \times 10^{-5}$), in the healthy group $H$ and nonhealthy group $N$, respectively. Remark: The relative abundances for all detectable species in a microbiome (metagenome) sample sums to 1.

(b) For $m$, the prevalence fold change $f_m^{H,N}$ and prevalence difference $d_m^{H,N}$, defined as $\frac{p_{H,m}}{p_{N,m}}$ and $p_{H,m} - p_{N,m}$, respectively, is identified.

(c) Let $\theta_f$ and $\theta_d$ be defined as the minimum thresholds for $f_m^{H,N}$ and $d_m^{H,N}$, respectively. For all detectable species in a microbiome sample, those that satisfy $f_m^{H,N} \geq \theta_f$ and $d_m^{H,N} \geq \theta_d$ are identified. These species are included as an element of "Health-prevalent" species $M_H$, or the set of species more frequently observed in group $H$ than in group $N$.

(d) To identify "Health-scarce" species $M_N$, or the set of species more frequently observed in group $N$ than in group $H$, steps (b) through (c) are repeated with the following considerations:

 i. For $m$, let $f_m^{N,H}$ and $d_m^{N,H}$ be defined as $\frac{p_{N,m}}{p_{H,m}}$ and $p_{N,m} - p_{H,m}$, respectively.

 ii. The same thresholds $\theta_f$ and $\theta_d$ are used to identify $M_N$. In this regard, the species that are eventually chosen to compose $M_H$ and $M_N$ are both dependent on $\theta_f$ and $\theta_d$.

 iii. Finally, all detectable species that satisfy $f_m^{N,H} \geq \theta_f$ and $d_m^{N,H} \geq \theta_d$ are included in $M_N$.

*Identifying $\psi_{M_H}$ (or $\psi_{M_N}$), that is, the "collective abundance" of species in $M_H$ (or $M_N$) in a microbiome sample:*

(a) $\psi_{M_H}$ is defined as the "collective abundance" of $M_H$ species in a microbiome sample. The calculation of $\psi_{M_H}$ takes into consideration the following:

 i. Species richness, that is, the numeric count of "present" species of MH.

 ii. (geometric) Mean of their relative abundances.

(b) *Basic assumptions*:

 i. $\psi_{M_H}$ is positively correlated with $R_{M_H}$, or the richness of $M_H$ species. Thus, (their correlation) $\rho(\psi_{M_H}, R_{M_H}) > 0$. Remark: Due to the possible large discrepancy between the cardinality (set size) of $M_H$ and that of $M_N$, the *proportion* of "present" $M_H$ species is used. As such, $R_{M_H}$ is replaced with $\frac{R_{M_H}}{|M_H|}$. Thus, $\rho(\psi_{M_H}, \frac{R_{M_H}}{|M_H|}) > 0$.

 ii. $\psi_{M_H}$ is positively correlated with $\langle M_H \rangle$, or the mean abundance of species in $M_H$. Thus, $\rho(\psi_{M_H}, \langle M_H \rangle) > 0$. Remark: As it is common in microbiome data to have discrepancies between species' relative abundances to span several orders of magnitude, the geometric mean, rather than the arithmetic mean, is more appropriate to represent the mean relative abundance of $M_H$ species. More specifically, the Shannon's diversity index, which is a weighted geometric mean (by definition) and commonly applied in ecological contexts, is used. Thus, for simplicity, $\langle M_H \rangle \approx \sum_{j \in I_{M_H}} |n_j ln(n_j)|$ is assumed, where $I_{M_H}$ is the index set of $M_H$, and $n_j$ is the relative abundance of species $j$ in $I_{M_H}$.

(c) *Overview*:

 i. Given the assumptions in (b), as well as the nonnegativity of $\frac{R_{M_H}}{|M_H|}$ and $\sum_{j \in I_{M_H}} |n_j ln(n_j)|$, $\psi_{M_H}$ is simply formulated as a *product* of the aforementioned two traits. Thus, let $\psi_{M_H} = \frac{R_{M_H}}{|M_H|} \sum_{j \in I_{M_H}} |n_j ln(n_j)|$.

 ii. Analogously, let $\psi_{M_N} = \frac{R_{M_N}}{|M_N|} \sum_{j \in I_{M_N}} |n_j ln(n_j)|$

*Identifying $h_{i,M_H,M_N}$, that is, ratio of $\psi_{M_H}$ to $\psi_{M_N}$, in sample $i$:*

(a) Formally, the log ratio of $\psi_{M_H}$ to $\psi_{M_N}$ in sample $i$ can be written as

$$h_{i,M_H,M_N} = \log_{10} \left( \frac{\frac{R_{M_H}}{|M_H|} \sum_{j \in I_{M_H}} \left| n_j ln(n_j) \right|}{\frac{R_{M_N}}{|M_N|} \sum_{j \in I_{M_N}} \left| n_j ln(n_j) \right|} \right). \tag{4}$$

(b) By definition, $|M_H|$ and $|M_N|$ is the highest richness that can be obtained by $M_H$ and $M_N$ species, respectively, in a particular microbiome sample. However, the possibility that these maximum values are rarely obtained cannot be ruled out; if so, then consequently, having a larger set size of $M_H$ (or $M_N$) can generally result in a lower distribution of $\frac{R_{M_H}}{|M_H|}$ (or $\frac{R_{M_N}}{|M_N|}$), potentially leading to biases in $h_{i,M_H,M_N}$ when $|M_H| \gg |M_N|$ or $|M_H| \ll |M_N|$. Therefore, the upper limits that can be eventually used in replacement of $|M_H|$ and $|M_N|$ in Eq. (4) should reflect more of what is actually observed in real microbiome data, for example, samples ranked according to the magnitude observed between $R_{M_H}$ and $R_{M_N}$. In this regard, the following procedure to find alternative measures for $|M_H|$ and $|M_N|$ is used:

 i. Identify $R_{M_H}$ and $R_{M_N}$ for all microbiome samples in groups $H$ and $N$.

 ii. Rank order all samples consecutively by two criteria: first, by all values of $R_{M_N}$ in ascending order (from lowest to highest); and then, by all values of $R_{M_H}$ in descending order (from highest to lowest). This sorting strategy prioritizes having the highest possible $R_{M_H}$ (but with the constraint of having $R_{M_N} \approx 0$) for the most top-ranked samples; and having the highest possible $R_{M_N}$ (but with the constraint of having $R_{M_H} \approx 0$) for the most bottom-ranked samples.

 iii. Let $k_H$ be the closest integer to 1% of the number of samples in group $H$. As $H$ is composed of 2636 samples, let $k_H$ be 26. Analogously, as $N$ is composed of 1711 samples, let $k_N$ be 17.

 iv. Denote $|M_H|'$ as the median $R_{M_H}$ from the top $k_H$ samples, and denote $|M_N|'$ as the median $R_{M_N}$ from the bottom $k_N$ samples.

 v. Replace $|M_H|$ and $|M_N|$ in Eq. (4) with $|M_H|'$ and $|M_N|'$, respectively.

(c) In summary, the ratio of $\psi_{M_H}$ to $\psi_{M_N}$ in gut microbiome sample $i$ can be written as

$$h_{i,M_H,M_N} = \log_{10} \left( \frac{\frac{R_{M_H}}{|M_H|'} \sum_{j \in I_{M_H}} \left| n_j ln(n_j) \right|}{\frac{R_{M_N}}{|M_N|'} \sum_{j \in I_{M_N}} \left| n_j ln(n_j) \right|} \right). \tag{5}$$

*Calculating the balanced accuracy of $h_{M_H,M_N}$:*

(a) The relative abundances of species in $M_H$ and those in $M_N$ for microbiome sample $i$ can be provided as input features for: (i) $\psi_{M_H}$ and $\psi_{M_N}$, respectively; and (ii) $h_{i,M_H,M_N}$, which in turn can classify sample $i$ as healthy (i.e., $h_{i,M_H,M_N} > 0$), nonhealthy (i.e., $h_{i,M_H,M_N} < 0$), or neither (i.e., $h_{i,M_H,M_N} = 0$).

(b) The classification accuracy or predictive performance of $h_{i,M_H,M_N}$ is found by testing it on all samples in groups $H$ and $N$, and then by finding the balanced accuracy $\chi_{M_H,M_N}$ defined in Eq. (3).

*Determining optimal sets $M_H^\gamma$ and $M_N^\gamma$:*

(a) The final, optimal sets of $M_H^\gamma$ and $M_N^\gamma$ are found by first considering a range of thresholds $\theta_f$ and $\theta_d$. Every pair of $\theta_f$ and $\theta_d$ gives different sets of $M_H$ and $M_N$, and in turn, different values of balanced accuracy $\chi_{M_H,M_N}$ (see Supplementary Table 1).

(b) The final, optimal sets of $M_H^\gamma$ and $M_N^\gamma$ (and their corresponding $\theta_f^\gamma$ and $\theta_d^\gamma$) are determined as those that result in the highest balanced accuracy $\chi_{M_H,M_N}^{max}$.

**MetaCyc pathway functional profiling of stool metagenomes**. MetaCyc pathway-level relative abundances in each stool metagenome were quantified by the HUMAnN v2.0 pipeline[63] using default parameters. The EC-filtered UniRef90 gene family database was integrated within the pipeline. Pathways that were unmapped (or unintegrated) were excluded from the analyses.

**Designing a classifier based upon Random Forests**. A classifier based upon a Random Forests algorithm was designed and curated in Python v3.6.4., while model implementation was performed in the "scikit-learn" Python package v0.23.1.

**Stool sample collection and processing**. All stool samples from patients with RA were obtained following written informed consent. The collection of biospecimens was approved by the Mayo Clinic Institutional Review Board (#14-000616). Stool samples from patients with RA were stored in their house-hold freezer ($-20\,°C$) prior to shipment on dry ice to the Medical Genome Facility Research Core at Mayo Clinic (Rochester, MN). Once received, the samples were stored at $-80\,°C$ until DNA extraction. DNA extraction from stool samples was conducted as follows: aliquots were created from parent stool samples using a tissue punch, and the resulting child samples were then mixed with reagents from the Qiagen Power Fecal Kit. This included adding 60 μL of reagent C1 and the contents of a power bead tube (garnet beads and power bead solution). These were then vigorously vortexed to bring the sample punch into solution and centrifuged at $18,000 × G$ for 15 min. From there, the samples were added into a mixture of magnetic beads using a JANUS liquid handler. The samples were then run through a Chemagic MSM1 according to the manufacturer's protocol. After DNA extraction, paired-end libraries were prepared using 500 ng genomic DNA according to the manufacturer's instructions for the NEB Next Ultra Library Prep Kit (New England BioLabs). The concentration and size distribution of the completed libraries was determined using an Agilent Bioanalyzer DNA 1000 chip (Santa Clara, CA) and Qubit fluorometry (Invitrogen, Carlsbad, CA). Libraries were sequenced at 23–70 million reads per sample following Illumina's standard protocol using the Illumina cBot and HiSeq 3000/4000 PE Cluster Kit. The flow cells were sequenced as $150 × 2$ paired-end reads on an Illumina HiSeq 4000 using the HiSeq 3000/4000 Sequencing Kit and HiSeq Control Software HD 3.4.0.38. Base-calling was performed using Illumina's RTA version 2.7.7.

**Reporting summary**. Further information on research design is available in the Nature Research Reporting Summary linked to this article.

## Data availability

Raw sequencing data accession IDs of all publicly available stool metagenome samples (and their corresponding studies) used in all analyses of this study are available in Supplementary Data 1 and Supplementary Data 4. Sequences for the dataset containing rheumatoid arthritis stool metagenomes used for GMHI validation have been deposited at NCBI's Sequence Read Archive (SRA) data repository (BioProject number PRJNA598446), and can be downloaded without any restrictions. The deposited sequences include .fastq files for 49 patients with rheumatoid arthritis. Measurements were taken from distinct samples. Human reads were identified and removed prior to data upload.

## Code availability

R scripts demonstrating how to reproduce all findings shown in the main figures, as well as how to calculate GMHI for a given stool metagenome sample, are available at https://github.com/jaeyunsung/GMHI_2020.

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

## Acknowledgements

This work was supported, in part, by the Mayo Clinic Center for Individualized Medicine (to V.K.G., M.K., U.B., K.Y.C., N.C., and J.S.), and Mark E. and Mary A. Davis to Mayo Clinic Center for Individualized Medicine (J.S.).

## Author contributions

V.K.G. and J.S. conceived the problem. V.K.G., M.K., and J.S. designed all analytical methodologies. V.K.G. and K.Y.C. performed the computational experiments. All authors analyzed the data. V.K.G. and J.S. wrote the manuscript, with contributions from other authors. J.M.D. is the principal investigator of the Mayo Clinic Rheumatology Biobank, from which stool samples were collected from patients with rheumatoid arthritis. All authors reviewed and approved the final manuscript.

## Competing interests

V.K.G. and J.S. disclose that a patent application was filed relating to the materials in this manuscript. All other authors declare no competing interests.
