## [Peer Review File · Nature Communications]

Reviewers' Comments:

Reviewer #1:

Remarks to the Author:

Based on a large cohort of >4K samples, the authors derive a novel species prevalence metric that separates healthy metagenome samples from non-healthy ones. The metric was designed to rely on species prevalence rather than abundances which makes it robust against batch effects and some common confounding. Validation with an additional test set of about 600 individuals demonstrates that the metric is robust even when applied to diseases not present in the training data set.

The question of what constitutes a healthy microbiome is one of the unsolved key problems in the field right now and the manuscript shows a novel and creative way to get closer to an answer. The major strength of the novel microbiome health index presented here is that the authors took a lot of care to circumvent some of the common problems in designing a score based on samples from heterogeneous studies and protocols. In particular, I found the strategy to use a prevalence-based score rather than an abundance score quite appealing. I do have some doubts about using species abundance rather than other taxonomic ranks or gene abundances. Also it should be mentioned that the score classifies absence of diagnosed disease rather than actual health status and that there may be geographical/cultural biases since the training set is mostly (but not exclusively) composed of samples from the US, Europe and Asia. Nevertheless the manuscript takes a good first step at defining health through the microbiome and will be of interest to a wide readership and the medical community. The manuscript is well written and provides a clear path from motivation to results.

Suggested major changes

1) The paper develops a novel microbiome health index but I feel like the "health" part would require some additional validation. The index itself is designed to separate disease samples from their controls and samples with abnormal BMI. However, it is unclear whether those controls are indeed healthy individuals as the only thing one knows for sure is that they did not show symptoms severe enough to be classified as diseased in the particular study. In order to validate that the microbiome index indeed quantifies health I would like to see at least a validation with a data set that contains a detailed clinical characterization of the cohort to check whether individuals with the highest health index do indeed show better clinical labs, higher indices of self-reported well-being or less ambulant treatments than individuals with a lower index. The authors own RA cohort may be helpful here. Alternatively, the authors could train a genus-level score (as suggested below) and validate the health status on the American Gut data set (<https://doi.org/10.1128/mSystems.00031-18>) which includes a wide array of health measures and self-evaluations.

2) I was surprised the authors chose species as the summary rank for their health index. If one wanted to design a score specifically for metagenome samples better performance would probably be achieved by using bacterial gene abundances directly as this is much closer to a functional analysis. The authors claim themselves that one would expect less functional heterogeneity than taxonomic heterogeneity and show that the identified species share similar genes. I do think that basing an index on taxon abundances is still valuable but would probably be more useful on the genus-level as this opens up validation with many more 16S amplicon sequencing data sets (where reliable taxonomic classification is usually only achieved down to the genus-level). At the very least I would like to see a comparison of accuracies on various taxonomic ranks to show that the species level does indeed show better performance than other ranks.

3) The index will naturally depend on the training set and may not extend well to cohorts different from the training data. Even though the authors tried to generate a diverse data set it still included very few samples that are not from developed countries. There are only a handful of samples not

from the US, Europe or Asia, and it looks like the health index does not perform well for data sets from different cohorts. For instance see the study from Obregon-Tito et. al. (Peru) in Figure 2 but also several studies from China where the healthy cohorts are primarily classified as "less healthy" (lower index). This limitation should be mentioned in the text as its misinterpretation in a potential clinical setting could have negative consequences for underrepresented groups.

Suggested minor changes

4) Some studies are missing in Figure 2. For instance the study from Sankaranarayanan et. al. The figure caption should explain why studies were omitted here.

5) The gene copy number analysis presented in Figure 3 seemed a bit inaccurate since it only used data from reference strains which are probably not representative of the actual strains in the samples. Metagenome data allows to quantify gene copy number directly from the sequencing data. This can be achieved by metagenomic assembly and de novo gene prediction. Metagenomic binning should still allow to connect gene abundances back to the Metaphlan marker genes. This would be much more representative of shared functions than some arbitrary reference strains.

6) I feel like the formula for the health index (h) should be motivated and explained a bit better. The description in the supplement does describe *how* the index is calculated but I would have liked to read *why* the authors chose that particular formula. For instance, the weighting with the Shannon diversity makes sense to me as it prefers samples with a high diversity of health-related microbes, but the description is a bit complicated and may not be accessible to a wide readership - especially one that is not familiar with alpha diversity or the Shannon index.

7) The authors claim that their measure is robust against batch effects but this is not demonstrated. The authors should show that the distribution of the index for healthy individuals does not vary between studies and does not depend on library size.

8) I would appreciate if some more of the source code is provided to reproduce the study. For now the authors include a script that calculates the health index but do not provide materials that would allow running the Metaphlan analysis on the full data set used in the study. The provided SRA accession for the authors' data does not seem to exist (may still be under embargo).

Reviewer #2:

Remarks to the Author:

In their manuscript, Gupta and colleagues posit a functional form for a microbiome index, parameterize the index using a large dataset of cross-sectional microbiome studies, then investigate the metagenomic meaning of the index and compare its performance in classifying cases vs. controls in the dataset of microbiome studies.

I have serious concerns about the validity and utility of this approach. I was confused by the rationale for the work and the organization of the manuscript. My approach will be to re-construct the authors' argument and key claims, noting what I think are the key issues in each case.

Rationale

As I understood, this work is motivated by three rationales.

First, there are barriers to understanding the role of the microbiome in health and disease when using individual studies (line 76 and following). Therefore, meta-analyses comparing health and disease across studies, populations, and diseases is important to microbiome science.

Table 2 and Figure 3 seem to be the fulfillment of this meta-analysis rationale, but the results are presented through the lens of the creation of the novel index, which makes it difficult to interpret the results as a meta-analysis per se.

If a meta-analysis were the focus of the paper, I would have expected the authors to carefully survey previous meta-analyses and make a clear advancement over them in terms of dataset, methodology, or interpretation. It was not clear to me how the novel index was a methodological advance in the identification of health- and disease-associated species.

Second, a common analysis in microbiome studies comparing cases and controls is bacterial alpha diversity, with the expectation that health individuals will have higher diversity than ill ones. However, alpha diversity is not a "reliable and accurate" metric (l. 67), and new metrics must be developed to "provide a significant advancement in current microbiome science" (l. 70). The authors therefore developed a novel index, the GHMI.

I was confused why a novel metric will advance microbiome science, and I found the Introduction's explanation vague. The authors themselves cite Shade's "Diversity is the question, not the answer", which I think posits the question that the authors do not answer here: how does the novel metric explain something previously unknown about the microbiome?

Put another way, if alpha diversity analyses are themselves misguided, why is it good to have an index that is more accurate than alpha diversity?

To my mind, a microbiome index has two important features, accuracy and interpretability. "Accuracy" means that the index must accurately separate cases and controls. "Interpretability" means that the index's separation of cases and controls must lead to some biological hypothesis, as per Shade. The author's index appears more accurate, but I found its mathematical formulation confusing. The identification of health- and disease-associated species, the first step in the index's parameterization, is interpretable, but that brings me back to the meta-analysis rationale above.

Furthermore, if the twin goals are accuracy and interpretability, I would be curious whether simpler, more interpretable functional forms of the index achieve similar accuracy. How does, say, a metric defined as the number of health-associated "present" (e.g., $>10^{-5}$ abundance) compare to the full-blown GHMI?

Third, metrics that can reliably distinguish between health and disease would have a clinical application: continuous monitoring of an individual's stool's bacterial alpha diversity would "detect significant changes or abnormalities in comparison to his/her normal baseline measurements", triggering "additional diagnostic procedures and/or therapeutic interventions". (l. 71 ff.)

Clinically, what is needed more than a generalized alarm bell are diagnostics. If a patient is having their stool routinely analyzed, it would be good to know what was wrong with them! That being said, there is some theoretical utility for a generalized alarm bell. If this were the main focus of the manuscript, then I would pose some testing questions:

- If the goal is to develop an index that is usefully longitudinally, why use the cross-sectional data in this manuscript to develop the index? A timeseries approach would entail vastly different design and evaluation criteria.
- If the goal is the detect changes in comparison to a patient's "normal baseline", why not simply compare today's sample against yesterday's? Each patient would serve as their own control?
- How does the GHMI compare to alternatives? How much better is this metric, which requires fairly extensive parameterization, than alpha diversity? How much worse is it than ensemble methods (e.g., random forest) that are inscrutable but usually very accurate?
- How much extra information does a longitudinal microbiome index add on top of other clinical markers? If the datastream from an Apple Watch gives me more information, I'll just use that

rather than collecting and sequencing stool!

I expect that a clearer focus on one of these rationales could improve the paper. For example, l. 584 ff. mentions theoretical roles that the GHMI could play in FMT, both as a classifier for healthy donors (i.e., rationale #3?) and as a way to select probiotic cocktail communities (i.e., rationale #1). However, there is no discussion of the (lack of) literature about how microbiome screening could improve FMT donor quality, nor of the knowledge gaps in cocktail designs. By trying to address all three rationales at once, I fear that none of them are done any justice.

Methods

As I understood, the GHMI requires two steps of parameterization. First, health- and disease-associated bacteria are identified. Second, the thresholds θ_f and θ_d are selected.

If this is accurate, then I have major reservations about many of the claims made in the manuscript about the accuracy of the index. If it was trained on this dataset, and health- and disease-associated bacteria defined using this dataset, it is no surprise that the index can distinguish between health and disease. (The larger surprise, that there is a cross-study and cross-indication signal of disease, although a major focus of Duvall et al.'s meta-analysis, is not much discussed here.) The typical approaches for investigating a classifier's accuracy, like cross-validation and ROCs, are not used; instead the authors make the claim to accuracy based only on a small validation data set.

Minor points

- Fig 2b, etc.: Why a PCA rather than an MDS, if the goal is to look for separation?
- Fig 2b: Is the PERMANOVA significant when accounting for studies or indications? MiRKAT is a tool that can test for separation while accounting for covariates.
- Fig 4c: Does Shannon diversity separate cases and controls within individual studies? This is akin to what's shown in Fig 5b, but I wonder if lumping all studies together makes Shannon look a worse classifier than it is.
- Claims about "core" microbiome (l. 153) should be introduced by what's known in that literature
- l. 204: A major weakness in the microbiome literature is poor characterization of effect sizes. With the large sample sizes used in this study, a difference in microbiome composition could easily be statistically significant without being scientifically interesting or clinically meaningful.
- l. 212: This approach seems very ad hoc. Why this particular formulation, and not one of dozens of others? What are the design criteria?
- l. 233: I think it's confusing to the reader to show a difficult equation and say to look in the Supplement for an explanation. If the GHMI is the centerpiece of the paper, its development, rationale, and form should be a key part of the text.
- From Supp Table 3, it looks like the classifier's accuracy is not very sensitive to the choices of the threshold parameters. How sensitive are the results to the choice of health- and disease-associated species, and to the threshold parameters?
- l. 295: I found this confusing: doesn't the GHMI have two thresholds and ~100 selections of species as health- or disease-associated? Why is that not "arbitrary"?
- l. 300: The use of the word "potential" implies that the microbiome is causative in disease. This is generally not known.
- l. 312: Why was this one result on sphingolipids the one discussed? It feels like cherry-picking.
- l. 344: Is 71% good? How much better is it than alpha diversity? How much worse is it than a random forest classifier, which is inscrutable but usually a better predictor?
- l. 375 ff. This should be the first Results section! If the claim is that alpha diversity is insufficient, the first Results should substantiate that claim. Many, many studies have claimed that alpha diversity is associated with health. This isn't to say they are right and you are wrong, but it does mean that, if you want to claim they are all wrong, you need to back up the claim!

Reviewers' comments:

Reviewer #1 (Remarks to the Author):

Based on a large cohort of >4K samples, the authors derive a novel species prevalence metric that separates healthy metagenome samples from non-healthy ones. The metric was designed to rely on species prevalence rather than abundances which makes it robust against batch effects and some common confounding. Validation with an additional test set of about 600 individuals demonstrates that the metric is robust even when applied to diseases not present in the training data set.

The question of what constitutes a healthy microbiome is one of the unsolved key problems in the field right now and the manuscript shows a novel and creative way to get closer to an answer. The major strength of the novel microbiome health index presented here is that the authors took a lot of care to circumvent some of the common problems in designing a score based on samples from heterogeneous studies and protocols. In particular, I found the strategy to use a prevalence-based score rather than an abundance score quite appealing. I do have some doubts about using species abundance rather than other taxonomic ranks or gene abundances. Also it should be mentioned that the score classifies absence of diagnosed disease rather than actual health status and that there may be geographical/cultural biases since the training dataset is mostly (but not exclusively) composed of samples from the US, Europe and Asia. Nevertheless the manuscript takes a good first step at defining health through the microbiome and will be of interest to a wide readership and the medical community. The manuscript is well written and provides a clear path from motivation to results.

Authors' response: We very much thank the reviewer for this summary, and for taking her/his valuable time to carefully understand the purpose of our work and its strengths. Before addressing in depth the Reviewer's specific comments, we first provide our thoughts on the valid points mentioned above:

- "I do have some doubts about using species abundance rather than other taxonomic ranks or gene abundances."
 - As detailed in our response to **Major Comment #2**, we've applied our analytical strategy (from the derivation of GMHI to the evaluation of its classification performance) on all taxonomic clades and metagenomic functional profiles.
- "Also it should be mentioned that the score classifies absence of diagnosed disease rather than actual health status and that there may be geographical/cultural biases since the training dataset is mostly (but not exclusively) composed of samples from the US, Europe and Asia."
 - Despite there being no universally-recognized definition of "healthy", we strongly agree that it is important to be clear and precise. Thus, in consideration of the Reviewer's suggestion, we've mentioned throughout our manuscript that GMHI is technically a measure of "known disease absence" or "presence/absence of diagnosed disease", rather than a scale of actual health status. Please see lines 30, 50, 66, 154, 182, 558, and 565 of the revised manuscript.
 - While we definitely tried to be as inclusive as possible (during our dataset search) of various geographies, ethnicities/races, and cultures, we do acknowledge that complete elimination of biases is practically impossible. This is, in part, due to the paucity of high-quality shotgun metagenomic data across diverse populations in comparison to 16s rRNA amplicon datasets (which come with their own limitations, such as widespread use of different hypervariable

regions, and inability to survey the Eukaryota domain of life; PMID: PMC3979728). In light of the Reviewer's suggestion, we've mentioned this point in our manuscript (please see lines 625–630 of the revised manuscript). Certainly, for future works, we plan to iteratively expand our application to reach an even broader range of geographies, ethnicities/races, and cultures.

Next, we address all of the reviewer's suggestions below:

Suggested major changes

1) The paper develops a novel microbiome health index but I feel like the “health” part would require some additional validation. The index itself is designed to separate disease samples from their controls and samples with abnormal BMI. However, it is unclear whether those controls are indeed healthy individuals as the only thing one knows for sure is that they did not show symptoms severe enough to be classified as diseased in the particular study. In order to validate that the microbiome index indeed quantifies health I would like to see at least a validation with a data set that contains a detailed clinical characterization of the cohort to check whether individuals with the highest health index do indeed show better clinical labs, higher indices of self-reported well-being or less ambulant treatments than individuals with a lower index. The authors own RA cohort may be helpful here. Alternatively, the authors could train a genus-level score (as suggested below) and validate the health status on the American Gut data set (<https://doi.org/10.1128/mSystems.00031-18>) which includes a wide array of health measures and self-evaluations.

Authors' response: We agree with the reviewer's concern on whether we are appropriately associating GMHI with actual health. Indeed, this is a tricky endeavor, as the definition of “healthy” is not standardized. To address the concern of whether GMHI can objectively quantify certain aspects of health, we looked for statistical associations between GMHI and well-recognized components of physiological wellness from clinical lab tests (as the reviewer suggests). More specifically, we searched for correlations with GMHI and the following, as reported in their original studies: circulating blood concentrations of fasting blood glucose (from 785 subjects), triglycerides (from 915 subjects), cholesterol (from 521 subjects), low-density lipoprotein cholesterol (LDLC; from 848 subjects), and high-density lipoprotein cholesterol (HDLC; from 841 subjects). Of note, self-reported well-being, treatment regimens, and other questionnaire data were either not provided at all or too sparsely collected to have any practical or statistical significance.

When selecting for moderate correlations or better, i.e., $|\text{Spearman's } \rho| \geq 0.3$ ($P < 0.001$), we identified HDLC as the only feature that was significantly associated with GMHI ($\rho = 0.34$, $P < 2.2 \times 10^{-16}$); in addition, we identified significantly higher abundances of HDLC in subjects with positive GMHI compared to those with negative GMHI (Mann-Whitney U test; $P < 2.0 \times 10^{-16}$):

This moderately positive correlation is encouraging for linking GMHI to actual health, as HDLC in the bloodstream is commonly considered as “good” cholesterol, and could be protective against heart attack and stroke, according to the American Heart Association (www.heart.org). We view these findings to be of high importance, as it not only demonstrates integration of clinical data with gut microbiome, but also hints at the possibility of GMHI serving as an effective and reliable predictor of cardiovascular health. Thus, we’ve included descriptions of this analysis and its results in the revised manuscript (please see **Figure 2** and lines 293–318 of the revised manuscript). In contrast, fasting blood glucose ($\rho = -0.06$), triglycerides ($\rho = -0.13$), cholesterol ($\rho = 0.15$), and LDLC ($\rho = 0.09$) were noted to have only weak and/or insignificant correlations with GMHI.

The reviewer suggested the possibility of training a genus-level score (performed below) to validate on the American Gut dataset, which is a fascinating study in its own right. Although this suggestion looks reasonable on the surface, we refrained due to the following:

- The American Gut dataset is mostly 16s rRNA gene amplicon data. Directly comparing taxonomic abundances between metagenomic data and 16s data is generally not advised due to several reasons, including differences in their clade (or OTU) detection techniques; and technical biases from sequencing different hypervariable regions in the 16s rRNA gene. However, for future efforts, we encourage the development of a complementary GMHI using 16s rRNA gene sequencing data.
- After a close examination, we found that the “wide array of health measures and self-evaluations” from the American Gut study are not focused on specific health measures (e.g., clinical laboratory tests on blood or urine); rather, most of the self-evaluations were in regards to diet/nutrition/alcohol consumption, BMI, exercise habits, food allergies, and other criteria that, in our view, are not clear and objective measures of health.

2) I was surprised the authors chose species as the summary rank for their health index. If one wanted to design a score specifically for metagenome samples better performance would probably be achieved by using bacterial gene abundances directly as this is much closer to a functional analysis. The authors claim themselves that one would expect less functional heterogeneity than taxonomic heterogeneity and show that the identified species share similar genes. I do think that basing an index on taxon abundances is still valuable but would probably be more useful on the genus-level as this opens up validation with many more 16S amplicon sequencing data sets (where reliable taxonomic classification is usually only achieved down to the

genus-level). At the very least I would like to see a comparison of accuracies on various taxonomic ranks to show that the species level does indeed show better performance than other ranks.

Authors' response: Thank you for this concern. Before we provide details on how we addressed this suggestion, let us first explain why we specifically chose species-level information—rather than genus-level or functional (i.e., gene or pathway) abundances—as the ideal search space of features for GMHI:

- All authors of this manuscript share the vision that the most transformative impact human microbiome research can have on the clinical practice for complex, chronic disease is in the creation of robust probiotic therapies. These future therapies, once conceived, will most likely be composed of a small set of microbes (either naturally occurring or synthetically engineered) designed to be delivered inside or onto the patient. (there is a very promising avenue of research on gut microbiome-derived pharmaceuticals, but this lies outside the scope of this study.) Now, in order for these probiotic communities to be formulated, one would need to know precisely which strain(s) to use, as the delivery of an entire phylum, order, or genus doesn't make much practical sense. However, in our view, current computational techniques for comprehensive, strain-level detection in complex microbial communities still have much to improve. Therefore, given the level of taxonomic precision required to realize this vision, as well as the potential to advise and inspire similar future efforts, we decided to conduct our study—from designing the feature selection criteria, to training/validating the classification model, and to comparing classification performances against other predictors (e.g., Shannon diversity, richness, 80% abundance coverage)—entirely upon species-level taxonomy information. We clarify this in lines 107–109, 600–606 of the revised manuscript.

With this being said, we do agree that investigating accuracies of GMHIs derived from all taxonomic ranks, as well as from functional annotations, is a worthwhile endeavor. To this end, we applied our GMHI identification pipeline on abundances of all other taxonomic ranks, as well as on abundances of MetaCyc metabolic pathways obtained through HUMAnN2 (Franzosa *et al.* Nature Methods (2018), PMID: PMC6235447). Of note, to get as much functional insight into the gut microbiota as possible, we chose to query annotated biochemical (MetaCyc) pathways rather than bacterial gene abundances.

The highest 'average classification accuracy' (defined as χ in our manuscript) for each of these were found to be the following: Phylum, 42.1%; Class, 60.1%; Order, 62.4%; Family, 67.2%; Genus, 68.2%; Species, 69.7% (as originally described in our manuscript); and MetaCyc pathways, 59.4%. As evidenced by these results, GMHI based on taxonomic species shows the best classification performance, providing further support to our original findings. Interestingly, a GMHI based on MetaCyc pathway abundances showed second-to-worst performance. We mention this analysis in lines 220–223 of the revised manuscript, and include these data in **Supplementary Table 4**.

3) The index will naturally depend on the training set and may not extend well to cohorts different from the training data. Even though the authors tried to generate a diverse data set it still included very few samples that are not from developed countries. There are only a handful of samples not from the US, Europe or Asia, and it looks like the health index does not perform well for data sets from different cohorts. For instance see the study from Obregon-Tito *et al.* (Peru) in Figure 2 but also several studies from China where the healthy cohorts are primarily classified as "less healthy" (lower index). This limitation should be mentioned in the text

as its misinterpretation in a potential clinical setting could have negative consequences for underrepresented groups.

Authors' response: Certainly, we agree with the reviewer's first point that the capability of any statistically-inferred prediction model to perform well on a validation cohort will depend on the scale and breadth of the training dataset. Indeed, shortcomings to either criterion are known to limit the robustness of biomarkers in the clinical setting (PMCID: PMC3418428); and this can be a concerning issue for those in under-represented populations. This is why, during the very early stages of this study, we paid close attention to collecting as many high-quality metagenome samples as possible, and as broadly/diversely as possible.

Despite our earnest efforts, we do acknowledge that there are very few studies and samples from under-developed countries and minority ethnicities/races, or not from the USA, Europe, or China (as the reviewer points out). The datasets we report in our study were all of what we could find at the time of sample collection (March 2018). One probable reason for this is the relative paucity in shotgun metagenomic datasets from less-recognized communities; this reflects the current state of human microbiome research, and highlights an important area where the scientific community can improve. We hope that our study, which was conducted primarily on a massive collection of "crowd-sourced" data, can initiate serious discussions regarding the broader inclusion of subjects from under-developed countries and minority ethnicities/races. As suggested by the reviewer, we mention this discussion in lines 625–630 of the revised manuscript.

It does appear in Obregon-Tito *et al.* (Peru) and in several studies from China (Qin *et al.*, Zhang *et al.*, Jie *et al.*, Feng *et al.*, Liu *et al.*, and He *et al.*) that "the health index does not perform well", i.e., "the healthy cohorts are primarily classified as "less healthy" (lower index)". However, we would like to respectfully clarify that only statistically significant differences (Mann-Whitney U test, $P < 0.05$) in health index distributions should be considered for further interpretation (marked with a * or *** if GMHI performed as expected; marked with a Ψ or $\Psi\Psi\Psi$ if GMHI performed opposite to expectations). A closer inspection shows that from a total of 28 possible non-healthy phenotype vs. healthy comparisons across different cohorts, GMHI performed as expected, i.e., supporting our original claim of higher GMHIs in healthy groups, in a total of 11 times; whereas GMHI performed opposite to expectations 2 times. In contrast, the other metrics performed as expected in the following number of non-healthy phenotype vs. healthy comparisons: 2 for Shannon diversity; 4 for 80% abundance coverage; and 3 for species richness. We clarify this in lines 444–447 of the revised manuscript, with the hope to avoid any potential misunderstandings regarding the robustness of our health index.

Suggested minor changes

4) Some studies are missing in Figure 2. For instance the study from Sankaranarayanan *et al.* The figure caption should explain why studies were omitted here.

Authors' response: We thank the reviewer for pointing this out. Our criteria for selecting which cohorts to perform intra-study, case-control comparisons is to have both groups be composed of at least 10 samples, which we deemed as reasonably sufficient sample size. Thus, in **Figure 5** (mentioned by the reviewer as **Figure 2**), we show only the 12 independent studies that satisfy this sample size cut-off. As suggested, we've clarified this point in lines 405–409 of the revised manuscript, as follows: "Specifically, in each of the twelve studies (out of 34 total) wherein at least 10 stool metagenome samples from both case (i.e., disease or abnormal bodyweight conditions) and control (i.e., healthy) subjects were available, we compared GMHI,

Shannon diversity, 80% abundance coverage, and species richness between healthy and non-healthy phenotype(s)”. In addition, we removed Obregon-Tito *et al.* (20 samples of Healthy vs. 9 samples of Overweight) and Karlsson *et al.* (8 samples of Healthy vs. 14 of Symptomatic atherosclerosis), which were mistakenly included in the original analysis. We also updated **Figure 5** to reflect this change.

5) The gene copy number analysis presented in Figure 3 seemed a bit inaccurate since it only used data from reference strains which are probably not representative of the actual strains in the samples. Metagenome data allows to quantify gene copy number directly from the sequencing data. This can be achieved by metagenomic assembly and de novo gene prediction. Metagenomic binning should still allow to connect gene abundances back to the Metaphlan marker genes. This would be much more representative of shared functions than some arbitrary reference strains.

Authors’ response: We sincerely thank the reviewer for catching this. We definitely see your point. Certainly, we want to avoid making spurious claims regarding the functional potential of our Health-prevalent and Health-scarce species from arbitrarily-chosen reference strains. We agree that there is a better way, in particular by taking into consideration the gene abundances in the metagenomic sequencing data, as suggested by the reviewer. Rather than using genomes of reference strains, we’ve addressed this concern accordingly:

- We used a metagenomic binning approach to construct KEGG gene copy number profiles for all 50 Health-prevalent and Health-scarce species. More specifically, gene families (annotated through UniRef90 identifiers) were identified in each metagenome sample of our meta-dataset using HUMAnN2 (Ref. 54) with default parameters. HUMAnN2, which can identify a gene family’s total abundance broken down into the contributions from individual species, was used to generate organism-specific, gene abundance profiles for all samples. Next, abundance profiles of only the genes that mapped to a particular Health-prevalent or Health-scarce species were retained (202,826 of ~1.7 million genes from all 4,347 metagenome samples). Then, organism-specific gene abundances were summed together across all samples, and then normalized to Copies Per Million (CPMs); this results in cumulative copy number abundances per gene and per species. Afterwards, protein sequences of gene families were downloaded from the UniProt database (<https://www.uniprot.org/>) and were mapped onto the Kyoto Encyclopedia of Genes and Genomes (KEGG) database (KEGG Release 94.0, April 1, 2020) to determine their functional KEGG orthologs (27,312 of the total 202,826 genes mapped to 4,714 KEGG functional orthologs). Finally, gene (KEGG ortholog) copy number profiles were constructed for all 50 Health-prevalent and Health-scarce species.
- We revised a subsection of the **Methods** section to document these steps. Please see lines 802–815 of the revised manuscript.

With this new strategy of quantifying gene copy number directly from sequencing data (as suggested by the reviewer), we performed gene copy number variation analysis to identify potential metabolic functions specific to Health-prevalent species. Our results are described as follows, and included in lines 243–258 of the revised manuscript:

- To uncover genes that could explain differences in functional potential between the Health-prevalent (n = 7) and Health-scarce (n = 43) species, we constructed KEGG gene copy number profiles of all 50 species using a metagenomic binning approach (see **Methods**). In **Supplementary Fig. 5** (shown in

the heatmap below), we show the 193 KEGG functional orthologs that were identified to have significant differences in copy number between the Health-prevalent and Health-scarce species ($q < 0.05$, Mann-Whitney U test with Benjamini-Hochberg FDR correction; **Supplementary Table 7**).

- Hierarchical clustering of gene copy number profiles revealed that six of the seven Health-prevalent species (*Alistipes senegalensis*, *Bacteroides bacterium-ph8*, *Bifidobacterium adolescentis*, *Bifidobacterium angulatum*, *Bifidobacterium catenulatum*, and *Sutterella wadsworthensis*) clustered together, reflecting their functional similarities. Interestingly, all 193 genes were found to have, on average, higher copy numbers in Health-prevalent species than in Health-scarce species. In particular, these genes include enzymes involved in the metabolism of sugars and polysaccharides (e.g., α -amylase, α , α -trehalase, Fructose-bisphosphate aldolase, D-xylulose reductase, Xylan 1,4- β -xylosidase) and lipids (e.g., Long-chain acyl-CoA synthetase, Enoyl-acyl carrier protein reductase). Furthermore, through KEGG pathway enrichment analysis, we found that these genes with

differential copy number were over-represented in KEGG pathway modules related to amino acid metabolism (e.g., Arginine biosynthesis, Isoleucine biosynthesis, Leucine biosynthesis, and Urea Cycle), nucleotide biosynthesis (e.g., Nucleotide sugar biosynthesis and Guanine ribonucleotide biosynthesis), and Lipid A biosynthesis (Kdo₂-lipid A biosynthesis, Raetz pathway) ($P < 0.05$, hypergeometric test; **Supplementary Table 8**). Although it yet remains unclear regarding how, and in which context, these genes and over-represented pathways are utilized by Health-prevalent species, our findings nonetheless shed further light on the underlying functional potential of microbes prevalent in the gut of healthy individuals.

In order to address Reviewer #2's request to better streamline the text and delivery of the main message, we have moved the figure pertaining to this analysis into the supplementary.

6) I feel like the formula for the health index (h) should be motivated and explained a bit better. The description in the supplement does describe *how* the index is calculated but I would have liked to read *why* the authors chose that particular formula. For instance, the weighting with the Shannon diversity makes sense to me as it prefers samples with a high diversity of health-related microbes, but the description is a bit complicated and may not be accessible to a wide readership - especially one that is not familiar with alpha diversity or the Shannon index.

Authors' response: We emphatically agree to any suggestion regarding how the clarity of our work can be improved. Moreover, this particular area of focus was also mentioned by Reviewer #2. We revised the manuscript accordingly:

- To provide more clarity behind the *why* of our formula (as the reviewer pointed out), we made an effort to better explain the overall motivation of its conceptual design, as well as the logical rationale of each major step. Please see lines 144–155 and 260–277 of the revised manuscript.
- We've moved much of the technical jargon into the **Methods** section of the main text. Please see lines 707–800 of the revised manuscript.

We sincerely hope these efforts have made the text more accessible to a wide readership.

7) The authors claim that their measure is robust against batch effects but this is not demonstrated. The authors should show that the distribution of the index for healthy individuals does not vary between studies and does not depend on library size.

Authors' response: We would like to kindly note that the results of our analyses in **Figure 5** ("GMHI generally outperforms other microbiome ecological characteristics in distinguishing case and control across multiple study-specific comparisons.") and in **Figure 6** ("GMHI demonstrates strong reproducibility on validation datasets and outperforms Shannon diversity.") were to specifically address this robustness and batch effects issue. However, we agree that further demonstration of robustness against batch effects is warranted. To address the reviewer's two suggestions, we performed the following:

- We tested the hypothesis that library size (i.e., read count) of a sample is significantly correlated with its GMHI. First, we visualized this relationship for all metagenome samples in the following scatter-plot:

In our view, we did not observe a strong trend between the two parameters. In addition, we modeled GMHI using a mixed-effects linear regression model ('lmer' function in the R package 'lme4'), wherein model covariates consisted of read count and study of origin (the latter as a random effect to accommodate for intra-study variance). Our model found no significant association between library size and GMHI ($P = 0.45$). Based on these results, we were not able to reject the null hypothesis, and conclude that GMHI does not depend on library size. We mention this analysis in lines 289–290 of the revised manuscript, and show our results in **Supplementary Figure 6**.

- We checked to see whether or not the distributions of the indices for healthy individuals vary between studies. Among the 34 studies used in our training dataset (as referenced in **Table 2**), 31 studies were initially chosen for a closer analysis, as each of these studies contain gut microbiome samples from healthy subjects. We note that Sankaranarayanan *et al.* was not considered, as it has only a single sample from healthy; in addition, we merged the two HMP1 studies, i.e., Huttenhower *et al.* (HMP1) and Lloyd-Price *et al.* (HMP1-II). Below, we show the distribution of GMHIs of every sample from the 29 independent sources (i.e., studies):

As is often the case for outputs from statistically-inferred models, we observed wide variation among the GMHI distributions from study to study. Among all pairwise comparisons between study groups, we found only one pair of cohorts whose distributions were significantly different from each other (Dwass-Steel-Critchlow-Fligner test followed by Holm-Bonferroni method to control for family-wise error rate). This shows that, by and large, the distributions of the index for healthy individuals do not vary much between studies. Furthermore, we found that most healthy cohorts (22 of the 29 independent sources) show positive GMHI distributions based on their medians. We mention this analysis in lines 290–291 of the revised manuscript, and show our results in **Supplementary Figure 7**.

8) I would appreciate if some more of the source code is provided to reproduce the study. For now the authors include a script that calculates the health index but do not provide materials that would allow running the Metaphlan analysis on the full data set used in the study. The provided SRA accession for the authors' data does not seem to exist (may still be under embargo).

Authors' response: Absolutely! We definitely value the importance of assisting the community in reproducing our work. For all of our results shown in the main figures, we provide our original R scripts in our laboratory's GitHub link. The data required to run our scripts are also in this online directory. As such, we changed the text in our **Methods** section accordingly:

- (prior) "**Code availability.** An R script on how to calculate GMHI for a given stool metagenome sample is available at https://github.com/jaeyunsung/GMHI_2020."
- (now) "**Code availability.** R scripts demonstrating how to reproduce all of our findings shown in the main figures, as well as how to calculate GMHI for a given stool metagenome sample, are available at https://github.com/jaeyunsung/GMHI_2020."

And yes, the SRA accession to download the raw sequence files of our Rheumatoid Arthritis gut microbiome data will not work until we've granted permission (the PRJNA accession # is correctly provided in the manuscript). We will lift the embargo to download the data once our manuscript has been accepted for

publication. However, for this review process, we grant the reviewer temporary access to view our submission status at this non-traceable link:

<https://dataview.ncbi.nlm.nih.gov/object/PRJNA598446?reviewer=tn9eo9v66meg2rjpv6hvf8lt7>.

Reviewer #2 (Remarks to the Author):

In their manuscript, Gupta and colleagues posit a functional form for a microbiome index, parameterize the index using a large dataset of cross-sectional microbiome studies, then investigate the metagenomic meaning of the index and compare its performance in classifying cases vs. controls in the dataset of microbiome studies.

I have serious concerns about the validity and utility of this approach. I was confused by the rationale for the work and the organization of the manuscript. My approach will be to re-construct the authors' argument and key claims, noting what I think are the key issues in each case.

Authors' response: Thank you, Dr. Oleson, for your careful and astute critique of our work. We can tell that you've put a lot of time into finding how we can improve the paper, which is our ultimate goal. We also commend your choice to having a transparent review process; we need more of this in academia.

We are truly sorry to hear that we were unable to clearly and precisely articulate the rationale, validity, and utility of our work in the original draft. Surprisingly, this assessment is in stark contrast to Reviewer #1's comments, which are as follows:

- *"The major strength of the novel microbiome health index presented here is that the authors took a lot of care to circumvent some of the common problems in designing a score based on samples from heterogeneous studies and protocols. In particular, I found the strategy to use a prevalence-based score rather than an abundance score quite appealing."*
- *"Nevertheless the manuscript takes a good first step at defining health through the microbiome and will be of interest to a wide readership and the medical community."*
- *"The manuscript is well written and provides a clear path from motivation to results."*

Nevertheless, we've tried our best to address all of your concerns and suggestions. We expect that most (and hopefully all) satisfy your standards of excellence; and for where we fail, we hope to be given another opportunity to put even more effort to improve.

Rationale

As I understood, this work is motivated by three rationales.

Authors' response: We respect the reviewer's viewpoint that the motivation of our work was not made precisely clear. In brief, our work was motivated by a single primary rationale, as our title suggests: To create a simple measure to quantify the degree of general health status (i.e., presence/absence of diagnosed disease) based on the microbiome of a stool specimen. (currently, there is no reliable metric for monitoring and predicting general health based on stool metagenomic profiling alone.) In other words, our mission was to be in

a position to answer the following: “Does my gut microbiome reflect more of a healthy or non-healthy state, and to what degree?”.

In order to build our index, we must first find solutions to the following critical problems. We suspect that, in our earnest attempt to address these issues simultaneously, the delivery of our narrative may have been perceived as convoluted and confusing.

- Q1: Upon what biological basis should we formulate the mathematical formula of our index?
 - We envision that the most intuitive way to determine how closely one’s microbiome resembles that of a healthy (or non-healthy) population is to quantify the balance between health-associated microbes *relative to* disease-associated microbes. Therefore, our index is a *rational equation* between two sets of microbial species: those that are *more* frequently observed in healthy compared to non-healthy populations (i.e., ‘Health-prevalent’ species) vs. those that are *less* frequently observed in Healthy compared to Non-healthy populations (i.e., ‘Health-scarce’ species).
- Q2: Then how can we identify ‘Health-prevalent’ and ‘Health-scarce’ species?
 - One would need a large-enough sample size to truly identify robust signals. In order to obtain the largest collection of stool metagenomic data, we *pooled together* the enormous compendium of publicly-available datasets, which were derived from healthy and non-healthy human subjects. As detailed in the main manuscript, we used these data to identify two sets of species associated with healthy gut microbiomes: 7 and 43 species more (‘Health-prevalent’) and less (‘Health-scarce’) frequently observed, respectively, in healthy compared to non-healthy groups of people. This discovery alone advances our understanding of the composition of a healthy gut microbiome that has been long sought after. Finally, with these two sets of species, we went on to tune the parameters of a predefined formula, as well as to test its classification performance.
- Q3: In regards to classification performance, to which metrics should we compare our index?
 - For this, we chose to compare our index with Shannon diversity and other ecological properties. Furthermore, the application of our index is designed to be independent of whether one chooses to analyze her/his microbiome at a single time-point or longitudinally.

We hope this clarification makes sense, and creates a clearer picture for the reviewer as he proceeds with reviewing our responses below. And to address the reviewer’s serious concerns regarding focus and clarity, we’ve done some major restructuring of the manuscript to get the aforementioned points across better. Mainly, we’ve included these main points upfront, and moved some technical jargon to the **Methods** section. However, we still kept most of the math (either in the main text or the supplementary) for our more mathematically-inclined readers. Below, we provide further details on how we revised the manuscript based on each and every comment.

First, there are barriers to understanding the role of the microbiome in health and disease when using individual studies (line 76 and following). Therefore, meta-analyses comparing health and disease across studies, populations, and diseases is important to microbiome science. Table 2 and Figure 3 seem to be the fulfillment of this meta-analysis rationale, but the results are presented through the lens of the creation of the novel index, which makes it difficult to interpret the results as a meta-analysis per se.

Authors' response: We were incorrect in our use of the term “meta-analysis”. Here, we are certain that the reviewer considers a meta-analysis to be, in general, a comparative investigation of findings across various studies—which are independent of each other but address the same scientific question—in order to derive conclusions about a particular body of research. This is not our original intention nor what was performed; rather, we performed a ‘pooled analysis’ by combining samples from multiple studies to re-analyze in aggregate. Thereby, we are taking advantage of stool metagenomic datasets abundantly available to the public. This was described throughout our manuscript, as shown in the following excerpts:

- **“A meta-dataset of human stool metagenomes integrated across 34 independent published studies.** An overview of our multi-study integration approach, wherein we acquired 4,347 raw shotgun stool metagenomes (2,636 and 1,711 metagenomes from healthy and non-healthy individuals, respectively) from 34 independent published studies, is depicted in Fig. 1.”
- “Herein, we address this challenge accordingly: i) by integrating massive amounts of publicly available data (4,347 publicly-available, shotgun metagenomic data of gut microbiomes from 34 published studies), we identified a small consortium of 50 microbial species associated with human health.”
- “In the long term, in order to find robust gut microbiome-based diagnostic or predictive markers, we envision integrating even larger data repositories to take into consideration more sources of heterogeneity.”
- “Our efforts to curtail these batch effects include: i) consensus preprocessing, i.e., downloading all raw shotgun metagenomes (.fastq files) and re-processing each sample uniformly using identical bioinformatics methods;”
- **Multi-study integration of human stool metagenomes** section in **Methods**.

We do recognize the value of being clear on the semantics. Therefore, we've replaced all mentions of “meta-analysis” in our manuscript to “pooled analysis”.

If a meta-analysis were the focus of the paper, I would have expected the authors to carefully survey previous meta-analyses and make a clear advancement over them in terms of dataset, methodology, or interpretation. It was not clear to me how the novel index was a methodological advance in the identification of health- and disease-associated species.

Authors' response: We agree that, if a meta-analysis were the focus of our study, there is no excuse for not having done a careful survey of previous meta-analyses, especially in regards to what has been done; what were the conclusions; what were the limitations. Fortunately, our study is not a meta-analysis, but a pooled analysis by combining samples from multiple studies to re-analyze in aggregate (as noted above). We've replaced all mentions of “meta-analysis” in our manuscript to “pooled analysis”. Lastly, we address the reviewer's last remark: The biological basis of the index itself is to quantify the balance between health-associated microbes and disease-associated microbes (see lines: 67, 146, 561, and 567); and not a methodology to identify health- and disease-associated species per se. We clarify this in lines 144–155 of the revised manuscript.

Second, a common analysis in microbiome studies comparing cases and controls is bacterial alpha diversity, with the expectation that health individuals will have higher diversity than ill ones. However, alpha diversity is not a “reliable and accurate” metric (l. 67), and new metrics must be developed to “provide a significant advancement in current microbiome science” (l. 70). The authors therefore developed a novel index, the GHMI.

I was confused why a novel metric will advance microbiome science, and I found the Introduction's explanation vague. The authors themselves cite Shade's "Diversity is the question, not the answer", which I think posits the question that the authors do not answer here: how does the novel metric explain something previously unknown about the microbiome?

Authors' response: We respectfully clarify the following:

- The primary goal of this study is not to make novel discoveries on the human gut microbiome per se, e.g., host interactions, ecological dynamics. Rather, our main objective is to develop, demonstrate, and validate a novel translational application of the gut microbiome.
- The specific application is for distinguishing human subjects who are healthy from those who are not. Primarily, we asked ourselves the following: If we can create a marker for health that is better than alpha diversity (in regards to accuracy *and* interpretability), how is this not an advancement in human microbiome research pertaining to health?

Put another way, if alpha diversity analyses are themselves misguided, why is it good to have an index that is more accurate than alpha diversity?

Authors' response: We are a bit puzzled by the premise of this question. We have not stated anywhere in the manuscript that "alpha diversity analyses are themselves misguided"; to clarify, we state that, in the context of distinguishing health from disease, GMHI not only generally outperforms other microbiome ecological characteristics, but also demonstrates stronger reproducibility on validation datasets (as evidenced by our data). In other words, our primary objective was to provide a better alternative to alpha diversity (in regards to classification accuracy and biological interpretation), but this does not necessarily mean that "alpha diversity analyses are themselves misguided".

To my mind, a microbiome index has two important features, accuracy and interpretability. "Accuracy" means that the index must accurately separate cases and controls. "Interpretability" means that the index's separation of cases and controls must lead to some biological hypothesis, as per Shade. The author's index appears more accurate, but I found its mathematical formulation confusing.

Authors' response: We emphatically agree to any suggestion regarding how the clarity of our work can be improved. Moreover, this particular area of focus was also mentioned by Reviewer #1. We revised the manuscript accordingly:

- To provide more clarity behind the *why* of our formula (as the reviewer pointed out), we made an effort to better explain the overall motivation of its conceptual design, as well as the logical rationale of each major step. Please see lines 144–155 and 260–277 of the revised manuscript.
- We've moved much of the technical jargon into the **Methods** section of the main text. Please see lines 707–800 of the revised manuscript.

We sincerely hope these efforts have made the text more accessible and less confusing.

The identification of health- and disease-associated species, the first step in the index's parameterization, is interpretable, but that brings me back to the meta-analysis rationale above.

Authors' response: We kindly note that a meta-analysis of various gut microbiome studies was not our goal.

Furthermore, if the twin goals are accuracy and interpretability, I would be curious whether simpler, more interpretable functional forms of the index achieve similar accuracy. How does, say, a metric defined as the number of health-associated "present" (e.g., $>10^{-5}$ abundance) compare to the full-blown GHMI?

Authors' response: We envision that the most intuitive way to determine how closely one's microbiome resembles that of a healthy (or non-healthy) population is to quantify the balance between health-associated microbes *relative to* disease-associated microbes. Therefore, our index is a *rational equation* between two sets of microbial species: those that are *more* frequently observed in healthy compared to non-healthy populations (i.e., 'Health-prevalent' species) vs. those that are *less* frequently observed in Healthy compared to Non-healthy populations (i.e., 'Health-scarce' species).

In our view, quantifying the presence of only the health-associated microbes has three major limitations:

- Generally in supervised classification, it is common practice to take into consideration the features that are associated with both phenotypes when building a classifier.
- The health-associated microbes, despite being found to have higher prevalence in healthy subjects, are not totally absent in subjects with disease. This emphasizes the importance of weighing both sets of microbes against each other.
- If our metric is to be defined purely by the count of health-associated microbial species that are present, then defining the appropriate threshold for future predictions is not a trivial endeavor. And even if we did determine a threshold, say from an internal cross-validation procedure, we don't strongly feel that the answer to the question of "Why that particular number?" would be straightforward. In contrast, given that our index is a logarithm of a fold-change ratio, a positive and negative value signifies more and less of health-associated microbes compared to disease-associated microbes, respectively.

Nonetheless, we performed the following analysis of using only the health-associated microbes, as requested by the reviewer:

- Since there are 7 microbial species identified as 'Health-prevalent', we classified each of the 4,347 metagenome samples in the training dataset as healthy if at least 1 of the 7 Health-prevalent species was present. This led to an average classification accuracy of 54.9%. Analogously, we classified each sample as healthy if at least 2 of the 7 Health-prevalent species were present (average classification accuracy: 61.3%). Continuing in an iterative manner, we obtained an average classification accuracy of 65.3%, **66.3%**, 61.1%, 54.5%, and 51.4% when the minimally required count of present Health-prevalent species was set to 3, 4, 5, 6, and 7, respectively.
- Next, we used this approach on the 679 metagenome samples of the independent validation dataset; for this, we set 4 as the minimally required count of present Health-prevalent species (for a sample to be classified as healthy), as this threshold gave the best results with the training data. The average classification accuracy on the validation dataset resulted in 59.3%. In stark contrast, GMHI displayed far better classification performance by achieving an average classification accuracy of 69.7% and 73.7% in the training and validation datasets, respectively.

A summary of these methods and results are provided in lines 470–476 of the revised manuscript and in **Supplementary Table 12**.

Third, metrics that can reliably distinguish between health and disease would have a clinical application: continuous monitoring of an individual's stool's bacterial alpha diversity would "detect significant changes or abnormalities in comparison to his/her normal baseline measurements", triggering "additional diagnostic procedures and/or therapeutic interventions". (l. 71 ff.)

Authors' response: Some clarification is necessary prior to our addressing the next question. We are not at all advocating for the "continuous monitoring of an individual's stool's bacterial alpha diversity", as the reviewer implies above. Rather, we are advocating for a "simple and biologically interpretable metric" composed of gut microbiome information that is a more "reliable and accurate measure to explain variations in health traits" than alpha diversity. Now, once this method is developed (hence, the goal of this study), we can then possibly use it to "detect significant changes or abnormalities in comparison to her/his normal baseline measurements"; and if a certain anomaly is detected, "additional diagnostic procedures and/or therapeutic interventions" can possibly be employed.

We were simply introducing an exciting, yet hypothetical, scenario of how our index may be applied as clinically-actionable information. Yet, we realize that several questions have been raised by the reviewer (below) based on this purely hypothetical scenario. To avoid further possible confusion, we've moved this part to the **Discussion** section in lines 631–641 of the revised manuscript.

Clinically, what is needed more than a generalized alarm bell are diagnostics. If a patient is having their stool routinely analyzed, it would be good to know what was wrong with them! That being said, there is some theoretical utility for a generalized alarm bell.

Authors' response: All co-authors, including a clinical team composed of Dr. John M. Davis III (Associate Professor of Medicine, Division of Rheumatology, Department of Medicine, Mayo Clinic), Dr. Konstantinos N. Lazaridis (Professor of Medicine, Division of Gastroenterology and Hepatology, Department of Medicine, Mayo Clinic), and Dr. Heidi Nelson (Emeritus Professor of Surgery, former Chair of Department of Surgery, Mayo Clinic), have discussed this particular comment with great interest. Our collective response is as follows:

- Understanding what constitutes a deviation from "normal" health, and learning how to rapidly and robustly detect such deviations, are highly important areas of study in academic medicine today. Without a doubt, precisely addressing these challenges will help accelerate the development of new technologies for detecting early signs of disease *prior to the occurrence of specific, diagnosable symptoms*. Therefore, the creation of algorithm-driven markers that can infer one's general health state, especially from biospecimens that can be collected regularly and non-invasively, is a very promising avenue forward. Furthermore, with further development, our gut microbiome-based predictor could serve as an appealing contribution towards comprehensive medical and preventive health screening programs. Results from such tests can then serve as an entry point for follow-up tests and procedures. In this sense, "generalized alarm bells" goes beyond merely having "some theoretical utility", and are just as in need as novel diagnostics designed for a particular disease. Both goals are not mutually exclusive and should be pursued together. And yes, identifying the specific malady is crucial once symptoms arise, but that is not what is being sought after in this study (that would be the next step). In our view, we demonstrate unequivocally the proof-of-concept that our health index—based on a snapshot of the gut microbiome—could have practical merit in the clinical setting, and we fully stand by the results and conclusions of this study.

Let's consider an example regarding credit scores. Say, after a long steady period of having a good credit score (750+), one suddenly receives a report of 590. Assuming this person is a financially-responsible being, this precipitous drop in score would certainly lead one to check her/his latest bank statements, credit card bills, mortgage payments, etc. Hence, although the cause for a significant drop in credit score is initially unclear, it would nonetheless spur action to find out the why and what to do next. Analogously in the case of maintaining wellness and preventing disease, our hope is that GMHI may one day serve as a tool for turning microbiome data into actionable information.

If this were the main focus of the manuscript, then I would pose some testing questions:

- If the goal is to develop an index that is usefully longitudinally, why use the cross-sectional data in this manuscript to develop the index? A timeseries approach would entail vastly different design and evaluation criteria.

Authors' response: We respectfully clarify that our primary motivation is *not* to “develop an index that is usefully longitudinally”; the application of our index is designed to be independent of whether one chooses to analyze her/his microbiome at a single time-point or longitudinally. If we create a health metric that works today, then of course we should make it useful tomorrow, a week from now, a month later, or after a major perturbation (e.g., antibiotic consumption, recovery from food poisoning). This is analogous to the routine use of a cholesterol test to evaluate cardiovascular health; a credit score (which has its own field of complex algorithms) for one's financial health; and so forth. Hence, our index is a metric that can be used at any point in time, and as frequently as one wishes. No complicated dynamic modeling approaches were involved nor deemed necessary.

- If the goal is the detect changes in comparison to a patient's "normal baseline", why not simply compare today's sample against yesterday's? Each patient would serve as their own control?

Authors' response: We thank the reviewer for these questions. Our multi-faceted answer is as follows:

- Absolutely, but how? Detect changes *in exactly what* between today's and yesterday's samples? And why? — these were some of the main questions we had during the inception of this study.
- If we aim to have a general predictor of health status, it makes sense to use a metric *specifically designed* for evaluating health status by using actual metagenomic data spanning a wide range of healthy and non-healthy phenotypes. This is in contrast to ecological properties (e.g., Shannon diversity, richness), which were originally designed for studying ecology per se.
- When we used the term “normal baseline”, we were actually loosely alluding to a general time-point in the past (which includes yesterday!) when the subject was in an asymptomatic state and feeling well.
- Yes, we're totally on board with each patient serving as her/his own control. We would like to note that our current ongoing efforts involve this strategy so that we may perform this suggested analysis in future studies. Although tempted, we felt expanding on this lies a bit outside the scope of this study.

- How does the GHMI compare to alternatives? How much better is this metric, which requires fairly extensive parameterization, than alpha diversity? How much worse is it than ensemble methods (e.g., random forest) that are inscrutable but usually very accurate?

Authors' response: The reviewer brings up a good point in regards to comparing the classification performance of GMHI to that of a fairly simple metric (Shannon diversity) and to that of a more intricate classification algorithm (Random Forest). Our analysis is as follows:

- First, we examined how well Shannon diversity can distinguish healthy from non-healthy groups. As Shannon diversity doesn't have a clear cut-off value to serve as a threshold for discriminating the two groups (in contrast, a sample with a positive and negative GMHI value is classified as healthy and non-healthy, respectively), we decided to apply three different thresholds and evaluate their performances separately: among the Shannon diversity measurements from all 4,347 samples of the training dataset, we selected: i) the 1st quartile (=2.50); ii) the median (=2.85); and iii) the 3rd quartile (=3.11). More specifically, any sample with a Shannon diversity equal to or greater than each threshold is classified as healthy; otherwise, as non-healthy. The average classification accuracy on the training dataset (4,347 samples) when using a threshold of Q_1 , median, and Q_3 was found to be 52.9%, 53.6%, and 53.5%, respectively. Furthermore, on the independent validation dataset (679 samples), the average classification accuracy when using a threshold of Q_1 , median, and Q_3 was found to be 43.0%, 47.0%, and 48.5%, respectively. In stark contrast, GMHI displayed far better classification performance by achieving an average classification accuracy of 69.7% and 73.7% in the training and validation datasets, respectively. Clearly, GMHI outperforms Shannon diversity. We've mentioned this analysis in lines 470–476 of the revised manuscript and in **Supplementary Table 13**.
- Next, we performed a similar analysis using a Random Forest classifier ('scikit-learn' Python package version 0.23.1; data curation and model implementation was performed in Python version 3.6.4) trained and tested upon our dataset of 4,347 metagenome samples. The model achieved a remarkable average classification accuracy of 98.5%—however, building complex decision rules entails the risk of over-fitting. Surely enough, this nearly perfect accuracy was mostly in part a result of outstanding over-fitting, evidenced by the poor performance of 52.3% (average classification accuracy) on the 679 samples of the validation dataset. Our results show that building more complex classifiers can entail a great risk of over-fitting onto the training data, and thus can lead to poor generalization onto unseen cases. We've mentioned this analysis in lines 476–480 of the revised manuscript.

- How much extra information does a longitudinal microbiome index add on top of other clinical markers? If the datastream from an Apple Watch gives me more information, I'll just use that rather than collecting and sequencing stool!

Authors' response: We respectfully clarify that GMHI was not designed to supersede well-established clinical health parameters. Simply put, Gut Microbiome Health Index (GMHI) provides insight into the presence/absence of diagnosed disease using the gut microbiome's balance between health-associated (i.e., Health-prevalent) and disease-associated (i.e., Health-scarce) species. We envision that it may be used to complement widely-applied clinical measures of health, such as circulating HDL cholesterol, triglycerides, c-reactive protein (CRP), fasting blood glucose level, auto-antibodies, etc. If one chooses to weigh more interest in sleep patterns, heart rate, number of steps taken, and other parameters that can be gauged by an Apple Watch, then that is certainly her/his choice.

I expect that a clearer focus on one of these rationales could improve the paper. For example, l. 584 ff. mentions theoretical roles that the GHMI could play in FMT, both as a classifier for healthy donors (i.e.,

rationale #3?) and as a way to select probiotic cocktail communities (i.e., rationale #1). However, there is no discussion of the (lack of) literature about how microbiome screening could improve FMT donor quality, nor of the knowledge gaps in cocktail designs. By trying to address all three rationales at once, I fear that none of them are done any justice.

Authors' response: We removed the speculation regarding FMT and probiotic cocktail design from the Discussion section. We were simply presenting a hypothetical scenario of an application, but we see how it could be a distraction to the delivery of our main point. We no longer feel it is in the manuscript's best interest to include these discussions, as our algorithm was not specifically designed for these specific purposes.

Methods

As I understood, the GHMI requires two steps of parameterization. First, health- and disease-associated bacteria are identified. Second, the thresholds θ_f and θ_d are selected. If this is accurate, then I have major reservations about many of the claims made in the manuscript about the accuracy of the index. If it was trained on this dataset, and health- and disease-associated bacteria defined using this dataset, it is no surprise that the index can distinguish between health and disease.

Authors' response: This is an incorrect understanding of our algorithm, and we regret not having been clearer. To clarify, there are three major steps in our computational pipeline:

1. For every possible pairwise combination of the prevalence fold-change threshold (θ_f) and the prevalence difference threshold (θ_d) (further details on the definitions of θ_f and θ_d can be found in lines 164–167 of the revised manuscript), Health-prevalent (M_H) and Health-scarce (M_N) species that simultaneously satisfy both thresholds in the training dataset (composed of 4,347 stool metagenome samples) are obtained. Thus, each pair of thresholds leads to its respective set of Health-prevalent and Health-scarce species.
2. Then, each set of Health-prevalent (M_H) and Health-scarce (M_N) species are used to find their 'collective abundance' in sample i ($\Psi_{M_H,i}$ and $\Psi_{M_N,i}$, respectively). In turn, h_{i,M_H,M_N} , which is the log-ratio of $\Psi_{M_H,i}$ to $\Psi_{M_N,i}$ is used to classify that sample i as healthy ($h_{i,M_H,M_N} > 0$), non-healthy ($h_{i,M_H,M_N} < 0$), or neither ($h_{i,M_H,M_N} = 0$). Accordingly, the average classification accuracy (χ_{M_H,M_N}), defined as the average of the proportion of 2,636 healthy and of 1,711 non-healthy samples (from our training dataset) that were correctly classified, is found. Thus, each set of M_H and M_N species leads to its respective χ_{M_H,M_N} .
3. Finally, the classification model h_{i,M_H,M_N} (along with its inputs M_H and M_N) that results in the highest average classification accuracy on the original training data is chosen as our final model, i.e., the Gut Microbiome Health Index.

Technically, parameterization of the classification model is only necessary when finding $\Psi_{M_H,i}$ and $\Psi_{M_N,i}$ as each is dependent on the species' identities of M_H and M_N , respectively. But fundamentally, our entire automated computational pipeline begins with the choice of the two prevalence thresholds during the comprehensive, pairwise screening process. Of note, we chose to simultaneously test two thresholds, rather than one, in order to increase our confidence in the robustness of M_H and M_N , as well as to overcome limitations/biases that can occur from using only one type of threshold.

We are not exactly sure what the reviewer meant by his premise "If it was trained on this dataset, and health- and disease-associated bacteria defined using this dataset, ...". We hope the following clarification helps: In

our study, we first used the ‘apparent accuracy’, i.e., classification performance of a model when tested on the training data from which it was derived, to see whether it was worth pursuing further validation in cross-validation (discussed below) and/or on an external validation dataset. There would be little motivation to pursue further with our classification model if the apparent accuracy came out to be quite poor. The resulting average classification accuracy of 69.7% was high enough, in our view, to warrant downstream analyses.

We hope these explanations satisfy the reviewer, and possibly correct, any misunderstanding of the GMHI derivation process. To provide more clarity behind the *why* of our formula, we made an effort to better explain the overall motivation of its conceptual design, as well as the logical rationale of each major step. Please see lines 144–155 and 260–277 of the revised manuscript. Furthermore, step-by-step protocols of all major steps are now included in the **Methods** section (see lines 707–800).

(The larger surprise, that there is a cross-study and cross-indication signal of disease, although a major focus of Duvall et al.'s meta-analysis, is not much discussed here.)

Authors’ response: We agree with the reviewer’s comment! One reason why GMHI’s classification performance extends reasonably well onto the independent validation dataset may indeed be due to our metric reliably capturing cross-study and cross-indication signals of disease (as well as cross-study signals of health). Please see lines 493–498 of our revised manuscript on how we included Duvall et al.’s meta-analysis study in the interpretation of our own results:

“Overall, the remarkable reproducibility of GMHI implies that the highly diverse and complex features of gut microbiome dysbiosis implicated in pathogenesis were reasonably well captured during the dataset integration and original formulation of GMHI. Thereby, our results support previous findings by Duvall et al. in regards to the presence of a generalized disease-associated gut microbial signature, which was observed to be shared across multiple studies and pathologies¹⁷.”

The typical approaches for investigating a classifier's accuracy, like cross-validation and ROCs, are not used; instead the authors make the claim to accuracy based only on a small validation set.

Authors’ response: Actually, it is widely accepted in the biomarker discovery community that the best standard to investigate the robustness of a classification model is to demonstrate its reproducibility on an independent external validation dataset. (the senior author of this study has solid scientific training in using computational algorithms for biomarker discovery, as evidenced in his publications; PMID: PMC3723500, PMC4201588, PMC3418428, PMC2921829, and PMC3315840). Cross-validation and ROCs are typically performed on samples from the same cohort (although technically the training data and test data samples/observations do not overlap), and this can lead to biases and high variance in classification performances. Perhaps related to this point, only a miniscule proportion of all predictive models claiming high cross-validation accuracies or high AUCs have successfully translated into actual clinical practice; so this is basically why an independent validation dataset was sought after, and demonstrated upon, in our study.

Despite the higher value we place on using a truly independent validation dataset for evaluating classification performance, we’ve performed 10-fold cross-validation, per the reviewer’s request. Remarkably, 10-fold cross-validation resulted in an accuracy of 69.6% (please see **Supplementary Table 5**), which is nearly identical to the average classification accuracy of 69.7% achieved on our original training dataset. These

cross-validation results, as well as the performance on the validation dataset, demonstrates the power of integrating existing samples across various sources and health conditions to identify truly robust signals and insight. We discuss this analysis in lines 224–227 of the revised manuscript.

Lastly, we would like to know on what objective basis the reviewer considers an independent external validation dataset composed of 679 shotgun metagenome samples to be “small”. We would certainly like to know which studies successfully demonstrate robustness of a gut microbiome signal on an independent dataset of ~680 shotgun metagenome samples or more. Of course, we know that having more and more samples is always better; however, given the relative paucity of high-quality metagenome samples relative to 16S rRNA gene amplicon data, what we currently have is all that we could find in the literature at the time of our study. Also, we’ve sequenced our own cohort of patients with Rheumatoid Arthritis to add more credibility to our validation results.

Minor points

- Fig 2b, etc.: Why a PCA rather than an MDS, if the goal is to look for separation?

Authors’ response: We thank the reviewer for bringing this to our attention. Actually, in the legend of original **Fig. 2** (now **Fig. 1**), and in the **Results** section, as well as in the **Methods** section, we stated that **Figs. 2b** (now **Fig. 1c**) and **2C** (now **Fig. 1d**) are Principal Coordinates Analysis (PCoA) plots (PCoA is a version of MDS). To avoid possible confusion, we’ve changed the x-/y-axis labels of **Figs. 1c** and **1d** from “PC” to “PCo” (for Principal Coordinate).

- Fig 2b: Is the PERMANOVA significant when accounting for studies or indications? MiRKAT is a tool that can test for separation while accounting for covariates.

Authors’ response: We re-ran our PERMANOVA analysis in **Fig. 1c** to see whether the two populations (i.e., Healthy and Non-healthy) were significantly different from each other while adjusting for each sample’s study origin. More specifically, PERMANOVA on the Bray-Curtis distance matrix between samples (based on relative abundances of microbial species) was performed with 999 permutations (‘adonis2’ function in the R ‘Vegan’ package version 2.5.6), while random permutations were constrained within studies by using the ‘strata’ option. After accounting for studies, we still identified a significant difference between the distributions of these two groups ($R^2 = 0.017$, $P < 0.05$). We mention this addition to the PERMANOVA analysis in the legend of **Fig. 1** and in the **Methods** section (see lines 139–140 and 699) of the revised manuscript.

- Fig 4c: Does Shannon diversity separate cases and controls within individual studies? This is akin to what’s shown in Fig 5b, but I wonder if lumping all studies together makes Shannon look a worse classifier than it is.

Authors’ response: Great question. To address whether cases can be separated from controls within individual studies, we performed the following analysis:

- Analogous to what is shown in **Fig. 5** (wherein healthy was compared to each separate non-healthy phenotype within individual studies), we compared healthy against a general non-healthy phenotype, in which all disease samples were lumped together, when applicable. Importantly, comparisons were still made within individual studies. Our criterion for selecting which cohorts to perform intra-study, case-control comparisons was to have both groups to be composed of at least 10 samples, which we

deemed as reasonably sufficient sample size; in this case, there were only 12 studies (i.e., cohorts) that satisfied this sample size cut-off.

- We found that there were statistically significant differences in GMHI between cases and controls ($P < 0.05$, Mann-Whitney U test) in 6 of the 12 studies. In contrast, we found statistically significant differences in Shannon diversity, richness, and 80% abundance coverage between cases and controls in 2, 3, and 3 (of 12) studies, respectively.

In summary, Shannon diversity (defined at the species-level) does not robustly separate cases and controls within individual studies; in addition, GMHI outperforms the other metrics even when cases are grouped together. This analysis is mentioned in lines 470–476 of the revised manuscript and in **Supplementary Table 13**.

- Claims about "core" microbiome (l. 153) should be introduced by what's known in that literature.

Authors' response: This is a good catch. In order to be clear and objective when interpreting our findings, and to avoid using terminology that does not have a precise and/or universal definition, we felt it would be better to remove the clause "suggesting species-level members of a 'core' human gut microbiome". Moreover, this finding is of relatively small significance compared to our major findings.

- l. 204: A major weakness in the microbiome literature is poor characterization of effect sizes. With the large sample sizes used in this study, a difference in microbiome composition could easily be statistically significant without being scientifically interesting or clinically meaningful.

Authors' response: We emphatically agree with the reviewer's comment encouraging the use of effect size measures. Adding this information is good practice, especially in cases where differences (between two groups) are found to be statistically significant even though population means/medians look nearly identical. To accommodate, we've shown the Cliff's Delta effect size (d) in all figures wherein healthy and non-healthy groups are compared against each other, i.e., Figs. 2b, 3a–d, 6a, 6c, 6e, and 6g.

Of note, Cliff's delta is a non-parametric effect size measure that quantifies the amount of difference between two groups of observations. Briefly, the delta statistic measures how often one value in one distribution is higher than the values in the second distribution; it is a difference between probabilities (and not between means), and thus ranges from -1 to +1. Crucially, it does not require any assumptions about the shape or spread of the two distributions, and is therefore a very useful complementary analysis for the Mann-Whitney U test.

- l. 212: This approach seems very ad hoc. Why this particular formulation, and not one of dozens of others? What are the design criteria?

Authors' response: We kindly note that the motivation and design criteria underlying the formation of $\Psi_{M,H}$, its basic assumptions, and overview of its derivation, are now explained in lines 144–155 of the revised manuscript.

- I. 233: I think it's confusing to the reader to show a difficult equation and say to look in the Supplement for an explanation. If the GHMI is the centerpiece of the paper, its development, rationale, and form should be a key part of the text.

Authors' response: We emphatically agree to any suggestion regarding how the clarity of our work can be improved. Moreover, this particular area of focus was also mentioned by Reviewer #1. We revised the manuscript accordingly:

- To provide more clarity behind the *why* of our formula (as the reviewer pointed out), we made an effort to better explain the overall motivation of its conceptual design, as well as the logical rationale of each major step. Please see lines 144–155 and 260–277 of the revised manuscript.
- We've moved much of the technical jargon into the **Methods** section of the main text. Please see lines 707–800 of the revised manuscript.

- From Supp Table 3, it looks like the classifier's accuracy is not very sensitive to the choices of the threshold parameters. How sensitive are the results to the choice of health- and disease-associated species, and to the threshold parameters?

Authors' response: Indeed, as the reviewer points out in **Supplementary Table 3**, there is not much variance amongst the top average classification accuracies despite differences in the two prevalence thresholds θ_d and θ_r (on the other hand, the identities of the Health-prevalent and Health-scarce species, which are determined by the two thresholds as noted above, seem to widely vary depending on the choice of thresholds.) This is not a fault or a disadvantage per se; however, we certainly do see the value in doing a basic sensitivity analysis of classification performance (i.e., average classification accuracy) with respect to one threshold, while keeping the other constant. Our findings can be summarized as follows:

- Classification performance and prevalence difference generally portray an inverse correlation:

- Classification performance displays a very weak but positive correlation for smaller values of prevalence fold-change, but then follows an inverse correlation for higher values of prevalence fold-change:

This analysis is mentioned in lines 229–231 in the revised manuscript and in **Supplementary Fig. 2**. In addition, as we clarified above that the choice of thresholds directly determines the identities of Health-prevalent and Health-scarce species, we decided to omit a sensitivity analysis between classification performance and the two sets of species as we feel it would not be practically meaningful to do so.

- I. 295: I found this confusing: doesn't the GHMI have two thresholds and ~100 selections of species as health- or disease-associated? Why is that not "arbitrary"?

Authors' response: Here, the reviewer is referring to the sentence: “Importantly, our metric requires very little parameter-tuning and foregoes the use of arbitrary thresholds, e.g., ‘low’ or ‘high’ alpha-diversity”. When we wrote the word “arbitrary” here, we were referring to loosely concocted narratives (mainly put forth by lay media sources) along the lines of: “high diversity being good for health, and low diversity being a sign of bad health”. Clearly, this is an oversimplification of reported statistically significant associations between gut diversity and health. We are not disputing these findings; rather, we ask what precisely is “low”?; And what is meant by “high”?; and can we build a classifier? In this sense, we simply believe that having more quantitative precision is needed in today’s clinically-driven microbiome research. And in order to avoid possible confusion, we’ve changed the term “arbitrary thresholds” in our original draft to “qualitative assessments”. Please see line 288 of the revised manuscript.

As explained in our response to a previous comment, each pair of thresholds (that are first comprehensively screened) leads to its respective set of Health-prevalent (M_H) and Health-scarce species (M_N); these species are not chosen arbitrarily. Furthermore, our choice to simultaneously test two types of prevalence thresholds, rather than one, was in order to increase our confidence in the robustness of M_H and M_N , as well as to overcome limitations/biases that can occur from using only one type of threshold. Now, if the reviewer is wondering why not three thresholds or more, that’s simply because we didn’t want to constrain our selection of features too rigidly.

- I. 300: The use of the word "potential" implies that the microbiome is causative in disease. This is generally not known.

Authors' response: Our use of the word "potential" characterizes the functional possibilities from only genetic content. Although we do not have gene expression data, protein abundance data, or data from metabolic assays for the detection of a biochemical function actually occurring, gene-level analyses can still provide significant clues into the possibilities that can occur. Hence, this is why we deliberately chose our wording (in the main text) as follows: "...potential metabolic functions specific to Health-prevalent species.", "...differences in functional potential between...", and "...analyzing both taxonomic composition and functional potential are both...". Furthermore, this is common practice in genome/meta-genome analysis papers, as provided in the references below:

- <https://www.nature.com/articles/ismej20143>
 - "To validate the inferred functions determined by PICRUSt, we performed WMS sequencing – the conventional means of assessing microbiome functional potential – on a subset of banked stool samples from anti-TNF- α -treated mice (n=6)"
- [https://www.cell.com/trends/microbiology/fulltext/S0966-842X\(17\)30251-2](https://www.cell.com/trends/microbiology/fulltext/S0966-842X(17)30251-2)
 - "Recent large-scale metagenomic studies have provided insights into its structure and functional potential."
- <https://www.nature.com/articles/s41564-017-0084-4>
 - "The present study has provided an overview of the fecal metatranscriptome in a prospective, large-scale cohort of elderly males; identified core and variably transcribed pathways; delineated how these differ from metagenomic functional potential; and ascribed them to specific contributing organisms."
 - "This indicates that, as in a single organism's genome, only a subset of fecal functional potential is active under the circumstances captured by a typical sample."
- <https://www.nature.com/articles/ismej201041>
 - "...but no comprehensive study has yet addressed their composition and functional potential in permafrost."

In addition, we respectfully clarify that we were in no way shape or form implying that, due to these biochemical functions, "the microbiome is causative in disease". More specifically, we wrote that these metabolic functions are specific to the Health-prevalent species, and not to health or to any disease.

- I. 312: Why was this one result on sphingolipids the one discussed? It feels like cherry-picking.

Authors' response: We thank the reviewer for this concern. Actually, the analysis pertaining to this comment was entirely modified in light of an excellent suggestion from Reviewer #1. We describe our modified strategy of using a metagenomic binning approach to identify enriched functional genes and modules pertaining to the Health-prevalent species in the **Methods** section of the revised manuscript (see lines 802–815), and show the results of our new analysis in **Supplementary Fig. 5** and **Supplementary Tables 7** and **8**. This time, we avoided any seemingly biased analysis on a particular feature of interest.

- I. 344: Is 71% good? How much better is it than alpha diversity? How much worse is it than a random forest classifier, which is inscrutable but usually a better predictor?

Authors' response: Please see our response to both questions above.

- I. 375 ff. This should be the first Results section! If the claim is that alpha diversity is insufficient, the first Results should substantiate that claim. Many, many studies have claimed that alpha diversity is associated with health. This isn't to say they are right and you are wrong, but it does mean that, if you want to claim they are all wrong, you need to back up the claim!

Scott Olesen, PhD
OpenBiome

Authors' response: We thank the reviewer for this suggestion. As mentioned above, we have done some major restructuring of the manuscript to get the aforementioned points across more directly. In regards to the narrative structure of our manuscript, we feel that the following outline—focused around the main figures and tables—makes the most logical sense:

1. We introduce our multi-dataset integration pipeline (**Fig. 1a**) and some basic properties of the training data (i.e., the meta-dataset of 4,347 stool metagenome samples, **Table 1**, and **Fig. 1b–d**), upon which our index was built and tested.
2. We present the Health-prevalent and Health-scarce species that were found during the derivation of the GMHI formula (**Table 2**).
3. By showing that there is a moderately positive and significant correlation between GMHI and high-density lipoprotein cholesterol (**Fig. 2**), we were able to: i) associate GMHI with a key parameter of cardiovascular health; and ii) better establish the clinical relevance of GMHI.
4. Using the vast collection of samples in our training data, we present the classification performances of GMHI, Shannon diversity, 80% abundance coverage, and species richness all together (**Fig. 3**). Thus, we have placed our most important figures as early as we could, as suggested by the reviewer.
5. Building off the results in the previous figure, we take a closer look at changes in group proportions across the full range of GMHI values (**Fig. 4a**); and a direct head-to-head comparison of GMHI vs. Shannon diversity (**Fig. 4b**) to illustrate which metric better distinguishes healthy and non-healthy groups.
6. To identify the best metric after removing inter-study variation (which is a major source of batch effects), we perform case-control comparisons within individual studies to see which metric is the most robust and consistent (**Fig. 5**).
7. On an independent external validation dataset of 679 samples, we show that GMHI best stratifies healthy and non-healthy groups compared to the other three metrics (**Fig. 6**).

We respectfully clarify that never in our manuscript did we question or doubt the statistical association between alpha diversity and health. This has been reported in the scientific literature numerous times (interestingly enough, the effect sizes are rarely shown with the corresponding *P*-value). However, we do question its utility as a *classifier* and/or predictor of health. The main message of our manuscript is to provide a better quantitative marker by thinking a little differently. Our data throughout the manuscript clearly show that GMHI is a more accurate, robust, and clinically-relevant classifier for distinguishing the presence/absence of diagnosed disease.

Reviewers' Comments:

Reviewer #1:

Remarks to the Author:

I thank the authors for their thorough reply to my comments. At this point I feel that my concerns have been addressed. The authors now provide a more nuanced description of the index and associated biases and specifically note that it rather quantifies the absence of a diagnosis/observed disease rather than a healthy state. They also repeated their analyses on all major taxonomic ranks and provide a functional analysis that now uses the metagenomic data rather than relying on reference strains. I recommend some minor additions to connect the biomarker and functional metagenomics results.

The authors now found that HDLC is associated with a higher health index. At the same time they observe that the microbial genes most associated with the predicted index are either involved in amino acid or starch metabolism. This seems to indicate that the index is tightly connected to the metabolic state of the host. I wasn't surprised to see HDLC pop up here since host cholesterol levels do interact with the gut microbiome through bile acids and their receptors such as TGR5 and FXR. Increased starch breakdown by the gut microbiota has been associated with higher weight and body fat in mice (<https://doi.org/10.1038/nature05414> and <https://doi.org/10.1038/s41564-019-0569-4>). However, the authors observe the opposite trend from their functional metagenomic analysis. Obesity is a risk factor or comorbidity for many diseases which might explain the observed results. Thus, I feel that summarizing those results in a few phrases in the discussion will help to form a hypothesis which physiological feature of the host is actually captured by the derived index.

Christian Diener

Institute for Systems Biology

Reviewer #2:

Remarks to the Author:

First and foremost, I appreciate the authors' patience and care in responding to my comments. This paper clearly involved a lot of work and careful thought, and that should be the first line of any review. In such a technical and data-rich work, there was plenty of opportunity for us to misunderstand one another!

Overall, I think the manuscript is much improved. My major concerns about scope and framing were resolved; the authors should consider my remaining remarks on those topics as hopefully-constructive suggestions.

I do have some remaining reservations about some points of terminology and methodology, marked as "strong suggestions".

STRONG SUGGESTIONS

Meaning of θ_d

On a second reading, it struck me that θ_d actually sets a minimum prevalence. E.g., $\theta_d = 10\%$ means that a health-prevalent species, say, can be 0% prevalent in the cases but must be at least 10% prevalent in controls.

Define instead θ_{min} , which is the minimum prevalence of a health-prevalent species in controls (or -scarce species in cases). Then $\theta_d = \theta_{min} * (1 - \theta_f)$, so it seems these two parameterizations are equivalent?

My guess is the θ_d is more important for setting a minimum prevalence, and θ_f mostly controls the difference between the prevalences in the cases and controls. If that's true, then θ_{min} & θ_f might be a more interpretable parameterization than θ_d & θ_f .

Characterization of "performance"

The quantitative comparison of the index against richness, diversity, number of healthy species present, and random forest classifiers, as well as comparison of the index trained on species versus other taxonomic levels, greatly enhanced the paper.

My remaining concern here is about the use of the word "performance". I think the authors are in the right for saying that the species-trained index performs better than the phylum-trained index, because "performance" is equated with accuracy. But when I hear "performance" it makes me think also about model fit. Insofar as the species-trained model has many more degrees of freedom (i.e., each taxon is one of "good", "bad", or neither) than the phylum-trained model, it is no surprise that the species-trained model performs better than the phylum-trained model. Does it perform better given the greater number of degrees of freedom?

Because I do not believe the authors want to get into model fit and numbers of parameters, I suggest replacing "performance" with "accuracy".

OTHER SUGGESTIONS / COMMENTS

Overall framing

I appreciate the authors' changes to the text, which I think much more clearly motivates the work: this index is designed to be accurate (in a way that Shannon's diversity isn't) and robust (in a way that random forests aren't). It is a proof-of-concept for a clinically translatable tool.

I still think that the Introduction and Discussion could more actively engage with what seem like the two major theoretical underpinnings of the index's design: first, that "balance" between health-prevalent (I'll just say "good") and health-scarce (I'll just say "bad") bacteria reflects health status better than diversity; second, that the same "balance" applies across multiple disease states. These ideas are not unsupported by the literature, but they are also not uncontroversial. My suggestion is to make it clear in the Introduction that these are the two guiding inspirations for the formulation of the index, and rather than arguing that they are definitely true, saying that the intent was merely to determine if they were useful when designing a new index.

As an aside: I have a personal quibble with the word "balance" in microbiome literature. Normally "balance" is a good thing. The typical example is a "balance" between pro- and anti-inflammatory bacteria: you don't want a wholly pro- or anti-inflammatory state, somewhere in the middle is best. But interesting, in this manuscript, "balance" is actually not good: you want to be all the way on the "good" bacteria side. So maybe "balance" isn't the right word? Or maybe it is, and the other use of "balance" is bad? Or is this all just too much semantics?

And even if your motivation is translational, scientifically-motivated readers like me will definitely view the results through the lens of a hypothesis: it's like you hypothesized that "balance" is more important than diversity and that health vs. disease "looks the same" across diseases, and you found that that idea works when classifying bacteria. So reading this one way, it feels like a potentially profound statement about ecology.

Explanation of the functional form of ψ

This manuscript makes a clear argument for why the index h should be a ratio, and the log seems to make good sense given the data. However, I am still a bit puzzled by certain points around the functional form of ψ . (As a note, these questions might have arisen because I didn't understand what was already well-written and -explained!)

- Why is the constant c_{MH} needed? How was its value determined?
- Why are both richness RMH_i and abundance n_{ji} needed? Why are they multiplied together? What happens if you just used one or the other?
- Why not a simpler form, like the average abundance (sum over healthy species j of n_{ji} , divided by sum of n_{ji} for all species j)? The Methods makes a good argument for why you would want to use a geometric mean, but what happens if you don't?
- The presence of $n \log n$ in ψ is interesting, because it means that ψ is something about the Shannon diversity of the sample, but only when considering particular species. I'm not sure exactly what implications that has.

Small points

- l. 36 "remarkable": Quantify rather than claim "remarkable"?
- l. 95 "age": Are results confounded by age? In other words, what if the index merely predicts age, which in turn predicts disease state?
- l. 110 "viruses": An interesting feature of the index is that it need not be specific to bacteria; one could imagine applying it to health-prevalent vs. -scarce viruses or fungi.
- l. 124 "homogeneously dispersed": Statistical test to support this claim?
- Figure 1 "outliers": How were outliers defined?
- l. 173 "ensemble-averaged": I think this is true, but it required me thinking about it fairly carefully after two careful readings of this manuscript. So it might be something to explain a bit more.
- l. 210: I think this is technically the definition for `_balanced_` accuracy, rather than just "accuracy".
- l. 222: Why does phylum fitting have a classification accuracy less than 50%? Is this because balanced accuracy is the metric? If this were plain old accuracy, I would expect a completely random classifier to have 50% accuracy.
- l. 222: I actually think it's pretty interesting that the family-level index performs just about as well as the species-level index. Given the greater number of parameters at the species level, it seems like using families or something similar might be more "robust", especially since species names change?
- l. 234: It seems to me that it should follow from the definition of h and the training of the classifier that $h > 0$ implies greater relative abundance of health-prevalent species. Does that logic only hold if increased prevalence is correlated with increased abundance? My point is that this might be more of a "sanity check" than a "result".
- l. 262: Prevalence is an important part of the algorithm, but I never saw (I might have missed it!) how "present" was defined. Does just 1 read from a species make the species "present"? Does changing that threshold change the behavior of the model? If tweaking that threshold makes accuracy change, does that mean the current model is overfitted, or that it's taking advantage of deep sequencing?
- I found it difficult to interpret the cholesterol results. Is the fact that LDL level is correlated with the index mean something about biology (LDL is closely associated with the microbiome?) or about the metric (LDL is controlled by "balance" in the microbiome?) or about the data (only LDL has a strong enough signal to be detected?)?
- l. 304: Estimates of Spearman's ρ should have confidence intervals.
- Figure 2: I think this is the same data in the two plots? Is there a way to rotate or align them so I can see that (b) is the marginal distribution of the points in (a)?
- l. 323: Cliff's delta is a great metric here! I think this makes a big improvement to the interpretability. But I do think Cliff's delta is likely unfamiliar to most readers; it could be good to include a one-sentence explanation of what it means (+1 means 100% of A points are greater

than B points; 0 means 50%; -1 means 0%?).

- l. 333: Are there other metrics other than richness, Shannon, and 80% coverage considered?
- Figure 3: Does the index's improved performance in any particular indication "drive" its improved performance? Like, if you removed disease X from the training set, would the index's predictive ability decline more than you would expect simply because you shrunk the training set?
- l. 369-371: I didn't understand this sentence
- l. 401 "the most": Implies it is more than any predictor of health status, when in fact the claim is just about the other 3 metrics the index was compared to
- l. 445 "in 6 of 12": multiple hypothesis correction required?
- l. 473: I found it very surprising that random forest, which is a highly flexible algorithm with access to the same information as the index (e.g., the species names are not "blinded" in the way they are with richness or other diversity metric), performed worse than the index. I think this has some implications about how random forest, essentially a set of decision trees, doesn't have the same format as the rational equation approach of the index, and that's what gave improved performance to the index over random forests?
- l. 478: Rather than classification accuracy, the random forest's out of bag (OOB) error should be reported, since that is the value that accounts for overfitting.
- l. 485: effect size?
- l. 558 "clinical need": I appreciate that the authors, who are actually doctors, know better than I do about what the clinical need is! I would, however, appreciate a citation to that effect for the benefit of other readers.
- l. 630: This could use a citation too. I'm sure there are many, but this one comes to mind:
<https://pubmed.ncbi.nlm.nih.gov/31683111/>
- l. 636-641: I think this is a very clear articulation of the paper's motivation, and I would have preferred to read it in the Introduction!

REVIEWER COMMENTS

Reviewer #1 (Remarks to the Author):

I thank the authors for their thorough reply to my comments. At this point I feel that my concerns have been addressed. The authors now provide a more nuanced description of the index and associated biases and specifically note that it rather quantifies the absence of a diagnosis/observed disease rather than a healthy state. They also repeated their analyses on all major taxonomic ranks and provide a functional analysis that now uses the metagenomic data rather than relying on reference strains. I recommend some minor additions to connect the biomarker and functional metagenomics results.

Authors' response: We very much thank the reviewer for all constructive feedback for our manuscript. We are pleased to hear that the original concerns have been adequately addressed.

The authors now found that HDLC is associated with a higher health index. At the same time they observe that the microbial genes most associated with the predicted index are either involved in amino acid or starch metabolism. This seems to indicate that the index is tightly connected to the metabolic state of the host. I wasn't surprised to see HDLC pop up here since host cholesterol levels do interact with the gut microbiome through bile acids and their receptors such as TGR5 and FXR. Increased starch breakdown by the gut microbiota has been associated with higher weight and body fat in mice (<https://doi.org/10.1038/nature05414> and <https://doi.org/10.1038/s41564-019-0569-4>). However, the authors observe the opposite trend from their functional metagenomic analysis. Obesity is a risk factor or comorbidity for many diseases which might explain the observed results. Thus, I feel that summarizing those results in a few phrases in the discussion will help to form a hypothesis which physiological feature of the host is actually captured by the derived index.

Christian Diener
Institute for Systems Biology

Authors' response: The reviewer brings to our attention an interesting question that we ourselves had overlooked: What is the tripartite relationship amongst (host) metabolism, gut microbiome metabolic function, and our gut health index? Certainly, this appeal is driven by our findings that: i) GMHI is correlated with serum high-density lipoprotein cholesterol (HDLC) concentrations; and ii) genes of the health-prevalent species were found to be enriched (in the metagenomes) for enzymes contributing to the metabolism of simple sugars (fructose, xylulose), glucose compounds (trehalose and starch), and more complex biochemical structures (xylan and Lipid A). Certainly, there is a plethora of literature evidence demonstrating a mechanistic link between the gut microbiome, host metabolism, and metabolic syndrome (such as obesity), including the famous Turnbaugh *et al.* Nature 2006 study mentioned by the reviewer.

Despite this interesting link, we've elected to exercise a bit of caution in formulating certain hypotheses without a clear, unequivocal understanding behind the observations mentioned above. For example, high HDLC levels in serum may be a reflection of various factors (e.g., genetics, dietary habits, lifestyle) other than gut microbiome; and enriched genes within the microbiome (which have been shown to be sensitive to host diet) still reflect functional potential, and not active function. Therefore, as tempting as it may be to entertain

possible hypotheses regarding all the reviewer's aforementioned points, we prefer to be more on the conservative end and refrain from delving too far into statistical associations. We sincerely hope our viewpoint sounds fair and reasonable.

With that being said, we certainly wanted to respect the reviewer's suggestion to have a bit more reflection on the implications of our findings. Rather than speculating on complex relationships, we decided to further comment on our HDLC results only. In a very recent study published in *Cell Host & Microbe* (2020) <https://pubmed.ncbi.nlm.nih.gov/32544460/>, Kenny *et al.* found that host serum cholesterol levels (LDLC and HDLC) are influenced by gut bacteria through *ismA*, which is a microbial cholesterol dehydrogenase. By integrating paired metagenomics and metabolomics data from geographically diverse human cohorts, the authors identified a group of *ismA* genes in uncultured gut microorganisms, and found that these genes encode enzymes that convert cholesterol to coprostanol. Moreover, individuals harboring gut microbes with *ismA* genes were found to have significantly lower serum cholesterol levels (LDLC and HDLC). These findings are relevant to our own results regarding a gut microbiome and HDLC connection.

We've introduced this point in lines 317–322 of the revised manuscript, as follows: "In relevance to this point, a recent study by Kenny *et al.* showed that cholesterol metabolism by gut microbes may impact cardiovascular health by influencing serum cholesterol concentrations. Specifically, the investigators discovered that gut microbes can metabolize cholesterol into coprostanol through enzymes encoded by *ismA* (a cholesterol dehydrogenase); and that individuals harboring gut microbes with *ismA* genes have significantly lower serum total cholesterol levels²⁷."

Also, we've slightly modified our wording in line 303:

- (previous) "In order to validate whether GMHI can indeed quantify certain aspects of health,"
- (now) "In order to validate whether GMHI can indeed capture certain physiological features of health,"

Reviewer #2 (Remarks to the Author):

First and foremost, I appreciate the authors' patience and care in responding to my comments. This paper clearly involved a lot of work and careful thought, and that should be the first line of any review. In such a technical and data-rich work, there was plenty of opportunity for us to misunderstand one another!

Overall, I think the manuscript is much improved. My major concerns about scope and framing were resolved; the authors should consider my remaining remarks on those topics as hopefully-constructive suggestions.

I do have some remaining reservations about some points of terminology and methodology, marked as "strong suggestions".

Authors' response: We would like to extend our gratitude to the reviewer, as all comments were indispensable for improving the clarity and readability of our manuscript. Moreover, we are much pleased to hear that the original concerns have been resolved, and we welcome the opportunity to further improve our work, as we demonstrate below.

STRONG SUGGESTIONS

Meaning of θ_d

On a second reading, it struck me that θ_d actually sets a minimum prevalence. E.g., $\theta_d = 10\%$ means that a health-prevalent species, say, can be 0% prevalent in the cases but must be at least 10% prevalent in controls.

Define instead θ_{\min} , which is the minimum prevalence of a health-prevalent species in controls (or -scarce species in cases). Then $\theta_d = \theta_{\min} * (1 - \theta_f)$, so it seems these two parameterizations are equivalent? My guess is the θ_d is more important for setting a minimum prevalence, and θ_f mostly controls the difference between the prevalences in the cases and controls. If that's true, then θ_{\min} & θ_f might be a more interpretable parameterization than θ_d & θ_f .

Authors' response: Unfortunately, we are a bit puzzled by the premise of this suggestion. The minimum prevalence of a particular species (in either group) is not particularly relevant to our overarching goal of identifying the Health-prevalent (i.e., higher frequency in Healthy) and Health-scarce (i.e., lower frequency in Healthy) species. In fact, it was never considered. Let us try to clarify, with the hope of reaching a mutual understanding:

- To be clear, θ_f and θ_d have been defined as minimum thresholds for prevalence fold change (p_H/p_N) and prevalence difference ($p_H - p_N$), respectively.
- Yes, θ_d *can* suggest the minimum prevalence, but it would make more sense to just find what that prevalence is directly from the microbiome samples. However, these values serve really no practical purpose in our methodology.
- In regards to ease of interpretation behind our parameters: As authors of this manuscript, we believe that our prevalence thresholds (θ_f and θ_d)—as currently defined—make the most intuitive sense when comparing the magnitudes of the two prevalences. Specifically, all we're doing is determining the difference (subtraction) and fold-change (division) in prevalences to find microbes with considerable effect size between Healthy and Non-healthy.
- Detailed information on our prevalence-based strategy to identify microbial species associated with healthy human gut microbiomes, along with the motivation underlying our two thresholds, can be found in lines 155–188. We kindly suggest another read to confirm that both parties are on the same page.
 - Perhaps we may clarify our text a bit further? Or perhaps the reviewer can word the question a bit differently? (we genuinely would like to understand where the reviewer is coming from.) Otherwise, perhaps we've clarified enough in the points above?

And last but not least, the proposed situation of 0% prevalence doesn't occur in any of the 313 species used in our study for GMHI formulation. If so, the fold-change could not have been derived.

Characterization of “performance”

The quantitative comparison of the index against richness, diversity, number of healthy species present, and random forest classifiers, as well as comparison of the index trained on species versus other taxonomic levels, greatly enhanced the paper.

Authors’ response: We are pleased to hear this!

My remaining concern here is about the use of the word “performance”. I think the authors are in the right for saying that the species-trained index performs better than the phylum-trained index, because “performance” is equated with accuracy. But when I hear “performance” it makes me think also about model fit. Insofar as the species-trained model has many more degrees of freedom (i.e., each taxon is one of “good”, “bad”, or neither) than the phylum-trained model, it is no surprise that the species-trained model performs better than the phylum-trained model. Does it perform better given the greater number of degrees of freedom?

Authors’ response: This is an interesting topic to ponder upon. We agree that a larger feature set can indeed lead to a more accurately tuned model more often than not; however, there can be considerable exceptions involving the following:

- Having more features does not necessarily result in higher signal-to-noise. The possibility that more noise comes with an expanded feature-set is certainly not negligible.
- Many of the species are correlated with one another, or at least much more "so than amongst individual phyla. A consequence of having a lot of features demonstrating high(er) covariance is high(er) signal redundancy, i.e., highly repetitive patterns.
- Species’ abundances are, in theory, a subset of phyla’ abundances; in other words, higher abundances of Bacteroidetes should lead to higher abundances of at least some species within this phylum. Thus, a non-trivial portion of the variance captured at the phylum-level can also be captured at the species-level, and we question the practical significance of further head-to-head comparisons.
- Most importantly, a GMHI built upon MetaCyc pathways (# of pathways = 405) resulted in a much worse classifier performance (57.3% balanced accuracy) than an index from species (# of species = 313) information (69.7% balanced accuracy). Therefore, merely having greater degrees of freedom (i.e., higher number of features) is not necessarily predictive of better classification accuracy.

Our point is that, prior to making general assumptions, careful consideration of the properties of the features should be made. For us, we certainly did not sense any guarantee that species were going to give better classification accuracy than phyla.

To answer the question of “Does it perform better given the greater number of degrees of freedom?”, a Monte carlo random sampling approach, while keeping the number of selected features consistent for both taxonomic clades, would be one way (of many) to test this hypothesis. However, we strongly feel that this analysis (and its results) is far outside the main message of our manuscript. We would like to firmly establish the proof-of-concept first, and then, in future studies, optimize the model, learn its advantages and weaknesses in great detail, etc.

Because I do not believe the authors want to get into model fit and numbers of parameters, I suggest replacing “performance” with “accuracy”.

Authors’ response: We do recognize the value of being clear on the semantics, and thus we agree with the reviewer’s viewpoint. Please see all highlighted areas throughout the revised manuscript (lines 83, 163, 223, 233, 486, 587, and 599) where we’ve replaced “classification performance” with “classification accuracy”.

OTHER SUGGESTIONS / COMMENTS

Overall framing

I appreciate the authors’ changes to the text, which I think much more clearly motivates the work: this index is designed to be accurate (in a way that Shannon’s diversity isn’t) and robust (in a way that random forests aren’t). It is a proof-of-concept for a clinically translatable tool.

Authors’ response: We are pleased to hear this!

I still think that the Introduction and Discussion could more actively engage with what seem like the two major theoretical underpinnings of the index’s design: first, that “balance” between health-prevalent (I’ll just say “good”) and health-scarce (I’ll just say “bad”) bacteria reflects health status better than diversity; second, that the same “balance” applies across multiple disease states. These ideas are not unsupported by the literature, but they are also not uncontroversial. My suggestion is to make it clear in the Introduction that these are the two guiding inspirations for the formulation of the index, and rather than arguing that they are definitely true, saying that the intent was merely to determine if they were useful when designing a new index.

Authors’ response: We very much thank the reviewer for this comment. In fact, the whole subsection in the Results section titled “Motivation underlying the design of a gut microbiome-based health index” is dedicated to articulate the underlying rationale behind GMHI. However, we do agree with the reviewer that we can be more engaging and direct. (we kindly clarify that we’ve toned down our narrative against alpha-diversity as a health measure.) To address the reviewer’s comment without writing entirely new subsections, we’ve added the following brief statements in the main text:

- lines 75–77: “GMHI determines the likelihood of having a disease, irrespective of the diagnosis. This is done so by comparing the relative abundances of two sets of microbial species associated with good and adverse health conditions, ...”
- line 79: “...various disease states.”
- lines 577–580: “In this study, we provide a simple and biologically-interpretable measure to quantify the degree of presence/absence of diagnosed disease from a gut microbiome (stool metagenome) sample. Specifically, i) we envisioned that the most intuitive way to determine how closely one’s microbiome resembles that of a healthy (or non-healthy) population is to compare the relative abundances of two sets of microbes; ... ”

As an aside: I have a personal quibble with the word “balance” in microbiome literature. Normally “balance” is a good thing. The typical example is a “balance” between pro- and anti-inflammatory bacteria: you don’t want a wholly pro- or anti-inflammatory state, somewhere in the middle is best. But interesting, in this manuscript, “balance” is actually not good: you want to be all the way on the “good” bacteria side. So maybe “balance” isn’t the right word? Or maybe it is, and the other use of “balance” is bad? Or is this all just too much semantics?

Authors’ response: As previously mentioned, we strive to be crystal clear on the semantics. If one working definition of “balance” is to have near evenly distributed weights or abundances among Health-prevalent and Health-scarce species, then indeed, “balance” does not lead to “healthy” (but to be clear, we did not suggest this anywhere in the text); rather, it is desirable to have more Health-prevalent species relative to Health-scarce species to be seen as absent of diagnosed disease. To prevent any possible confusion that may arise from our use of “balance”, we’ve modified the main text accordingly:

- Introduction, lines 75–77:
 - (previous) “GMHI is designed to evaluate the balance between two sets of microbial species associated with good and adverse health conditions,”
 - (now) “GMHI determines the likelihood of having a disease, irrespective of the diagnosis. This is done so by comparing the relative abundances of two sets of microbial species associated with good and adverse health conditions,”
- Motivation underlying the design of a gut microbiome-based health index, lines 158–159:
 - (previous) “Therefore, we propose an index in the form of a rational equation between two sets of microbial species:”
 - (now) “Therefore, we propose an index in the form of a rational equation (and thereby yielding a dimensionless quantity) between two sets of microbial species:”
- Discussion, lines 580:
 - (previous) “we envisioned that the most intuitive way to determine how closely one’s microbiome resembles that of a healthy (or non-healthy) population is to quantify the balance between two sets of microbes;”
 - (now) “we envisioned that the most intuitive way to determine how closely one’s microbiome resembles that of a healthy (or non-healthy) population is to compare the relative abundances of two sets of microbes;”
- Discussion, lines 585–587:
 - (previous) “GMHI is a biologically-interpretable, quantitative metric formulated based on the balance between the collective abundances of Health-prevalent and of Health-scarce species.”
 - (now) “GMHI is a biologically-interpretable, quantitative metric formulated based on a ratio between the collective abundances of Health-prevalent and of Health-scarce species.”

Two other areas were left unchanged (lines 157, 207, and 292), as we felt that the word “balance” was actually used appropriately in these cases. We hope these modifications address the reviewer’s concern.

And even if your motivation is translational, scientifically-motivated readers like me will definitely view the results through the lens of a hypothesis: it’s like you hypothesized that “balance” is more important than

diversity and that health vs. disease “looks the same” across diseases, and you found that that idea works when classifying bacteria. So reading this one way, it feels like a potentially profound statement about ecology.

Authors’ response: It’s definitely thought-provoking that the same data can be viewed and interpreted from so many different angles! Certainly, these points (as well as many others throughout the manuscript) can be analyzed much more meticulously in follow-up analyses, especially after seeing how the scientific and medical communities judge our study once out in print. We hope to keep in touch after the dust settles.

Explanation of the functional form of ψ

This manuscript makes a clear argument for why the index h should be a ratio, and the log seems to make good sense given the data. However, I am still a bit puzzled by certain point around the functional form of ψ . (As a note, these questions might have arisen because I didn’t understand what was already well-written and -explained!)

- Why is the constant c_{MH} needed? How was its value determined?

Authors’ response: C_{MH} and C_{MN} were originally used to approximate the mathematical relationship between

ψ and $\frac{R_{MH,i}}{|M_H|} \sum_{j \in I_{MH}} |n_{j,i} \ln(n_{j,i})|$. The exact values of C_{MH} and C_{MN} were never required, as C_{MH}/C_{MN} was modeled to be close enough to 1 for simplicity of the analysis. Although the individual values of C_{MH} and C_{MN} were not necessary to conduct our analysis, we cannot ignore the possibility that these two parameters may introduce confusion (as the reviewer points out). In the interest of clarity, we’ve removed all mentioning of C_{MH} and C_{MN}

throughout the manuscript, and defined ψ as $\psi_{MH,i} = \frac{R_{MH,i}}{|M_H|} \sum_{j \in I_{MH}} |n_{j,i} \ln(n_{j,i})|$ instead. Notably, this change does not affect any of the results in our study.

- Why are both richness $RMHi$ and abundance n_{ji} needed? Why are they multiplied together? What happens if you just used one or the other?

Authors’ response: After identifying the Health-prevalent and Health-scarce species, we needed a mathematical formula to best approximate the ‘collective abundance’ ψ . Of course, we can consider many different microbial/ecological properties, which can then be formulated into almost infinite combinations of linear or non-linear equations. As detailed in lines 738–754, we considered both species’ richness and also their (geometric) mean relative abundances, as it seemed to make just plain sense to us (we believe the authors should have the freedom to derive any equation they want, as long as there are no inherent flaws in the mathematical presentation or underlying biological assumptions).

Given that we’ve defined ψ to be positively correlated with richness and with n_{ji} , a neat trick for using these variables simultaneously is to simply multiply them (we expect future versions of the model to be a bit more complicated once certain constraints are added to the formula). Here, the product is strongly preferred over the summation due to the wide difference in scales.

In regards to what would happen if we were to just use one or the other, this was partially addressed in the previous review round during our analysis with the count of Health-prevalent species. With respect, we would like to decline carrying out additional analyses to deconstruct and understand all the intricacies behind our equation, as we feel this may distract readers from the main storyline of our work. We are definitely open to follow-up analyses involving tinkering this and that to gain insight into which parameters are most impactful; and which can be ignored or treated approximately.

- Why not a simpler form, like the average abundance (sum over healthy species j of n_{ji} , divided by sum of n_{ji} for all species j)? The Methods makes a good argument for why you would want to use a geometric mean, but what happens if you don't?

Authors' response: We thank the reviewer for these questions. Although our equation for ψ may look a little daunting at first, ψ is just built on the richness and mean abundances of a particular set of microbial species. Thus, despite the seemingly complicated structure of the equation, the underlying assumptions are rather straightforward (in our view), and more importantly, the equation captures the parameters that we deem as essential for characterizing the 'collective abundance' of a group of species.

Indeed, there is a lot of interesting insight that can be learned if we were to tinker here and there. As stated in our previous response, all the intricacies of our equation need not be completely resolved in this original work. As a somewhat related example, consider the popularity of deep-learning-based, neural network algorithms, of which the physics of how these approaches actually make decisions are currently still being worked out by labs around the world.

- The presence of $n \log n$ in ψ is interesting, because it means that ψ is something about the Shannon diversity of the sample, but only when considering particular species. I'm not sure exactly what implications that has.

Authors' response: We appreciate this observation about our formula. To clarify, we were not specifically looking to use the Shannon diversity while constructing our formula; rather, the equation for Shannon diversity just so happened to closely resemble the mean abundance of the species set of interest (we briefly mention in lines 748–753 the reason for using the geometric mean over the arithmetic mean). Although we agree that it is an interesting coincidence, we feel that there will not be a significant improvement to the manuscript from further speculation.

Small points

Authors' response: In regards to all points below, we tremendously appreciate the meticulous effort and dedication put forth by the reviewer to help improve our manuscript!

- I. 36 "remarkable": Quantify rather than claim "remarkable"?

Authors' response: Thank you for pointing this out. We agree that it's best to back-up subjective claims with real numbers! We've changed the text accordingly:

- line 36
 - (previous) “... resulted in remarkable reproducibility in distinguishing ... ”
 - (now) “... resulted in a balanced accuracy of 73.7% in distinguishing ... ”

- I. 95 “age”: Are results confounded by age? In other words, what if the index merely predicts age, which in turn predicts disease state?

Authors’ response: Thank you for bringing this important matter to our attention. To address whether our results are confounded by age, we sought the correlation between this potential covariate (as reported in the original studies) and GMHI. Of note, 1,997 of 4,347 samples in our training set report the subjects’ age.

As shown above (left), our analysis shows that age and GMHI hardly demonstrate any meaningful relationship, i.e., Spearman’s $\rho = 0.034$ ($P = 0.13$). In addition (right), no significant difference in age was observed between subjects with positive GMHI ($n = 762$) and those with negative GMHI ($n = 1,208$) (Mann-Whitney U test; $P = 0.17$). In summary, we conclude that age is not a considerable confounder of our study’s results. We’ve mentioned this in lines 326–327, as follows: “... and even age ($\rho = 0.03$, 95% CI: [-0.01, 0.08]) were noted to have only weak or no meaningful correlations with GMHI.”

- I. 110 “viruses”: An interesting feature of the index is that it need not be specific to bacteria; one could imagine applying it to health-prevalent vs. -scarce viruses or fungi.

Authors’ response: We definitely agree. Although we decided to keep things relatively simple and stick to the three domains of life (Archaea, Bacteria, and Eukarya), one item on our bucket-list is to do an equivalent study with viruses. Fungi were considered in our study, but just didn’t show up as part of the Health-prevalent or Health-scarce species sets.

- I. 124 “homogeneously dispersed”: Statistical test to support this claim?

Authors' response: Although our original claim here was intended to be only a brief and qualitative summary, we agree with the reviewer that a description based on statistics would provide more accurate and objective insight. For this, we used the Analysis of group similarities (ANOSIM), which is a non-parametric procedure based on a permutation test of among- and within-group similarities. ANOSIM is widely used in the ecology literature for testing the null hypothesis of 'no difference between two or more groups of entities' (the original paper by K. R. Clark in the Australian Journal of Ecology (1993) has been cited 7,000+ times; <https://onlinelibrary.wiley.com/doi/abs/10.1111/j.1442-9993.1993.tb00438.x>). Importantly, ANOSIM tests whether groups are significantly different from each other by using a dissimilarity matrix, and hence goes well with our PCoA plot based on Bray-Curtis distances.

The ANOSIM test statistic R (an index of relative within-group dissimilarity) is interpreted as a correlation coefficient and can be viewed as a measure of effect size. ANOSIM on the 13 total groups (1 healthy and 12 non-healthy phenotypes) resulted in an R of 0.21 ($P = 0.001$), allowing us to conclude that the groups differ only weakly, i.e., among- and within-group dissimilarities are not that much different for the most part. We've modified the main text in lines 132–134 accordingly:

- (previous) “In the same PCoA plot in which the healthy and twelve non-healthy phenotypes were presented simultaneously, we found that the various phenotypes were homogeneously dispersed without any apparent sub-clusters (Fig. 1d).”
- (now) “In the same PCoA plot in which the healthy and twelve non-healthy phenotypes were presented simultaneously (Fig. 1d), we identified no clearly distinguishable sub-clusters and found that, for the most part, only a weak difference amongst groups was observed (ANOSIM R = 0.21, P = 0.001).”

In addition, we've added the following to the last line of the legend in **Figure 1**: “Among- and within-group dissimilarities differ only weakly (ANOSIM R = 0.21, P = 0.001).”

- Figure 1 “outliers”: How were outliers defined?

Authors' response: This was originally mentioned in lines 691–694 (*'Sample-filtering based on taxonomic profiles'* subsection in **Methods**): “A sample was considered as an outlier, and thereby removed from further analysis, when its dissimilarity exceeded the upper and inner fence (i.e., > 1.5 times outside of the interquartile range above the upper quartile and below the lower quartile) amongst all dissimilarities. This process removed 67 metagenome samples.”

In order for us to more easily point this out to the reader, we've included “... (see **Methods**).” in the legend of **Figure 1** (see lines 142–143).

- l. 173 “ensemble-averaged”: I think this is true, but it required me thinking about it fairly carefully after two careful readings of this manuscript. So it might be something to explain a bit more.

Authors' response: We thank the reviewer for pointing to us another area wherein we could be more clear. We've slightly modified the text in lines 182–185, with the hope that our change has made the sentence more straightforward:

- (previously) “An important strength of our prevalence-based strategy for identifying microbial associations is that it does not compare ensemble-averaged measurements between the two groups, which is challenging to justify when biological and technical heterogeneity (batch effects) could vary greatly across various independent studies.”
- (now) “An important strength of our prevalence-based strategy for identifying microbial associations is that it does not compare averages of measurements taken from various sources, which is challenging to justify when biological and technical heterogeneity (batch effects) could vary greatly across independent studies.”

- I. 210: I think this is technically the definition for _balanced_ accuracy, rather than just “accuracy”.

Authors’ response: We agree with the reviewer that the correct technical term is “balanced accuracy”, which is often used for evaluating machine-learning processes with skewed class distributions (not to confuse with “overall accuracy”, which is another term with a different working definition). Throughout the text, we’ve replaced all instances of “average classification accuracy” with “balanced accuracy” (see highlighted areas).

- I. 222: Why does phylum fitting have a classification accuracy less than 50%? Is this because balanced accuracy is the metric? If this were plain old accuracy, I would expect a completely random classifier to have 50% accuracy.

Authors’ response: That is correct. Phylum achieved a balanced accuracy of 42.1% (Healthy: 61.7%; Non-healthy: 22.5%; the average: 42.1%).

- I. 222: I actually think it’s pretty interesting that the family-level index performs just about as well as the species-level index. Given the greater number of parameters at the species level, it seems like using families or something similar might be more “robust”, especially since species names change?

Authors’ response: We appreciate and understand the reviewer’s perspective. It is not incorrect. However, we view things slightly differently for the following reasons:

- As we’ve elaborated in the previous round, we very much value the ability to be as precise as possible when calling taxonomic clades. For example, if we were to use family-level features in our pipeline, and we were to identify a particular family as either ‘Health-prevalent’ or ‘Health-scarce’, is it reasonable to generalize this finding for all strains of that family? We certainly do not think so. Thus, if all things were considered equal (or highly similar), we’d like to be as close to strain-level analyses as practically possible. This is why, from the beginning, we chose to perform species-level analyses, and we were fortunate to find that species (amongst all taxonomic clades) gives the best balanced accuracy.
- We understand that robustness of a classifier depends upon how frequently the same features are selected when given more or different datasets to train upon; however, we prefer to have a classifier with high accuracy, as we believe that, with more and more datasets and samples, the identity of the classifiers (i.e., Health-prevalent and Health-scarce species) will eventually converge. Thus, simply restricting the analytical pipeline towards having fewer features to train upon is not necessarily a great idea in the long-term.

- We are aware that species names can change (e.g., renaming, merging two or more species into one, or splitting a single species into multiple ones). Obviously, these changes are done for scientifically valid reasons, and not out of whim. Thus, although the changing of species' names can present a bit of an inconvenience to future studies, we are totally ok with this and will acknowledge these changes into future versions of our index.

- I. 234: It seems to me that it should follow from the definition of h and the training of the classifier that $h > 0$ implies greater relative abundance of health-prevalent species. Does that logic only hold if increased prevalence is correlated with increased abundance? My point is that this might be more of a “sanity check” than a “result”.

Authors' response: We appreciate the reviewer's astute question, and sincerely apologize for the confusion arising from our use of the word “confirm”. We were not at all suggesting that there is a clear correlation between prevalence and abundance of health-prevalent species in healthy samples (and analogously for health-scarce species in non-healthy samples). We've modified the wording in line 243 (originally line 234 in the previous version of our manuscript) accordingly:

- (previous) “We were able to confirm higher relative abundance levels of Health-prevalent and Health-scarce species in the healthy and non-healthy group, respectively (**Supplementary Fig. 3**).”
- (now) “Interestingly, we found higher relative abundance levels of Health-prevalent and Health-scarce species in the healthy and non-healthy group, respectively (**Supplementary Fig. 3**).”

- I. 262: Prevalence is an important part of the algorithm, but I never saw (I might have missed it!) how “present” was defined. Does just 1 read from a species make the species “present”? Does changing that threshold change the behavior of the model? If tweaking that threshold makes accuracy change, does that mean the current model is overfitted, or that it's taking advantage of deep sequencing?

Authors' response: The definition of ‘present’ has already been described in lines 172 and 719: “...‘present’ (or relative abundance $\geq 1.0 \times 10^{-5}$) ...”. Establishing a minimum threshold allows us to determine which relative abundance we can use (or not use) in our pipeline; simply using all non-zero abundances is not a good idea, as one can never know whether a very minuscule abundance (which often results in deep sequencing data) actually reflects reality, or otherwise is simply attributed to noise in the detection software. Figuratively speaking, a line needs to be drawn somewhere in the sand.

A rank-plot (left) was used to identify a minimum relative abundance for a species to be considered as ‘present’. Specifically, we plotted all non-zero relative abundances from species detected across all 4,347 samples in our training dataset (there are a total of 369,073 points plotted in decreasing order). For most of the points, we see a gradual decrease (in relative abundance) going from left to right, but then we see a precipitous drop once 1.0×10^{-5} is passed, reflecting a transition in slope properties. Additionally, we do not see many points past 1.0×10^{-5} (relative to the amount prior to this inflection

point). Hence, we consider this quantity to be a reasonable minimum threshold for 'presence'.

To answer the last two questions, we've found the model's classification performance when the following 'presence' thresholds were used: 0.001%, 0.01%, 0.1%, and 1% (there is no practical reason to use higher thresholds, as we lose too much abundance data); the resulting balanced accuracies were: 69.7% (as reported in our manuscript), 70.5%, 67.5%, and 63.4%. In general, we see that increasing the 'present' threshold leads to lower accuracy, which was somewhat expected due to the ensuing loss in abundance information. To us, it would have been much more difficult to believe if there was no change in accuracy.

In the biomarker discovery field, 'overfitting' is more relevant to the context wherein a great disparity exists between classification performance on the training data and that on the validation data. Thus, we believe that a discussion on 'overfitting' is not relevant here.

- I found it difficult to interpret the cholesterol results. Is the fact that LDL level is correlated with the index mean something about biology (LDL is closely associated with the microbiome?) or about the metric (LDL is controlled by "balance" in the microbiome?) or about the data (only LDL has a strong enough signal to be detected?)?

Authors' response: This analysis was done in response to Reviewer #1's suggestion to see whether there is any connection between GMHI and objective measures of health. Accordingly, we looked for statistical associations between GMHI and well-recognized components of physiological wellness from clinical lab tests and health surveys, as reported in their original studies. As we have clearly reported, we found a moderately positive correlation between GMHI and HDLC (we are not sure why the reviewer refers to LDL). We view these findings to be of high importance, as it not only demonstrates the integration of clinical data with gut microbiome, but also hints at the possibility of GMHI serving as an effective and reliable predictor of cardiovascular health.

We kindly note that these results do not—and were never intended to—suggest anything about host biology (beyond the statistical associations reported herein); about a mechanistic relationship between HDLC and the gut microbiome; or about detection limitations of the blood measurements. We would consider remarks regarding these issues to be valid only if confirmed via experiment.

- I. 304: Estimates of Spearman's ρ should have confidence intervals.

Authors' response: We appreciate this suggestion. We've included the 95% confidence intervals (CI) for every Spearman's ρ mentioned in the manuscript. Please see lines 312, 324–326, 330, 409, 413–416 for the upper and lower confidence limits of each ρ .

- Figure 2: I think this is the same data in the two plots? Is there a way to rotate or align them so I can see that (b) is the marginal distribution of the points in (a)?

Authors' response: We appreciate this suggestion. Yes, the two plots in **Figs. 2a** and **2b** use the same data, but tell slightly different messages: **Fig. 2a** shows the correlation between the two variables, while **Fig. 2b**

shows that there is a significant difference in HDLC between the ‘GMHI-positive’ (i.e., classified as healthy) and ‘GMHI-negative’ (i.e., classified as non-healthy) groups. Since both plots are not necessarily meant to be viewed and analyzed in unison, we felt it best to leave the figure as-is. (actually after trying a few times, we found that the figure looks a bit weird.)

- I. 323: Cliff’s delta is a great metric here! I think this makes a big improvement to the interpretability. But I do think Cliff’s delta is likely unfamiliar to most readers; it could be good to include a one-sentence explanation of what it means (+1 means 100% of A points are greater than B points; 0 means 50%; -1 means 0%?).

Authors’ response: We agree that providing a brief explanation would help readers who are unfamiliar with Cliff’s delta. In lines 339–341 (right after when Cliff’s delta was first mentioned), we’ve included the following:

- “(Of note, Cliff’s Delta (d) is a non-parametric effect size measure that quantifies how often one value in one distribution is higher than the values in the second distribution; it is a difference between probabilities, and thus ranges from -1 to +1)”

- I. 333: Are there other metrics other than richness, Shannon, and 80% coverage considered?

Authors’ response: No additional ecological characteristics were considered. We felt that these were enough to get our main message across while abiding by space constraints.

- Figure 3: Does the index’s improved performance in any particular indication “drive” its improved performance? Like, if you removed disease X from the training set, would the index’s predictive ability decline more than you would expect simply because you shrunk the training set?

Authors’ response: We thank the reviewer for this interesting suggestion! Indeed, whether our Index’ performance in separating healthy from non-healthy could be driven by a single non-healthy phenotype is important to know. We performed the suggested analysis, wherein we removed each phenotype one-by-one from the training data, and repeated our entire pipeline (from finding new sets of θ_f and θ_d , to finding the corresponding Health-prevalent and Health-scarce species, and then testing our tuned index to identify the balanced accuracies on the sample-reduced training data). As shown below, we plot all the accuracies when removing each of the twelve phenotypes. Note: the orange line corresponds to the original 69.7% accuracy when all healthy and non-healthy samples were considered.

Despite some minor discrepancies (min: 68.4%; max: 72.6%) from 69.7%, we did not observe any particularly notable decrease or increase in GMHI’s classification performance. Thus, we find little evidence to conclude that a particular non-healthy phenotype totally “drives” the classification performance to be higher than any other metric tested and discussed throughout our study.

- I. 369-371: I didn’t understand this sentence

Authors’ response: We thank the reviewer for pointing this out. To avoid further confusion, we thought it would be better to simply omit this sentence rather than entirely reword a non-essential phrase.

- I. 401 “the most”: Implies it is more than any predictor of health status, when in fact the claim is just about the other 3 metrics the index was compared to

Authors’ response: We see the reviewer’s point. To clarify that we’re only considering four total metrics here, we’ve modified lines 418 and 419 accordingly:

- (previous) “Intra-study comparisons of stool metagenome ecological characteristics between healthy and non-healthy phenotypes show GMHI as the most robust and consistent predictor of general health status.”
- (now) “Intra-study analyses show GMHI as the most robust and consistent predictor of general health status compared to other ecological characteristics.”

- I. 445 “in 6 of 12”: multiple hypothesis correction required?

Authors’ response: Multiple hypothesis correction is not required for this analysis. Here, we are not testing for statistical associations of multiple features (i.e., hypotheses) simultaneously; rather, we are simply counting the

frequency of studies (among 12) whose case-control comparisons resulted in being significant. It would not make sense for us to treat each study as an independent feature or hypothesis.

- I. 473: I found it very surprising that random forest, which is a highly flexible algorithm with access to the same information as the index (e.g., the species names are not “blinded” in the way they are with richness or other diversity metric), performed worse than the index. I think this has some implications about how random forest, essentially a set of decision trees, doesn’t have the same format as the rational equation approach of the index, and that’s what gave improved performance to the index over random forests?

Authors’ response: We kindly note that, in the original lines 472–474 (now lines 488–490 in the revised manuscript), wherein we write: “In regards to average classification accuracies on the training data, the classifiers based upon Health-prevalent species ($\chi = 66.3\%$) and Shannon diversity ($\chi = 53.6\%$) performed comparable to, or much worse than, GMHI ($\chi = 69.7\%$);”, we are clearly alluding to accuracies from using the Health-prevalent species and Shannon diversity, as the reviewer had previously instructed. Actually, the Random Forests classifier achieves a 98.5% balanced accuracy, but does quite poorly on the validation set (53.3% balanced accuracy). Obviously this steep drop is a classic result of over-fitting, but investigating the manifold reasons as to why seems outside the scope of this manuscript.

- I. 478: Rather than classification accuracy, the random forest’s out of bag (OOB) error should be reported, since that is the value that accounts for overfitting.

Authors’ response: We thank the reviewer for this suggestion. For the Random Forests classifier, there are indeed several methods to report model performance, including the OOB error/estimate. If this manuscript were to be solely about using Random Forests, we would certainly be open to using more sophisticated methods than the balanced accuracy. However, given that we would like to compare apples-to-apples as much as possible, we think it’s best to remain consistent, and thus report the same type of accuracy for all classifiers used throughout our manuscript (e.g., GMHI using all taxonomic clades, Shannon diversity, 80% coverage, richness, Health-prevalent species, Random Forests).

- I. 485: effect size?

Authors’ response: We thank the reviewer for spotting this. We’ve placed all effect sizes (Cliff’s Delta values) into **Supplementary Table 14**, as including them all in **Figure 6**, or in the main text, makes things look incredibly messy. Please see lines 501, 528, 565, and 571 in the revised manuscript.

- I. 558 "clinical need": I appreciate that the authors, who are actually doctors, know better than I do about what the clinical need is! I would, however, appreciate a citation to that effect for the benefit of other readers.

Authors’ response: We thank the reviewer for pointing this out. We agree that providing relevant citations would help establish the importance of our claim. We’ve provided the following references that motivate the need for our type of study:

- Allaband, C. et al. Microbiome 101: Studying, Analyzing, and Interpreting Gut Microbiome Data for Clinicians. Clin. Gastroenterol. Hepatol. 17, 218 (2019).

- Staley, C., Kaiser, T. & Khoruts, A. Clinician Guide to Microbiome Testing. Dig. Dis. Sci. 63, 3167–3177 (2018).
- McBurney, M. I. et al. Establishing What Constitutes a Healthy Human Gut Microbiome: State of the Science, Regulatory Considerations, and Future Directions. J. Nutr. 149, 1882–1895 (2019).
- Hagerty, S. L., Hutchison, K. E., Lowry, C. A. & Bryan, A. D. An empirically derived method for measuring human gut microbiome alpha diversity: Demonstrated utility in predicting health-related outcomes among a human clinical sample. PLoS One 15, (2020).

We felt that this part should be mentioned earlier, rather than being buried in the Discussion. Therefore, we've moved the following text from the Discussion to the Introduction (see lines 45–48):

- “As researchers uncover more details regarding which gut commensals may play a significant role in host health and disease, a promising translational application of this knowledge would be towards developing analytical tests or quantitative methods that provide indication of one's health based upon a gut microbiome snapshot^{9–12}.”

- I. 630: This could use a citation too. I'm sure there are many, but this one comes to mind: <https://pubmed.ncbi.nlm.nih.gov/31683111/>

Authors' response: We thank the reviewer for this very suitable reference. We've modified the text in lines 647–650 as follows:

- **(previous)** “Certainly, for future works, we plan to iteratively expand our application to encompass broader ranges of subjects, including those from under-developed countries and minority ethnicities/races.”
- **(now)** “Certainly, for future works, we plan to iteratively expand our application to encompass broader ranges of subjects, including those from under-developed countries and minority ethnicities/races, to better understand microbiome diversity and foster inclusion in microbiome research⁶⁴.”

- I. 636-641: I think this is a very clear articulation of the paper's motivation, and I would have preferred to read it in the Introduction!

Authors' response: As suggested, we've moved this section to the Introduction. Please see lines 50–59 in the revised manuscript.

Reviewers' Comments:

Reviewer #2:

Remarks to the Author:

The authors have addressed my concerns. Apologies for asking questions that are already addressed in the text (e.g., the definition of "present").

As a final thought/suggestion, the relative performance (and I say "performance" rather than "accuracy") of the index is, to me, convincingly shown by the comparison of accuracy of the various classifiers on the training and validation data sets. Right now this information is in a few places in the text (mostly lines 486-496), and I made myself this little table:

Classifier / Training accuracy / Validation accuracy

GHMI / 69.7% / 77.1%

Healthy / 66.3% / 59.3%

Shannon / 53.6% / 47.0%

Random / 98.5% / 52.3%

forest

It might be helpful to a reader to have all the different accuracies in one single (supplemental) table, to emphasize that GHMI performs best on the training set (with the exception of random forest) and is not subject to overfitting (which random forest definitely is, if the normal accuracy, rather than OOB error, is used).

REVIEWER COMMENTS

Reviewer #1 (Remarks to the Author):

None

Reviewer #2 (Remarks to the Author):

The authors have addressed my concerns. Apologies for asking questions that are already addressed in the text (e.g., the definition of "present").

Authors' response: We very much thank the reviewer for each and every constructive feedback for our manuscript! We are pleased to hear that all concerns have been adequately addressed.

As a final thought/suggestion, the relative performance (and I say "performance" rather than "accuracy") of the index is, to me, convincingly shown by the comparison of accuracy of the various classifiers on the training and validation data sets. Right now this information is in a few places in the text (mostly lines 486-496), and I made myself this little table:

Classifier	Training accuracy	Validation accuracy
GHMI	69.7%	77.1%
Healthy	66.3%	59.3%
Shannon	53.6%	47.0%
Random forest	98.5%	52.3%

It might be helpful to a reader to have all the different accuracies in one single (supplemental) table, to emphasize that GHMI performs best on the training set (with the exception of random forest) and is not subject to overfitting (which random forest definitely is, if the normal accuracy, rather than OOB error, is used).

Authors' response: We welcome any opportunity to improve the clarity and readability of our manuscript. In response to the reviewer's final suggestion, we have provided **Supplementary 8** as a way to summarize all accuracies mentioned throughout the manuscript. Also, we introduce this table in lines 369–370 accordingly: "In **Supplementary Table 8**, we provide a summary of all accuracies for classifying healthy vs. non-healthy by the various classifiers reported in this study."